# Seasonal variations of the highly time-resolved aerosol composition, sources, and chemical processes of background submicron particles in North China Plain

Jiayun Li[1, 4], Liming Cao[2], Wenkang Gao[1], Lingyan He[2]; Yingchao Yan[1,4], Yuexin He[1], Yuepeng Pan[1], Dongsheng Ji[1], Zirui Liu[1], Yuesi Wang[1,3,4]

[1]State Key Laboratory of Atmospheric Boundary Layer Physics and Atmospheric Chemistry (LAPC), Institute of Atmospheric Physics, Chinese Academy of Sciences, Beijing 100029, China

[2]Key Laboratory for Urban Habitat Environmental Science and Technology, Peking University Shenzhen Graduate School, Shenzhen, 518055, China

[3]Center for Excellence in Regional Atmospheric Environment, Institute of Urban Environment, Chinese Academy of Sciences, Xiamen 361021, China

[4]University of Chinese Academy of Sciences, Beijing 100049, China

Correspondence: Zirui Liu (liuzirui@mail.iap.ac.cn); Lingyan He (hely@pkusz.edu.cn)
Jiayun Li and Liming Cao contributed equally to this work.

**Abstract.** For the first time in the North China Plain (NCP) region, we investigated the seasonal variations of submicron particles (NR-PM$_1$) and their chemical composition at a background mountainous site of Xinglong using Aerodyne high-resolution time-of-flight aerosol mass spectrometry (HR-ToF-AMS). The average concentration of NR-PM$_1$ was highest in autumn (15.1 μg m$^{-3}$) and lowest in summer (12.4 μg m$^{-3}$), with a greater abundance of nitrate in spring (34%), winter (31%) and autumn (34%), and elevated organics (40%) and sulfate (38%) in summer. PM$_1$ in Xinglong showed higher acidity in summer and moderate acidity in spring, autumn and winter, with average pH values of 2.7 ± 0.6, 4.2 ± 0.7, 3.5 ± 0.5 and 3.7 ± 0.6, respectively, which is higher than those estimated in the United States and Europe. The size distribution of all PM$_1$ species showed a consistent accumulation mode peaking at approximately 600–800 nm (dva), indicating a highly aged and internally mixed nature of the background aerosols, which was further supported by the source appointment results using positive matrix factorization and multilinear engine analysis. Significant contributions of aged secondary organic aerosol (SOA) in organic aerosol (OA) were resolved in all seasons (> 77%), especially in summer. The oxidation state and the process of evolution of OAs in the four seasons were further investigated, and an enhanced carbon oxidation state (−0.45−0.10) as well as O/C (0.54–0.75) and OM/OC (1.86–2.13) ratios—compared with urban studies—were observed, with the highest oxidation state appearing in summer, likely because of the relatively stronger photochemical processing that dominated the formation processes of both less oxidized OA (LO-OOA) and more oxidized OA (MO-OOA). Aqueous-phase processing also contributed to the SOA formation and prevailed in winter, the role of which to MO-OOA was more important than that to LO-OOA. In addition, regional transport also played an important role in the variations of SOA, especially in summer that continuous increases in SOA concentration as a function of Ox was found to be associated with the increases of wind speed. Furthermore, backward trajectory analysis showed that higher concentrations of submicron particles were associated with air masses transported short distances from the southern regions in all four seasons, while long-range transport from Inner Mongolia

(western and northern regions) also contributed to summertime particulate pollution in the background areas of the NCP. Our results illustrate that the background particles in the NCP are influenced significantly by aging processes and regional transport, and the increased contribution of aerosol nitrate highlights how regional reductions in emissions of nitrogen oxide are critical for remedying occurrence of nitrate-dominated haze events over the NCP.

## 1. Introduction

With rapid industrialization, population expansion and urbanization, the North China Plain (NCP) has been seriously polluted in recent years (Tao et al., 2012;Du et al., 2015;Yuan et al., 2015;Zhao et al., 2019). The formation mechanisms of particulate pollution are complex because of the unfavorable meteorological conditions, complex source emissions, and geographical conditions. For example, sulfate dominates the secondary inorganic aerosols (SIAs) in heavy-industry cities such as Shijiazhuang and Handan, while in recent years nitrate has dominated those in Beijing because of the strict emissions reduction measures for coal combustion (Huang et al., 2018; Li et al., 2019a; Duan et al., 2019). High relative humidity (RH) favors heterogeneous reactions and hygroscopic growth, leading to an increase in secondary aerosols and further aggravating haze pollution over the NCP (Sun et al., 2013a; Liu et al., 2016). Moreover, haze pollution over the NCP can be exacerbated by its unique topography. Hu et al. (2014) found that heat from the Loess Plateau can be transported to the plain with westerly airflow, resulting in enhanced thermal inversion and suppression of the planetary boundary layer (PBL), thus weakening atmospheric diffusion. Furthermore, the southern NCP is an important pathway for water vapor and pollutant transport in the PBL because of the blocking effect of the Taihang Mountains (Tao et al., 2012).

Nearly all previous research on the characterization of fine particles over the NCP has been conducted at heavily polluted urban or suburban stations with strong local source emissions, with a few studies having been deployed at background stations. Early studies at background sites of the NCP mainly focused on average chemical compositions, source analysis, and the influence of regional transportation (Pan et al., 2013; Liu et al., 2018; Huang et al., 2017), which indicated that secondary aerosols dominate the aerosol particles at background sites and that regional transport affects the air pollution of the background atmosphere. However, these studies at background sites in the NCP were limited by a low temporal resolution of one or several days. High-resolution time-of-flight aerosol mass spectrometry (HR-ToF-AMS) has been widely used to characterize nonrefractory submicron particles (NR-PM$_1$) at numerous urban sites. However, the high time resolution studies at background sites in the NCP by HR-ToF-AMS are still limited. Zhang et al., (2017) investigated the NR-PM$_1$ species in 2015 in a background site (Shangdianzi) by a HR-ToF-AMS and Li et al., (2019b) conducted the observation in winter 2018 in a background site (Xinglong) by a Quadrupole AMS (Q-AMS) in the NCP. Both studies showed the dominant role of organics and nitrate and organics were highly oxidized in the background atmosphere in the NCP. However, both of the two studies were concentrated in wintertime, and the compositions of PM$_1$, the sources and oxidation state of organic aerosol (OA)

changed significantly in different background sites and seasons. For example, nitrate and OA dominated $PM_1$ in autumn and winter at Xinglong and Waliguan sites in the NCP and in spring at Mount Wuzhi site in Northern China (Zhu et al., 2016), while sulfate and OA dominated $PM_1$ in summer at Waliguan site in Western China (Zhang et al., 2019) and in spring at Lake Hongze site in Southern China (Zhu et al., 2016). Organics were highly oxidized at these background sites, while the oxidation state of organics in southern China were higher than that in Northern China (Zhu et al., 2016). Hydrocarbon-like OA (HOA) accounted for 30% in OA in spring at Wuzhi Mount in northern China. Biomass burning OA and aged biomass burning OA accounted for 18% and 40% in OA in summer, respectively, while HOA only accounted for 6%, in western China (Zhang et al., 2019). No primary organic sources were found in Southern China in spring (Zhu et al., 2016). Theses substantial differences highlighted the importance of long-term observations for understanding the seasonal characteristics of aerosol species and sources in the background atmosphere. What's more, pollutants in the background areas are frequently influenced by the regional transport from the urban areas, highlighting the importance of the seasonal variations of the impacts of regional transport on aerosol mass loadings and chemistry at background site. However, no systematic measurements with high time resolution of the mass–size distributions of chemical components in fine aerosol particles, covering four seasons, have yet been reported in the background atmosphere in the NCP, which would hinder our understanding on the evolution and chemical processes of secondary aerosol and regional transport on a regional scale. Only a study investigated the seasonal variations in $PM_1$ species at the background site in central east China at Mount Tai (Zhang et al., 2014b) based on unit mass resolution (UMR). The study only presented the characterization of the total OA; the sources of OA were not identified. What's more, there was no elemental information, which can determine the oxidation state of OA and characterize the evolution and processes of secondary organic aerosol (SOA). Actually, the evolution and formation mechanisms of SOA in the background atmosphere are still unclear. Currently, research focusing on this field tends to be concentrated in urban areas, and the results vary in different places and seasons. For example, photochemical processing was found to dominate the oxidation state of OA in haze events, whereas aqueous-phase processing was the main reason during foggy events in Hong Kong (Li et al., 2013; Qin et al., 2016). In urban Beijing, Xu et al. (2017) found that aqueous-phase processing dominated MO-OOA (more oxidized OA) formation in all seasons. Meanwhile, more recently, Li et al (2020) found that the impact of photochemistry on MO-OOA formation enhanced as the photochemical age increased in early autumn in Beijing. Due to the stronger atmospheric oxidizing capacity and higher oxidation state of organics in the background atmosphere than in the urban atmosphere over the NCP, the evolution and formation mechanisms of SOA would be largely different from those of urban areas, mainly due to the complex interactions of local emissions, chemical reactions, and meteorological influences. Whereas, these kinds of studies were still limited and hinder our understanding of background aerosol chemistry in polluted regions such as the NCP. Therefore, we present one year-round measurements of submicron aerosols at a regional background station in NCP, to explore the seasonal variations in aerosol sources and formation processes. Especially, the influence of photochemical and aqueous-phase processing on SOA productions [LO-OOA (less oxidized OA) and MO-OOA] were evaluated based on robust data analyses.

In this study, an HR-ToF-AMS was first deployed with instruments for the measurement of meteorological parameters and gaseous parameters during all four seasons at the Xinglong background station to investigate the seasonal variations in $PM_1$ species in the background atmosphere of the NCP. The seasonal variations in the submicron aerosols, including the variation in mass concentration, chemical composition, aerosol acidity, size distribution, diurnal variation, and meteorological

effects, are presented in detail in this paper. The seasonal sources, oxidation state, and evolutionary processes of LO- and MO-OOA production are fully explored. Finally, backward trajectory analyses were performed to investigate the different pathways and the regional transport influences on the background atmospheric aerosols during the four seasons over the NCP.

## 2. Experimental methods

### 2.1 Sampling sites

Xinglong background station is located in the north of Hebei Province, south of the Yanshan Mountains, 960 m above sea level, at a longitude and latitude of 117.67° and 40.40° respectively, about 115 km northeast of Beijing (Pan et al., 2013). Since Xinglong station is surrounded by forests and there are no serious pollutant emissions in this area, it can be considered as an ideal station to investigate haze episodes over the NCP on a regional scale. More details about Xinglong station can be found in Li et al. (2019b) and Tian et al. (2018).

**2.2 Instrumentation and operation**

From March to December 2019, an HR-ToF-AMS was deployed to measure the mass concentrations and chemical compositions of $NR-PM_1$. The sampling periods were from May 1 to 31, June 20 to July 26, October 12 to November 12, and November 25 to December 25, 2019. The ambient particles were sampled into the AMS through a URG cyclone (URG-2000-30ED) for removing coarse particles with size cutoffs of 2.5 μm, which was followed by use of a Nafion dryer to dry the

115 sampled aerosols to eliminate the impact of high humidity on particles. During these four campaigns, both the "V" and "W" modes were operated, and the temporal resolution was 3 min. The HR-ToF-AMS calibrations were carried out in strict according to the standards reported in previous studies (Jimenez, 2003;Zhang et al., 2014a).

Simultaneously, other measurements were also deployed during the whole campaign. Specifically, a Sharp-5030 instrument was used to measure the total $PM_1$ concentration. Gaseous species, including ozone ($O_3$), nitric oxide (NO), nitrogen

dioxide ($NO_2$), carbon monoxide (CO) and sulfur dioxide ($SO_2$), were measured using Thermo gas analyzers. A Milos520 (Vaisala, Finland) was used to obtain the meteorological parameters. An ammonia ($NH_3$) analyzer ($NH_3-H_2O$, Model 911-0016, LGR) was also used to simultaneously measure related gases. More details about the instruments can be seen in Li et al. (2019b).

**2.3 Data analysis**

125 The analysis software known as SQUIRREL (v1.57H) and PIKA (v1.16H) were used to analyze the size-resolved mass concentrations and mass spectra of OA, respectively. According to Canagaratna et al. (2015), the improved-ambient method was used to obtain the elemental compositions ratios. In this study, the relative ionization efficiency (RIE) values used were 1.1, 1.2, 1.3 and 1.4 for nitrate, sulfate, chloride and organics, respectively. RIE value of 4.0 were used for ammonium based on the ionization efficiency calibration results in each season. The particle collection efficiency (CE) was applied to account

130 for the incomplete detection of particles due to particle bounce (Aiken et al., 2009). According to Middlebrook et al. (2012), the CE value can be affected by the RH, aerosol acidity, and the ammonium nitrate mass fractions (ANMF). The ambient aerosols were dried using a Nafion dryer. Meanwhile, aerosols were neutral in spring, autumn and winter, and weakly acidic in summer (Fig. 2). Therefore, RH and aerosol acidity could not influence the CE values in all seasons. However, the ANMF values were mostly above 0.4 in spring, autumn and winter, indicating that $NH_4NO_3$ would substantially affect the CE values

135 in these three seasons. Through the combined analysis of the above three aspects, a constant CE of 0.5 was used in summer, and the CE values in the other three seasons were calculated according to Middlebrook et al. (2012) [CE = max (0.45, $0.0833 + 0.9167 \times$ ANMF)].

 The Positive matrix factorization (PMF) Evaluation Tool PET (v3.04A; Ulbrich et al., 2009; Zhang et al., 2011) was employed for the OA source apportionment in each season. The error matrix was modified; ions with low signal-to-noise ratios

140 were down-weighted or removed. It is common to use PMF analysis to identify the sources of OA. However, it is difficult to distinguish similar factors in areas with complex pollution sources. Owing to the high fraction of OOA in OA, it is extremely difficult to separate primary OA (POA) from oxygenated OA (OOA) in Xinglong using PMF analysis, because the POA factor is easily mixed with the OOA factor. For example, for spring, according to the PMF analysis (Figs. S2–S6), in the two- to four-factor solutions, POA factors were mixed with OOA factors because the HOA profile contained a higher-than-expected

145 contribution from m/z 44. In the 5-factor solution, a POA factor appeared, while OOA was over-split, some of which showed similar characteristics. The POA was finally identified as fossil fuel OA (FFOA), which is a typical profile in Xinglong. Details about the diagnosis information can be seen in Sect. 3.2 and in the supplementary material. In autumn, the solution of the PMF analysis was similar to with that in spring; a POA factor appeared until the 7-factor solution and OOA was over-split. The POA factor in autumn was also identified as FFOA. We constrained the FFOA profile separated by the 5-factor solution of PMF

150 analysis in spring, as well as the FFOA profile separated by the 7-factor solution of the PMF analysis in autumn, to better separate FFOA from OOA in spring and autumn, respectively. The a value of 0-0.5 with a space of 0.1 in each season was used to constrain the FFOA profiles to explore the solution space (Canonaco et al., 2013). As a result, three OA factors, including FFOA, LO-OOA and MO-OOA, were identified with ME-2 analysis in spring and autumn. In winter, the 3-factor solution (POA, LO-OOA and MO-OOA) from PMF analysis was good enough, so it was not necessary to use ME-2 analysis to separate

POA from OOA. In summer, no POA factor appeared in the 2- to 9-factor solutions of the PMF analysis, suggesting the faction of POA in OA in summer was too low to be identified. Therefore, the PMF analysis results of two OA factors (LO-OOA and MO-OOA) were used. Details of the diagnosis information can be seen in Sect. 3.2 and the supplementary material (Figs. S2–S20, Tables S1 and S2).

**2.4. Backward trajectory modeling**

The 48-h back trajectories were calculated every hour at a height of 200 m (above ground level) using the HYSPLIT (Hybrid Single-Particle Lagrangian Integrated Trajectories) model, version 4.8, in each season. Meteorological data archived by the Air Resource Laboratory, NOAA, were used, the resolution of which was $1° × 1°$. The cluster analysis algorithm was used to classify the back trajectories of each season.

**2.5. Aerosol acidity estimation**

Aerosol pH was predicted using the aerosol thermodynamic model ISORROPIA II, according to the equation as follows (Guo et al., 2015;Guo et al., 2017):

$$\text{pH} = -\log_{10} H_{aq}^+ = -\log_{10} \frac{1000 H_{air}^+}{W_i + W_o}, \qquad (1)$$

where $H_{air}^+$ is the H⁺ concentration per volume of air and can be obtained from the ISORROPIA II model. $W_i$ and $W_o$ are the liquid water content (LWC) of inorganic and organic components, respectively. $W_i$ and $H_{air}^+$ can be obtained by the ISORROPIA II model. A previous study in northern China showed that organic particle in water account for only 5% of the total LWC (Liu et al., 2017), so $W_o$ was not calculated in this study. The forward mode was used with just aerosol-phase data and NH₃ data input in this study, to avoid measurement error (Song et al., 2018;Guo et al., 2017). Notably, data for RH < 30% and RH > 95% were excluded because of the large uncertainty in LWC and pH values (Ding et al., 2019; Guo et al., 2015).

**3 Results and discussion**

**3.1 Mass concentrations of PM₁ species**

**3.1.1 Seasonality of the chemical composition of PM₁**

The annual mean mass concentrations of organics, nitrate, sulfate, ammonium, and chloride in PM₁ were $4.6 \pm 3.8$, $3.7 \pm 5.7$, $3.2 \pm 3.3$, $2.0 \pm 2.3$, and $0.1 \pm 0.1$ µg m⁻³, respectively, totaling $12.9 \pm 14.1$ µg m⁻³. This total PM₁ concentration was much lower than the values observed in urban and suburban areas in the NCP, such as 81 µg m⁻³ in urban Beijing (Hu et al., 2017), 187 µg m⁻³ in urban Handan and 178 µg m⁻³ in urban Shijiazhuang in Hebei Province in winter (Li et al., 2017; Huang et al., 2018), and 52 µg m⁻³ at the suburban station of Gucheng (Zhang, 2011). However, it was higher than those in national background areas in eastern and western China, such as 9.1 µg m⁻³ at Waliguan background station in summer, 4.4 µg m⁻³ at

the southern edge of the Tibetan Plateau in spring, and 10.9 μg m$^{-3}$ at Mount Wuzhi station in spring (Zhang et al., 2019; Zhu et al., 2016). The higher PM$_1$ mass concentration at Xinglong in the background atmosphere of the NCP compared to those in other remote areas in eastern and western China indicated the air pollution over the NCP is serious.

Seasonally, the average PM$_1$ concentrations were 13.7 ± 16.0, 12.4 ± 7.4, 15.1 ± 18.7, and 14.1 ± 13.3 μg m$^{-3}$ in the four seasons, respectively. OA showed the highest portion in NR-PM$_1$ in summer, accounting for 40% by mass. Nitrate was the highest SIA component in spring (34%), winter (31%), and autumn (34%). The low percentage of nitrate in summer (9.6%) might be attributable to the higher temperatures than in other seasons, which suppress the partitioning to particulate nitrate (Seinfeld and Pandis, 2016). Sulfate remained relatively low in spring (16%), autumn (21%), and winter (19%), but it increased to 38% of the NR-PM$_1$ mass in summer. Ammonium accounted for 13–17% of PM$_1$ concentrations in all four seasons.

As shown in Fig. 1, the proportions of OA in PM$_1$ gradually decreased as PM$_1$ increased in all seasons, suggesting that the enhanced SIA dominated the increase in PM$_1$, similar to the findings in previously reported research (Hu et al., 2017; Zhang et al., 2019). The proportions of nitrate in PM$_1$ increased slightly in spring, summer and autumn, corresponding to the increase in RH and suggesting that aqueous-phase reactions could be conducive to nitrate production. It is worth noting that the RH in spring was generally lower than that in autumn and winter. For example, when the PM$_1$ concentration was higher than 40 μg m$^{-3}$, the RH was above 40% in autumn and winter, but below 40% in spring. This might be attributable to the frequent dust events in spring, which are often accompanied by dry and cold air from northern regions. As a result, aqueous-phase reactions might be more conducive to nitrate formation in autumn and winter than in spring. In winter, the decreased percentage of nitrate with a high PM$_1$ concentration (PM$_1$ > 50 μg m$^{-3}$) was due to the increase in sulfate because of the coal emissions during the heating season. The proportions of ammonium remained stable in all seasons even when the PM$_1$ concentration was low, suggesting that ammonia is excessive over the NCP. Although the average PM$_1$ concentrations in the four seasons were similar (Table 1), the frequency distribution of PM$_1$ (Fig. 1, white curve) showed strong seasonal dependency. High-frequency and extremely-low-frequency PM$_1$ concentrations were observed when the PM$_1$ concentration was below 10 μg m$^{-3}$ and above 40 μg m$^{-3}$, respectively, in spring, autumn and winter. In summer, the frequency distribution of PM$_1$ did not change dramatically as PM$_1$ increased.

### 3.1.2 Seasonality of aerosol acidity

The acidity of PM$_1$ was evaluated in each season using the thermodynamic model ISORROPIA-II (Table 2). PM$_1$ in Xinglong showed moderate acidity in spring, autumn and winter, with average pH values of 4.2 ± 0.7, 3.5 ± 0.5 and 3.7 ± 0.6, respectively. Comparatively, the pH value in summer was the lowest (2.7 ± 0.6) among all seasons, similar to the findings of previous studies (Ding et al., 2019; Liu et al., 2017). The seasonal variation in the pH at Xinglong was similar to results reported in urban Beijing, except for spring and winter. The pH value in urban Beijing was highest in winter, followed by spring (4.4 ± 1.2), autumn (4.3 ± 0.8), and summer (3.8 ± 1.2) (Ding et al., 2019). The seasonal variation of pH in this study

was strongly related to the chemical composition of aerosols in each season. Previous studies show that, compared to an elevated nitrate concentration, an elevated sulfate concentration can result in higher acidity because of the low volatility of sulfate (Tan et al., 2018; Xu et al., 2019). In this study, the mass fraction of sulfate in $PM_1$ was highest in summer (37%) and lowest in spring (16%). Similarly, the nitrate-to-sulfate ratio was highest in spring (2.13) and lowest in summer (0.26). Recent studies have shown that sulfate has been effectively reduced in Beijing because of the strict emission control measures, and the mass fraction of nitrate in PM has increased significantly, with an increased $NO_3/SO_4$ ratio in Beijing ubiquitously observed (Xu et al., 2019; Song et al., 2019).

Notably, the pH in spring (4.2 ± 0.7) was similar to the value found in urban Beijing (4.4 ± 1.2) in the same season. In comparison, the pH values in autumn and winter were 0.5 to 1.1 lower than those found in urban areas of northern China, such as 4.3 ± 0.8 in autumn in Beijing, 4.5 (3.8–5.2) in winter in Zhengzhou, and 4.8 (3.9–5.9) in winter in Anyang (Ding et al., 2019; Wang et al., 2020). The higher LWC in urban areas may be one of the important reasons for its slightly lower acidity in autumn and winter compared to that in background areas in northern China. Aerosol acidity is closely related to LWC, with higher LWC usually accompanied by higher aerosol pH according to previous studies (Guo et al., 2015; Liu et al., 2017). In this study, the LWC in autumn and winter was 18 ± 38 and 12 ± 26, respectively, which was obviously lower than that in urban areas, such as 109 ± 160 µg m$^{-3}$ in autumn in Beijing (Ding et al., 2019), 220 (28–711) µg m$^{-3}$ in winter in Beijing (Liu et al., 2017), and 95 µg m$^{-3}$ in winter in Zhengzhou (Wang et al., 2020). The higher LWC in urban areas is mainly due to the high aerosol concentrations, which can enhance aerosol water uptake (Liu et al., 2017). In comparison, the $H_{air}^+$ concentration was 3.7 µg m$^{-3}$ in autumn and 1.2 µg m$^{-3}$ in winter, which were comparable with those observed in urban areas. Therefore, the lower LWC/$H_{air}^+$ ratio in Xinglong favored the slightly lower pH values, according to equation (1).

Moreover, the pH values in this study and previous studies in urban areas of the NCP were 0.8 to 3.5 units higher than those observed in the U.S. and Europe, such as 1.2 ± 1.1 in Crete, Greece, in winter (Bougiatioti et al., 2016); 0.9 ± 0.6 in Alabama, southeastern U.S., in summer (Guo et al., 2015); and 2.2 ± 0.6 in Yorkville, southeastern U.S., in autumn (Nah et al., 2018). The excessive $NH_3$ emissions in the NCP play an important role in the large gap (Song et al., 2019). In this study, the average $NH_3$ concentrations were 12, 19, 8 and 4 ppb in each season, with a maximum value of 39 ppb, while in the southeastern U.S., $NH_3$ generally ranged between 0.1 and 3.0 ppb (Weber et al., 2016). Another explanation might be the changes in chemical composition of aerosols over the NCP. The $NO_3/SO_4$ ratios in this study and urban areas of the NCP were obviously higher than those in other countries (Table 2). Thus, the relatively lower aerosol acidity in the NCP might be attributable to the excessive $NH_3$ and high $NO_3/SO_4$ ratios on a regional scale in this region.

### 3.1.3 Seasonality of meteorological effects on $PM_1$ species

The chemical composition of $PM_1$ exhibited distinct characteristics in the four seasons, which was due to the significant seasonal variation in meteorological conditions and emissions. As shown in Table 2, NOx exhibited its highest concentration

in winter and correlated well with chloride ($R^2 = 0.6$), suggesting a strong influence of fossil fuel combustion, such as coal combustion, while regional transport from heavily polluted regions may also contribute partly. $SO_2$ concentrations were low in all seasons and showed no obvious seasonal changes (1.0–1.9 ppb). The $O_3$ concentration was highest in summer, likely due to the high temperatures and enhanced photochemical processing.

Horizontal wind speed can affect the diffusion and transportation of pollutants. In spring and winter, with the increase in wind speed, the concentrations of almost all $PM_1$ species decreased, suggesting an impact of the dilution of winds on atmospheric aerosols (Fig. 2). However, the wind dilution ratios (percentage decrease in the concentration of the aerosol species for every 1 m s$^{-1}$ decrease in wind speed) were much lower than the observed value in urban Beijing in winter (Li et al., 2019b), but comparable to the value observed at a rural station (Wang et al., 2016), suggesting that aerosols in the background atmosphere were homogeneously distributed. Therefore, the winds showed a weaker influence in terms of diluting aerosol particles than the result observed in Beijing. In autumn, $PM_1$ species only decreased rapidly when wind speed was > 4 m s$^{-1}$, while SIAs increased rapidly from 1 to 4 m s$^{-1}$, suggesting a significant role of intermediate wind speed in SIA transport, similar to the findings of a previous study conducted in autumn in Xinglong (Li et al., 2019b). In summer, all $PM_1$ species decreased gradually as the wind speed increased, except OA and sulfate, suggesting a strong influence of regional transport on OA and sulfate formation, especially the former. The relationship between pollutants and wind direction in summer also differed from that in other seasons. All $PM_1$ species showed high concentrations in association with wind from the southern regions and low concentrations in association with wind from the northern regions, in spring, autumn, and winter. In summer, $PM_1$ species also showed relatively high concentrations in association with wind from the northeast regions. The results here might indicate an effect of regional transport from southern, heavily polluted regions on atmospheric aerosols at regional background sites of the NCP in all seasons, and that northern transport might also partially contribute in summer.

As shown in Fig. 2, we also investigated the effects of RH on the secondary aerosols. When RH was < 80%, SIAs (especially nitrate) increased significantly as RH increased in autumn and winter, suggesting a significant effect of aqueous-phase reactions on SIA formation. Previous studies in urban Beijing show a successive increase in SIA with increased RH (Li et al., 2019b; Liu et al., 2016), suggesting that aqueous-phase processing affects nitrate formation in both urban and background atmospheres. The wind speed also increased rapidly as RH increased from 60% to 80%, and then maintained a high level in autumn in Xinglong, while it has been found to continually decrease with increased RH at urban sites of the NCP (Huang et al., 2018; Li et al., 2019b). This behavior further indicates regional transport has more of an influence on the SIA concentration in the background atmosphere than in the urban atmosphere over the NCP in autumn. In spring, SIA only increased significantly at moderate RH levels as RH increased (< 60%), suggesting a weaker impact of aqueous-phase processing on SIA formation in spring than in autumn and winter. When RH was > 60%, the SIA concentrations decreased rapidly, and this decrease was accompanied by a rapid decrease in wind speed, suggesting the impact of regional transport also weakened.

Notably, the OA and sulfate concentrations were high even when RH was low (RH < 40) in summer, which was significantly different from what occurred in other seasons, suggesting an impact of photochemistry on the formation of sulfate and OOA. Furthermore, OA did not increase as RH increased in summer, suggesting different formation mechanisms of OA in summer than in other seasons, which is specifically investigated in Sect. 3.4.

### 3.1.4 Seasonality of the size distribution of the chemical components of PM$_1$

The size distribution of all PM$_1$ species in each season concentrated in accumulation mode (Fig. 3), with a peak diameter at approximately 600–800 nm (dva), indicating aerosols in the background atmosphere were highly aged and internally mixed (Jimenez, 2003). Compared to Beijing, OA in Xinglong had a larger peak diameter and a wider size distribution in each season (Hu et al., 2017). Compared to SIA, OA always had a higher concentration in the small-size mode (100–500 nm) in urban areas, likely caused by the existence of strong POA emissions (Zhang et al., 2014a; Liu et al., 2016). However, in Xinglong, the peak diameters of OA were close to those of SIA in the four seasons, indicating OA was highly oxidized in Xinglong.

The size distributions of SIA showed similar shapes in spring and autumn and peaked at approximately 700 nm, suggesting internal mixing (Liu et al., 2016). The mode diameters of the SIA in Xinglong (700–750 nm) were higher than those in Beijing (600–650 nm) in spring and summer (Hu et al., 2017). The differences in SIA peak diameters may have been caused by the stronger photochemical activity and long-range transport in Xinglong than in Beijing in these two seasons. The peak diameter of sulfate in summer was the highest in the four seasons, indicating that the sulfate was highly aged in summer in the NCP. The peak diameters (550–700 nm) of PM$_1$ species in winter were lower than those in the other three seasons, which might be attributable to the existence of relatively higher primary emissions in winter. Moreover, the greater level of new-particle formation in winter also resulted in smaller average sizes (Hu et al., 2016).

### 3.2 OA source appointment

Owing to the high fraction of OOA in OA, it is extremely difficult to separate POA from OOA in Xinglong using PMF analysis, because the POA factor is easily mixed with the OOA factor. For example, for spring, according to the PMF analysis (Figs. S2–S6), in the 2- to 4-factor solutions, POA factors were mixed with OOA factors because the POA-like profile contained a higher-than-expected contribution from m/z 44. In the 5-factor solution, a POA factor appeared, while OOA was over-split, some of which showed similar characteristics. The mass spectrum pattern of the POA factor (factor 3) mainly consisted of hydrocarbon ions ($C_nH_{2n+1}^+$ and $C_nH_{2n-1}^+$), which are commonly related to combustion emissions (Zhang et al., 2015; Sun et al., 2013b). The mass profile of the POA factor had some similarity with that of HOA and coal combustion OA (CCOA) (Hu et al., 2017; Elser et al., 2016). The correlation coefficient between POA and NOx was 0.58, and that between POA and chloride was 0.78, in spring, suggesting a significant contribution of coal combustion and traffic-related sources to the POA factor in Xinglong. Moreover, HOA and CCOA show remarkably similar mass spectrum patterns when m/z is below

120 (Sun et al., 2016; Sun et al., 2018), which is sometimes difficult to be separated by PMF analysis, so FFOA can be considered as a combined factor of HOA and CCOA (Sun et al., 2018). In this study, it was difficult to separate CCOA form HOA because of the low percentage of POA in OA. Therefore, the POA factor in spring could also be considered as FFOA, which is a typical profile in Xinglong. In autumn, the solution of the PMF analysis was similar with that in spring. A POA factor appeared until the 7-factor solution and OOA was over-split. The POA factor in autumn could also be considered as FFOA. We constrained the FFOA profiles separated by the 5-factor solution of PMF analysis in spring and the 7-factor solution in autumn to better separate FFOA from OOA in spring and autumn, respectively. As a result, three OA factors, including FFOA, LO-OOA and MO-OOA, were identified with ME-2 analysis in spring and autumn. In winter, the 3-factor solution (FFOA, LO-OOA and MO-OOA) from PMF analysis was good enough, so it was not necessary to use ME-2 analysis to separate POA from OOA. In summer, no POA factor appeared in the 2- to 9-factor solutions of the PMF analysis, suggesting the faction of POA in OA in summer was too low to be identified. Therefore, the PMF analysis results of two OA factors (LO-OOA and MO-OOA) were used. The proportions of FFOA to OA were relatively low in the other three seasons (14–23%), mainly because of the low anthropogenic emissions around Xinglong station. The concentrations of FFOA were higher at night than during the daytime during the four seasons, which was mainly due to the variations in the PBL. The lower PBL at night suppressed the diffusion of pollutants. Meanwhile, higher primary emissions, such as coal-burning emissions, at night than during the daytime, also partly contributed.

The high $f_{44}$ values were permanent in the mass spectrum of both LO-OOA and MO-OOA. The $f_{44}$ values for MO-OOA and LO-OOA in the four seasons ranged from 16.3 to 23.5% and 8.1 to 13.8%, respectively. The $f_{43}$ values for MO-OOA and LO-OOA ranged from 4.8 to 5.2% and 6.8 to 9.1%, respectively. These behaviors indicated that MO-OOA had a higher oxidation state than LO-OOA. The O/C ratios of the MO-OOA factors in the four seasons were 0.93, 0.93, 0.84 and 0.83, respectively—higher than those in the corresponding LO-OOA (0.69, 0.58, 0.67 and 0.49). In addition, the correlation coefficient between LO-OOA and nitrate (sulfate) was 0.79 (0.58) in autumn and 0.71 (0.44) in winter. Therefore, LO-OOA correlated well with nitrate in autumn and winter. MO-OOA, meanwhile, correlated well with both nitrate and sulfate in autumn (MO-OOA vs. nitrate: 0.86; MO-OOA vs. sulfate: 0.87) and winter (MO-OOA vs. nitrate: 0.79; MO-OOA vs. sulfate: 0.72), similar to the findings of previous studies in wintertime in Beijing (Hu et al., 2017). In comparison, LO-OOA had a low correlation coefficient with nitrate or sulfate, and MO-OOA had a correlation coefficient of 0.65 with sulfate, in summer. The poor correlation between LO-OOA and secondary inorganic species has also been found in a previous study (Sun et al., 2018).

OOA accounted for as much as 77–100% of the OA in the four seasons (Fig. 4). The percentages of OOA in OA during all seasons in Xinglong were much higher than those in urban Beijing (48–68%; Hu et al., 2017), slightly higher than the results observed at national background stations in Waliguan (75%; Zhang et al., 2019) in western China and at Lake Hongze in northern China (70%; Zhu et al., 2016), and comparable to those observed in a less-polluted atmosphere in Hong Kong (80–

85%; Li et al., 2015) and a rural site in Xingzhou, central China (82%; Wang et al., 2016), but lower than that observed at a national background station at Mount Wuzhi in eastern China (100%; Zhu et al., 2016). These characteristics indicate the occurrence of highly oxidized OA in Xinglong, which may be attributable to the high oxidizing ability and the strong impact of regional transportation in the background atmosphere of the NCP.

### 3.3 Diurnal variations

As shown in Fig. 5, nitrate exhibited drastic diurnal variation in each season, with a high concentration at night and low concentration during the daytime. This behavior was closely related to the variation of the PBL, which reduced the concentration of nitrate during the daytime and suppressed the diffusion of nitrate at night. Heterogeneous/aqueous-phase reactions and gas-to-particle condensation processes are the main pathways to forming fine-mode nitrate (Sun et al., 2018; Hu et al., 2017). The high concentration of nitrate in each season suggested the pathway of the hydrolysis of dinitrogen pentoxide ($N_2O_5$) to nitrate formation at night in Xinglong might be strong due to low NO concentrations and high $O_3$ concentrations, even at night. The NO concentrations at Xinglong Station in the four seasons were as low as 0.2 to 0.7 ppb (Table 1). Because of the low concentration of NO, it would be difficult for NO to react with $O_3$ and thus deplete $O_3$ so that $O_3$ could accumulate, even at night. $O_3$ concentrations at night in the four seasons were about 45, 70, 35 and 25 ppb, respectively, which showed that the background atmosphere exhibited high atmospheric oxidation capacity, even at night, especially in summer. The diurnal variation of nitrate showed an obvious increase from noon through the afternoon in each season, suggesting the increased nitrate concentrations were influenced by photochemical production. Nitrate exhibited its lowest concentration in summer, which can be attributed to the evaporation of $NH_4NO_3$ due to the high temperatures (Fig. 5).

In comparison to the diurnal patterns of nitrate, sulfate showed flatter diurnal cycles in each season, demonstrating the regional characteristics. In summer, sulfate increased rapidly from noon to evening, and the wind speed and $O_3$ concentration increased significantly, indicating a strong influence of the regional transport of sulfate and $O_3$ in summer. Specifically, the wind speed in autumn and winter increased slightly from 08:00 to 14:00 from about 1.8 to 2.8 m s$^{-1}$. However, in summer, the wind speed increased rapidly from 08:00 to 16:00 from 0.7 to 2.8 m s$^{-1}$, and the $O_3$ concentration increased significantly from 60 to 88 ppb, with an increase rate of 3.5 ppb h$^{-1}$ at the same time. As a result, photochemical processing enhanced sulfate formation during the regional transport during the daytime. At night, however, the high sulfate concentration might be attributable to the enhancement of aqueous-phase processing under high temperatures and humidity (Zhang et al., 2014b).

MO-OOA exhibited similar drastic diurnal profiles in spring, autumn and winter, peaking in the afternoon and at night. The high concentration of MO-OOA at night was likely due to the co-effect of the low PBL and aqueous chemistry under high RH conditions at night (Hu et al., 2017; Sun et al., 2018). Although the PBL expanded, the concentrations of MO-OOA increased significantly from 12:00 to 18:00, indicating an important role played by photochemical processes in MO-OOA production during the daytime in these three seasons. In summer, the concentration of MO-OOA showed its greatest increase

rate (0.18 μg·m$^{-3}$·h$^{-1}$) from 09:00 to 18:00, implying stronger photochemical production of MO-OOA than in other seasons. Note that the wind speed in summer also increased rapidly from 09:00 to 16:00, along with the increase of MO-OOA, which may suggest regional transport also partly contributed to the rapid increase in MO-OOA during the daytime in summer. The diurnal profiles of LO-OOA in each season were flatter than those of MO-OOA. The decreased PBL mainly resulted in a higher concentration of LO-OOA at night. The increased concentration of LO-OOA from noon through the afternoon was mainly due to photochemical processes. The highest concentration of LO-OOA in summer suggested a stronger photochemical production of LO-OOA than in other seasons.

The primary components, chlorine, and FFOA, exhibited similar diurnal variations, with lowest concentrations in summer and highest concentrations in winter. Moreover, both chlorine and FFOA showed flat diurnal cycles in summer, but dramatic diurnal cycles in winter with high concentrations at night and low concentrations during the daytime. These characteristics could be attributed to the relatively high emissions in winter and the suppressed PBL at night.

**3.4 SOA chemistry and oxidation state**

**3.4.1 Oxidation state of OA and SOA**

As shown in Table 3, during the four seasons in this study, the average oxygen-to-carbon (O/C) and hydrogen-to-carbon (H/C) ratios, organic-mass-to-organic-carbon (OM/OC) ratios and carbon oxidation state (OSc) were in the ranges of 0.54–0.75, 1.41–1.53, 1.86–2.13 and −0.45–0.10, respectively. The O/C ratio was highest in summer (0.75), second highest in spring (0.71), and lowest in winter (0.54). The high O/C ratios in summer (0.75) were also observed in urban Beijing and suburban areas of Hong Kong, indicating strong OOA formation via photochemical processing in summer. The low O/C ratios in winter (0.54) were related to a higher proportion of POA in OA than in other seasons. The O/C ratios in Xinglong in all seasons (0.54–0.75) were slightly higher than those in urban Beijing (0.47–0.53), mainly due to the weak influence of primary emissions. Meanwhile, the O/C ratios in Xinglong in all seasons were much higher than those obtained at rural/suburban sites in China and in western countries, e.g., Hong Kong (0.38–0.52), Kaiping (0.47), Fresno (0.35), Melpitz (0.52–0.54) and Mexico City (0.53) (Li et al., 2015; Huang et al., 2011; Poulain et al., 2011; Aiken et al., 2009; Ge et al., 2012; Hu et al., 2017), suggesting the OAs in Xinglong were highly aged. In addition, the O/C ratios in Xinglong in the four seasons were comparable with those observed at the background Lake Hongze site in northern China. The slightly higher O/C ratios of OA in national background areas in eastern and western China, such as 0.99 at Waliguan background station in summer and 0.98 at Mount Wuzhi site in spring (Zhang et al., 2019; Zhu et al., 2016), were due to the highly aged air mass during the long-range transport. Overall, the oxidation state of OA in Xinglong was far higher than that at urban/rural/suburban sites and comparable with that at background sites in eastern and western China.

Figure 6 compares the O/C ratios of OOA, LO-OOA and MO-OOA in a variety of environments grouped into two types

according to their location: urban and suburban/rural/background sites. For some sites, only one OOA factor was identified, while for others, two OOA factors, including LO-OOA and MO-OOA, were identified according to the different oxidation states. For the latter sites, the average O/C ratios of OOA can be reconstructed as a mass-weighted average of the O/C ratios of LO-OOA and MO-OOA, according to Ng et al. (2010). The O/C ratios of OOA at Xinglong in the four seasons (0.8–0.93) were comparable with those observed at the background Lake Hongze site (0.89) in northern China, but lower than those observed in national background areas in eastern and western China, which was due to the highly aged air mass during the long-range transport in national background areas. These results highlight the high atmospheric oxidizing capacity in background areas in both southern and northern China. The O/C ratios of OOA in Xinglong were far higher than those obtained at urban sites in lightly polluted areas, e.g., Hong Kong and Fresno, and those in suburban/rural/downwind areas, e.g., Jiaxing, Kaiping and Hong Kong, and comparable to those in urban Beijing in the NCP in the four seasons (Li et al., 2015; Huang et al., 2011; Ge et al., 2012; Huang et al., 2013; Hu et al., 2017; Xu et al., 2014). The comparable O/C ratios of the OOA in Xinglong and Beijing (Hu et al., 2017) in winter were probably due to the relatively higher proportion of POA in the OA than in the other seasons. Although the O/C ratios of OOA in Xinglong and Beijing were comparable and both at high levels, the formation mechanisms of OOA were distinct at the two sites, and a detailed discussion is provided in Sect. 3.4.2.

### 3.4.2 Evolution and formation of SOA

Figures 7 and 8 show the mass concentrations and fractions of LO-OOA and MO-OOA as functions of RH and odd oxygen (Ox = $O_3$ + $NO_2$) in all seasons, respectively. Both LO-OOA and MO-OOA increased significantly as RH increased when RH was < ~90% in autumn and winter. This behavior suggested aqueous-phase processing had a significant influence on OOA formation in these two seasons. MO-OOA increased more rapidly than those of LO-OOA as RH increased. As a result, the mass fractions of MO-OOA increased by 25% in autumn and by 12% in winter when RH increased from 20 to 90%. Corresponding, the mass fraction of LO-OOA decreased by 15% in autumn. These characteristics indicated that aqueous-phase processing plays more important role in MO-OOA formation than that in LO-OOA in these two seasons. Note that the mass fraction of MO-OOA did not increase as Ox elevated in winter, while it increased ~from 30% to 40% as RH increased from 30 to 90%. This characteristic suggested a more dominant important role of aqueous-phase processing on SOA formation than photochemical processing in this season. In addition, in autumn, the increases of LO-OOA and MO-OOA under high RH conditions (70% < RH < 90%) were associated with a significant increase in average wind speed from 1.8 to 2.9 m s$^{-1}$ (Fig. 7k), which facilitated the transport of water vapor and pollutants from the heavily polluted southern regions.

In spring, LO-OOA and MO-OOA only increased under moderate RH (RH < 70%) as RH increased. Notably, Ox also increased when RH was < 70% as RH increased. LO-OOA and MO-OOA increased rapidly at moderate Ox levels when Ox changed from 50 to 70 ppb, and then remained unchanged at high Ox levels. These observations suggested both photochemical and aqueous-phase processing contributed to LO-OOA and MO-OOA production in spring. RH maintained at low levels (RH

< 40%) as Ox increased, suggesting a weaker impact of aqueous-phase processing on OOA production in spring than in autumn and winter.

In summer, both LO-OOA and MO-OOA showed overall increasing trends as Ox increased, while RH showed a corresponding overall decreasing trend. This behavior indicates a strong influence of photochemical processing on both LO-OOA and MO-OOA production. Meanwhile, LO-OOA showed a continuously decreasing trend as RH increased in summer, except for a slightly increasing trend when RH increased from 40 to 60%, indicating photochemical processing dominated LO-OOA formation. MO-OOA increased significantly with Ox, while it increased slightly with RH (40% < RH < 60%) firstly, and then decreased with RH when RH was above 60%. This characteristic suggested photochemical processing dominated MO-OOA formation, but the role of aqueous-phase processing under moderate RH (40% < RH < 60%) conditions cannot be ruled out in summer. In urban Beijing, the impact of photochemical processing on LO-OOA production was significant, while on MO-OOA production it was limited in summer (Xu et al., 2017; Duan et al., 2020), mainly due to the higher atmospheric oxidation capability in the background atmosphere than in the urban atmosphere in summer. Furthermore, increases of LO-OOA and MO-OOA as functions of Ox were clearly associated with the increases of wind speed, which was more significant in summer than in other seasons. In comparison, in urban areas, such as Beijing, increase of LO-OOA were associated with the decreases of wind speed, that facilitated the accumulation of air pollutants (Xu et al., 2017). Such a difference between urban and background areas may be due to the influence of regional transport on the Ox and SOA concentrations in background areas. The continuous increases in SOA concentrations associated with the increases of wind speed in the background atmosphere may imply important role of regional transport in SOA formation in summer. Different variations in the O/C ratios of OA were observed in the four seasons. Clear increases in the O/C ratios of OA as RH increased were observed in all seasons except summer, suggesting an impact of aqueous-phase processing on the oxidation state of OA in spring, autumn and winter.

**3.5 Transportation pathways**

To explore the transportation pathways and the effects of regional and long-distance transport on fine particles at the background site in the NCP, we calculated the backward trajectories of $PM_1$ species with TrajStat and the HYSPLIT-4.8 model in four seasons. Both long-distance transport and regional air masses influenced Xinglong (Fig. 9). Based on the distances over which the air masses were transported, the clusters during the four seasons were defined as short, medium, and long transportation pathways. Specifically, clusters 2 in spring, 2 in summer, 4 in autumn, and 2 in winter were defined as short transportation pathways. Clusters 3 in spring, clusters 1, 3, 4 and 5 in summer, clusters 1 and 3 in autumn, and cluster 3 in winter were considered as medium transportation pathways. Cluster 2 in spring, cluster 5 in summer, cluster 2 in autumn, and cluster 1 in winter were considered as long transportation pathways.

The short-distance trajectories originated in the southwest/southeast of Xinglong—areas that suffer from serious pollution. The southwest trajectories started in southern Hebei Province and passed over Beijing. The southeast trajectories started at the

Bohai Sea and extended through Tianjin and Tangshan. Although the three short-distance clusters accounted for only 27–43% of all the air masses during each season, the $PM_1$ concentrations for the short-distance clusters from the southern regions were the highest of all the clusters, indicating that aerosol particles at Xinglong station were greatly affected by the regional transport from southern regions. The long-distance trajectories mainly came from the farther northwest regions of Xinglong. The long-distance clusters accounted for 26–49% of all the clusters in spring, autumn and winter. The long-distance clusters brought less-polluted aerosols from the northern regions. These results were also supported by the low $PM_1$ concentrations of 3.6–5.4 μg m$^{-3}$ associated with the long-distance trajectories.

During the seasonal observations in Xinglong, the pathways of the dominant air masses differed. The medium-distance clusters (clusters 1, 3, 4 and 5) were dominant in summer and are representative of a regional-scale transportation pathway. The air masses from the southern regions (clusters 2 and 3) accounted for 56% of all the air masses in summer, which was obviously higher than the percentage in other seasons (27–38%). Cluster 3 in summer started at Bohai Bay and passed through the Shandong Peninsula and over Bohai Bay. The $PM_1$ concentrations for clusters 2 (14.7 μg m$^{-3}$) and 3 (12.2 μg m$^{-3}$) were both high. These results suggest a dominant role played by southern transport in submicron aerosol concentrations over the NCP in summer. Furthermore, the transport distances of clusters from the north and west regions in summer were shorter than those in other seasons. In general, with a decrease in the transport distance of clusters from the north and west regions, particle concentrations gradually increase (Hu et al., 2017). Although the clusters from these regions in summer only accounted for 15% and 8% of all the air masses, respectively, the $PM_1$ concentrations for the two clusters (cluster 1: 12.8 μg m$^{-3}$; cluster 5: 10.2 μg m$^{-3}$) were both at high levels and similar to those associated with the southern air masses (cluster 2: 14.7 μg m$^{-3}$ and cluster 3: 12.2 μg m$^{-3}$). All these characteristics suggest that regional transport from Inner Mongolia (west and north regions of Xinglong) also partially contributes to the particle pollution in the background area of the NCP in summer.

## 4. Conclusion

The chemical components in $PM_1$ were investigated during all four seasons at a background station in the NCP using a HR-ToF-AMS measurement. The average mass concentrations of NR-$PM_1$ in the four seasons of spring, summer, autumn and winter were 13.7 ± 16.0, 12.4 ± 7.4, 15.1 ± 18.7 and 14.1 ± 13.3 μg m$^{-3}$, respectively. OA contributed the most to $PM_1$ in summer, accounting for 40% by mass. Nitrate was the greatest SIA component in spring (34%), winter (31%) and autumn (34%), while sulfate was the highest SIA component in summer (38%). $PM_1$ in Xinglong showed moderate acidity in spring, autumn and winter, with average pH values of 4.2 ± 0.7, 3.5 ± 0.5 and 3.7 ± 0.6, respectively, which were higher than those in the U.S. and Europe. Excessive $NH_3$ and high $NO_3/SO_4$ ratios contributed to the relatively lower aerosol acidity over the NCP in all seasons except in summer (pH: 2.7 ± 0.6) on a regional scale. The size distribution of all $PM_1$ species showed a consistent accumulation mode peaked at approximately 600–800 nm (dva), indicating a highly aged and internally mixed nature of the

background aerosols.

The PMF and ME-2 methods were used to analyze the mass spectrum in Xinglong in the four seasons and identified three OA factors, including FFOA, LO-OOA, and MO-OOA. SOA (LO-OOA + MO-OOA) dominated OA as much as 77–100% in the four seasons, especially in summer (100%). The oxidation state and the process of evolution of OAs in the four seasons were further investigated, and an enhanced carbon oxidation state (−0.45–0.10) as well O/C (0.54–0.75) and OM/OC (1.86–2.13) ratios, compared with urban studies, were observed, with the highest oxidation state appearing in summer, likely due to the relatively stronger photochemical processing that dominated the processes of both LO-OOA and MO-OOA formation. Aqueous-phase processing also contributed to the SOA formation and prevailed in winter, the role of which was more important in MO-OOA than in LO-OOA. In addition, regional transport also played an important role in the variations of SOA, especially in summer that continuous increases in SOA concentration as a function of Ox was observed to be associated with the increases of wind speed. The backward trajectory analysis showed that higher $PM_1$ concentrations were from the southern regions of Xinglong, with shorter transportation distances in all four seasons; whereas in summer, regional transport from Inner Mongolia (west and north regions of Xinglong) also partially contributed to the pollution over the NCP because of the similar $PM_1$ concentrations to air masses from Inner Mongolia and southern regions of Xinglong.

Our results illustrate that the background particles over the NCP are influenced significantly by aging processes and regional transportation, which is similar to the situation for background aerosols over southern and western China. In addition, the increased contribution of aerosol nitrate in background NCP, which contrast with sulfate-dominated submicron particles in southern and western China, highlights how regional reductions in emissions of nitrogen oxide are critical for remedying the occurrence of haze events over the NCP.

*Data availability.* The datasets can be accessed upon request to the corresponding authors.

*Author contributions.* LJ performed the research, designed the analysis approach, and wrote the paper. LJ, LZ and HL had the original idea. LZ and HL provided writing guidance and revised the paper. LJ and CL calibrated the HR-ToF-AMS and performed data evaluation. CL provided ME-2 analysis guidance and the HR-ToF-AMS instrument. LJ and GW operated and maintained the HR-ToF-AMS. All co-authors proofread and commented the manuscript.

*Competing interests.* The authors declare that they have no conflict of interests.

*Acknowledgements.* This study was supported by the Ministry of Science and Technology of China (Grant no. 2017YFC0210000), the National Natural Science Foundation of China (Grant no. 41705110), the Beijing Municipal Natural Science Foundation (8192045), and Beijing Major Science and Technology Project (Z181100005418014). The authors are

grateful for the support of the staff at Xinglong station.

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

**Table 1.** Seasonal variations in average meteorological parameters, gaseous precursors, and NR-PM$_1$ species. The data here are listed in the form of "average ± standard deviation".

|  | Spring | Summer | Autumn | Winter |
|---|---|---|---|---|
|  | 1 May-31 Mar | 22 Jun-27 Jul | 11 Oct-11 Nov | 25 Nov-26 Dec |
| **Meteorological parameters** |  |  |  |  |
| T (℃) | 3.9 ± 4.7 | 21.4 ± 3.9 | 8.1 ± 3.9 | -5.1 ± 3.1 |
| RH (%) | 31 ± 17 | 68 ± 19 | 49 ± 23 | 44 ± 20 |
| WS (m s$^{-1}$) | 3.3 ± 1.3 | 2.0 ± 1.3 | 2.1 ± 1.3 | 2.4 ± 1.4 |
| P (Hpa) | 913 ± 4 | 906 ± 3 | 918 ± 5 | 920 ± 5 |
| **Gaseous precursors** |  |  |  |  |
| O$_3$ (ppb) | 48 ± 11 | 78 ± 24 | 37 ± 13 | 29 ± 8 |
| NO (ppb) | 0.2 ± 0.2 | 0.2 ± 0.3 | 0.4 ± 0.8 | 0.7 ± 1.4 |
| NO$_2$ (ppb) | 4.0 ± 4.0 | 4.0 ± 2.0 | 12 ± 7 | 14 ± 10 |
| NOx (ppb) | 4.3 ± 4.1 | 4.3 ± 2.2 | 13 ± 9 | 15 ± 11 |
| SO$_2$ (ppb) | 1.0 ± 1.2 | 1.9 ± 1.7 | 1.3 ± 1.7 | 0.7 ± 1.2 |
| CO (ppm) | 0.55 ± 0.41 | 0.49 ± 0.25 | 0.54 ± 0.22 | 0.59 ± 0.33 |
| **Aerosol species /µg m$^{-3}$** |  |  |  |  |
| Org | 4.5 ± 4.7 | 4.9 ± 2.6 | 4.5 ± 3.6 | 4.8 ± 3.8 |
| FFOA | 0.7 ± 0.7 |  | 0.6 ± 0.5 | 1.1 ± 0.9 |
| LO-OOA | 1.2 ± 1.7 | 1.9 ± 1.1 | 1.3 ± 0.7 | 1.2 ± 1.2 |
| MO-OOA | 2.8 ± 2.8 | 3.0 ± 1.6 | 2.7 ± 2.6 | 2.3 ± 2.0 |
| SO$_4^{2-}$ | 2.2 ± 2.6 | 4.6 ± 3.1 | 3.1 ± 4.0 | 2.7 ± 2.8 |
| NO$_3^-$ | 4.7 ± 6.3 | 1.2 ± 14 | 5.1 ± 7.6 | 4.4 ± 5.2 |
| NH$_4^+$ | 2.1 ± 2.6 | 1.6 ± 1.2 | 2.3 ± 3.0 | 2.0 ± 2.0 |
| Cl$^-$ | 0.1 ± 0.1 | 0.01 ± 0.02 | 0.04 ± 0.06 | 0.2 ± 0.2 |
| NR-PM$_1$ | 13.7 ± 16.0 | 12.4 ± 7.4 | 15.1 ± 18.7 | 14.1 ± 13.3 |

**Table 2**. Mass concentrations of nitrate, sulfate, ammonium, LWC, $H_{air}^+$ and pH values in China and other countries.

| Site | Period | SO$_4^{2-}$ | NO$_3^-$ | NH$_4^+$ | LWC | $H_{air}^+$ | pH | Reference |
|---|---|---|---|---|---|---|---|---|
|  |  | (µg m$^{-3}$) | (µg m$^{-3}$) | (µg m$^{-3}$) | (µg m$^{-3}$) | (ng m$^{-3}$) |  |  |

| Site | Period | | | | | | | Reference |
|---|---|---|---|---|---|---|---|---|
| Xinglong, China | Spring | 2.2 ± 2.6 | 4.7 ± 6.3 | 2.1 ± 2.6 | 6 ± 13 | 0.5 ± 0.6 | 4.2 ± 0.7 | This study |
| | Summer | 4.6 ± 3.1 | 1.2 ± 1.4 | 1.6 ± 1.2 | 14 ± 24 | 42 ± 68 | 2.7 ± 0.6 | |
| | Autumn | 3.1 ± 4.0 | 5.1 ± 7.6 | 2.3 ± 3.0 | 18 ± 38 | 3.7 ± 6.9 | 3.5 ± 0.5 | |
| | Winter | 2.7 ± 2.8 | 4.4 ± 5.2 | 2.0 ± 2.0 | 12 ± 26 | 1.2 ± 2.4 | 3.7 ± 0.6 | |
| Beijing, China | Spring | 8.4 ± 7.7 | 12.6 ± 14.2 | 6.7 ± 7.2 | 21 ± 33 | 3.7 ± 15 | 4.4 ± 1.2 | Ding et al. (2019) |
| | Summer | 8.6 ± 7.5 | 9.5 ± 9.5 | 7.2 ± 5.6 | 50 ± 68 | 16 ± 18 | 3.8 ± 1.2 | |
| | Autumn | 6.5 ± 5.9 | 18.5 ± 19.5 | 8.2 ± 8.2 | 109 ± 160 | 8.1 ± 11 | 4.3 ± 0.8 | |
| Beijing, China | Winter | | | | 220 (28–711) | 11.7 | 4.2 (3.0–4.9) | Liu et al. (2017) |
| Zhengzhou, China | Winter | | | | 95 (10–250) | 2.0 | 4.5 (3.8–5.2) | Wang et al. (2020) |
| Anyang, China | Winter | | | | 70 (5–190) | 1.0 | 4.8 (3.9–5.9) | |
| Yorkville, US | Autumn | 1.6 ± 0.4 | 0.2 ± 0.1 | 0.4 ± 0.2 | 1.6 ± 1.7 | 10.0 | 2.2 ± 0.6 | Nah et al. (2018) |
| Alabama, US | Summer | 1.7 ± 1.2 | 0.08 ± 0.08 | 0.5 ± 0.3 | 5.1 ± 3.8 | 1.5 | 0.9 ± 0.6 | Guo et al. (2015) |
| Crete, Greece | Autumn | 2.3 ± 1.6 | 0.8 ± 0.6 | 0.5 ± 0.2 | 2.2 ± 1.7 | 2-4.5 | 1.2 ± 1.1 | Bougiatioti et al. (2016) |

**Table 3.** Elemental ratios and OSc (carbon oxidation state) in OA obtained from field observations at background, rural/suburban, and urban sites. (O/C: oxygen-to-carbon ratio; H/C: hydrogen-to-carbon ratio, OM/OC: organic-mass-to-organic-carbon ratio)

| Site | Site type | Period | O/C | H/C | OM/OC | Osc | Reference |
|---|---|---|---|---|---|---|---|
| Xinglong (China) | Background | Spring | 0.71 | 1.44 | 2.08 | −0.01 | This study |
| | | Summer | 0.75 | 1.41 | 2.13 | 0.10 | |
| | | Autumn | 0.61 | 1.47 | 1.94 | −0.26 | |
| | | Winter | 0.54 | 1.53 | 1.86 | −0.45 | |
| Lake Hongze (China) | | Spring | 0.67 | 1.52 | | −0.18 | Zhu et al. (2016) |
| Mount Wuzhi (China) | | Spring | 0.98 | 1.32 | | 0.64 | Zhu et al. (2016) |
| Waliguan (China) | | Summer | 0.99 | 1.43 | 2.44 | 0.55 | Zhang et al. (2019) |
| Hongkong (China) | Rural/Suburban | Spring | 0.38 | 1.35 | 1.64 | −0.59 | Li et al. (2015) |
| | | Summer | 0.52 | 1.36 | 1.84 | −0.32 | |
| | | Autumn | 0.42 | 1.39 | 1.71 | −0.55 | |
| | | Winter | 0.43 | 1.4 | 1.71 | −0.54 | |
| Kaiping (China) | | Autumn | 0.47 | 1.48 | 1.77 | −0.54 | Huang et al. (2011) |
| Melpitz (Germany) | | Summer | 0.52 | 1.51 | 1.83 | −0.47 | Poulain et al. (2011) |

| | | Autumn | 0.54 | 1.48 | 1.84 | −0.4 | |
|---|---|---|---|---|---|---|---|
| | | Winter | 0.53 | 1.48 | 1.83 | −0.41 | |
| Beijing (China) | Urban | Spring | 0.49 | 1.63 | 1.81 | −0.64 | Hu et al. (2017) |
| | | Summer | 0.53 | 1.61 | 1.88 | −0.54 | |
| | | Autumn | 0.46 | 1.58 | 1.77 | −0.66 | |
| | | Winter | 0.47 | 1.52 | 1.79 | −0.58 | |
| Mexico City (Mexico) | | Spring | 0.53 | 1.82 | 1.73 | −0.77 | Aiken et al. (2009) |
| Fresno, CA (U.S.) | | Winter | 0.35 | 1.75 | 1.63 | −1.05 | Ge et al. (2012) |

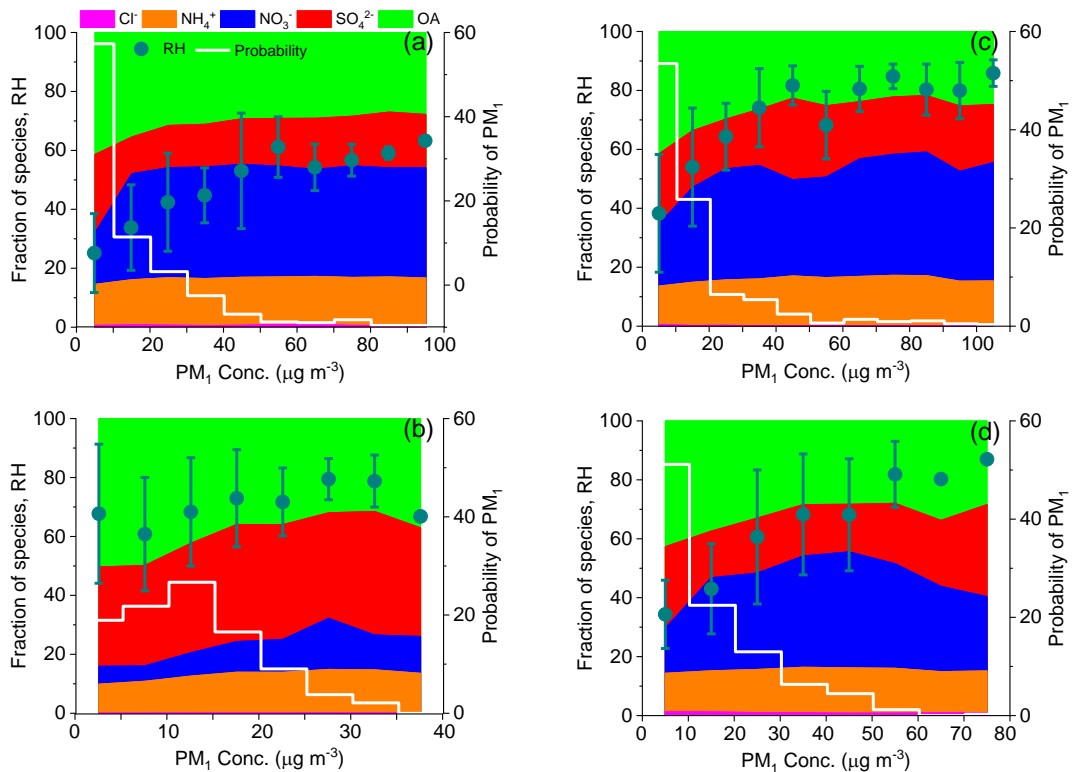

**Figure 1**. Fractions of PM$_1$ species in PM$_1$ as a function of PM$_1$ concentration, and the probability density of PM$_1$ (white

curves), in (a) spring, (b) summer, (c) autumn, and (d) winter. The average values and standard deviations of RH are illustrated.

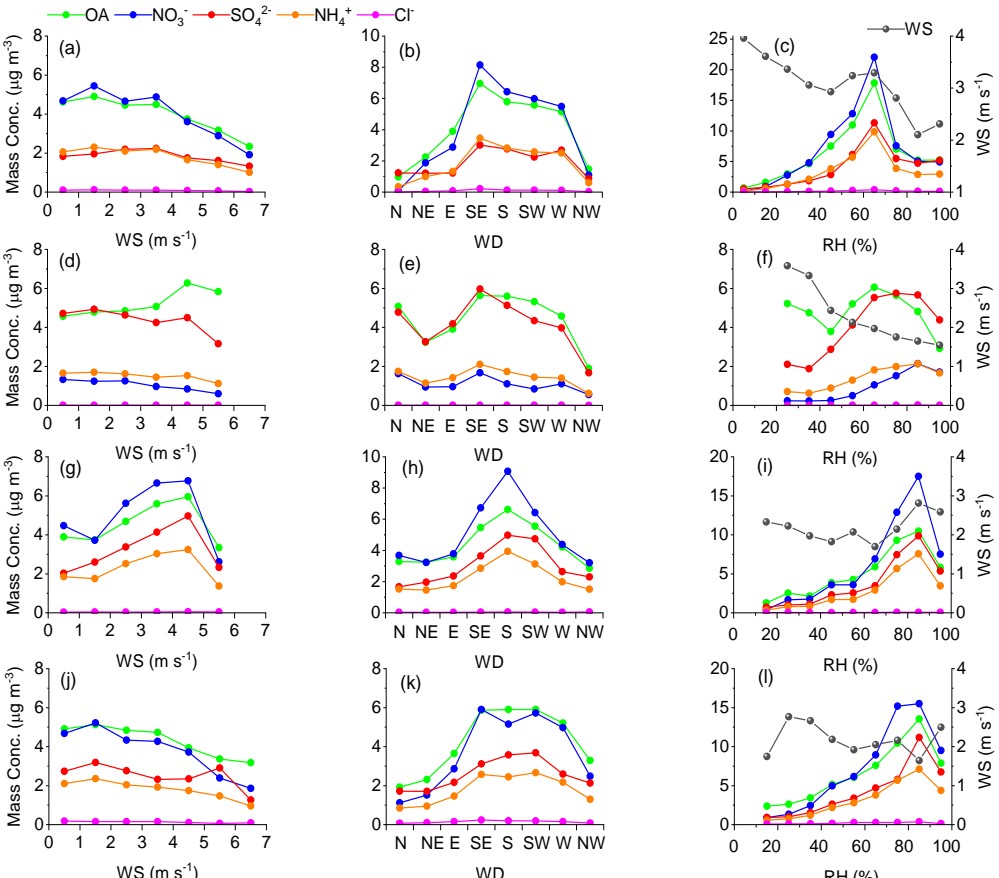

**Figure 2.** Variations in mass concentrations of PM$_1$ species as functions of WS (wind speed), WD (wind direction), and RH in (a) spring, (b) summer, (c) autumn, and (d) winter. (OA: organic aerosol).

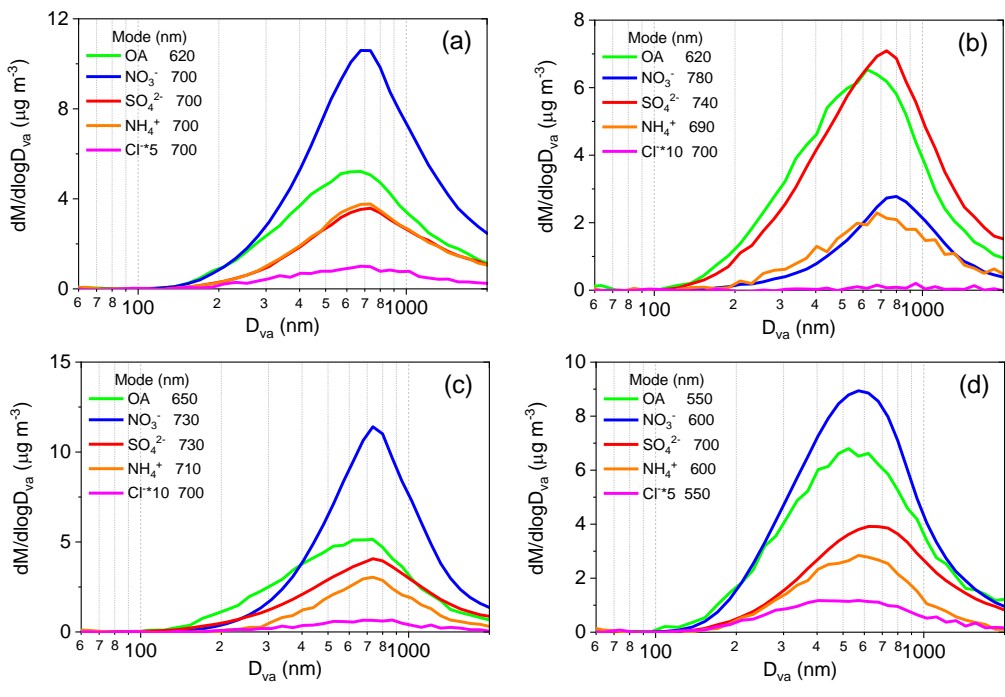

**Figure 3.** Mass size distributions of PM$_1$ species in (a) spring, (b) summer, (c) autumn, and (d) winter.

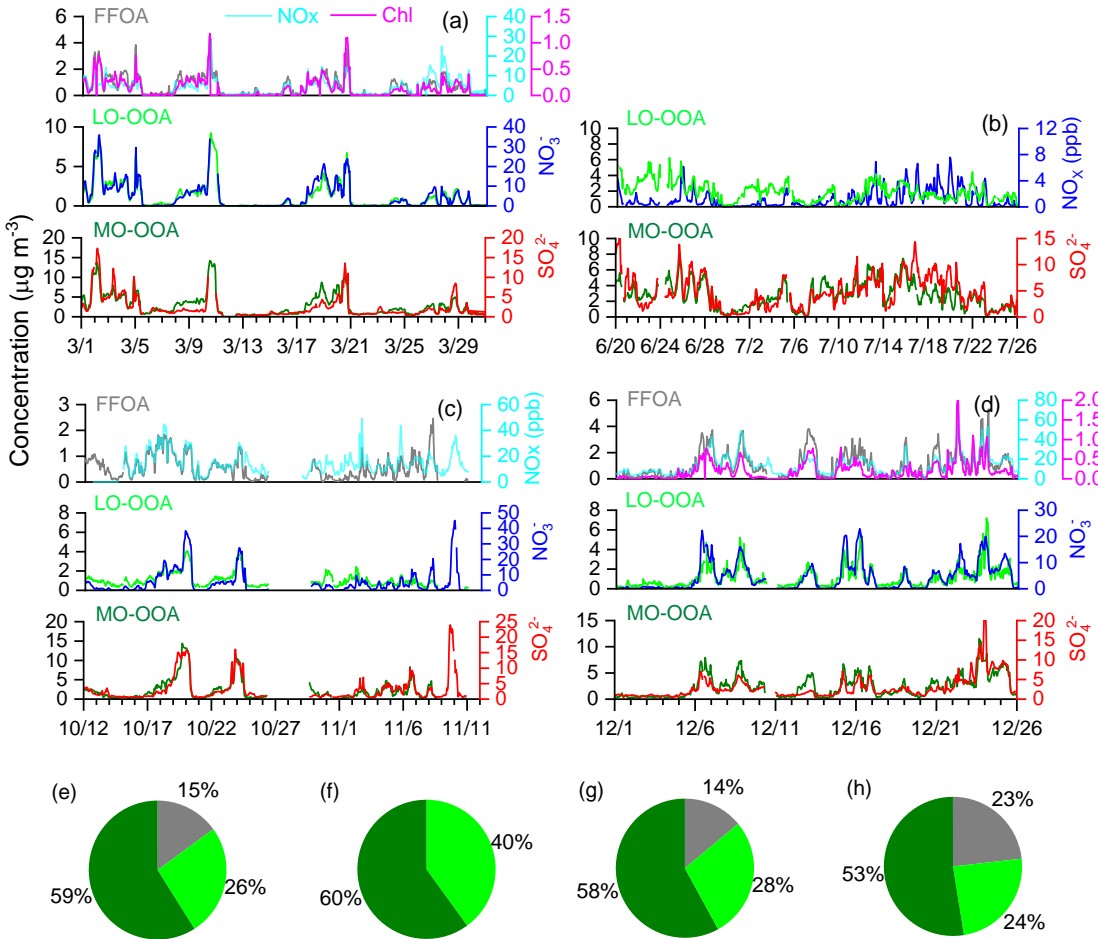

**Figure 4.** Time series of three OA factors in (a) spring, (b) summer, (c) autumn, and (d) winter: FFOA, LO-OOA, and MO-OOA. The time series of NOx, chloride, nitrate, and sulfate are shown for comparison. The pie charts depict the average OA compositions in (e) spring, (f) summer, (g) autumn, and (h) winter.

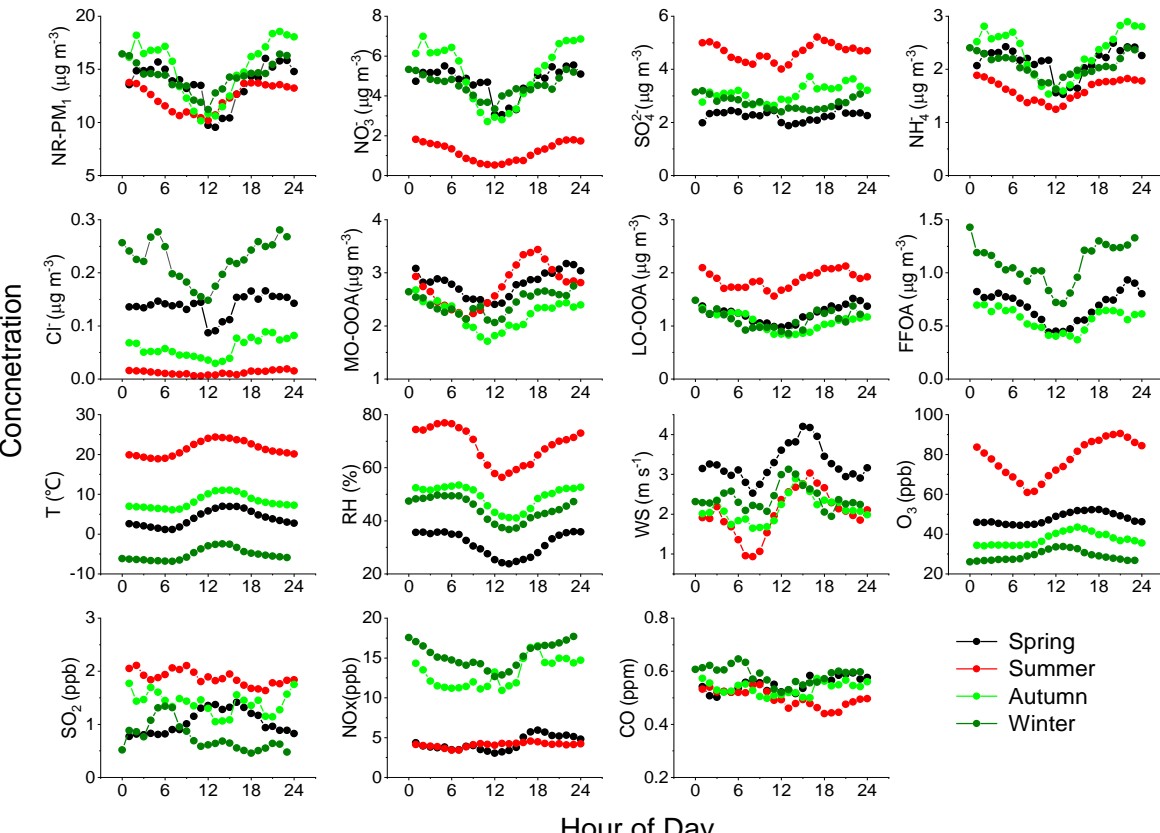

**Figure 5.** Diurnal variations of meteorological parameters, gaseous precursors, and PM$_1$ species in the four seasons.

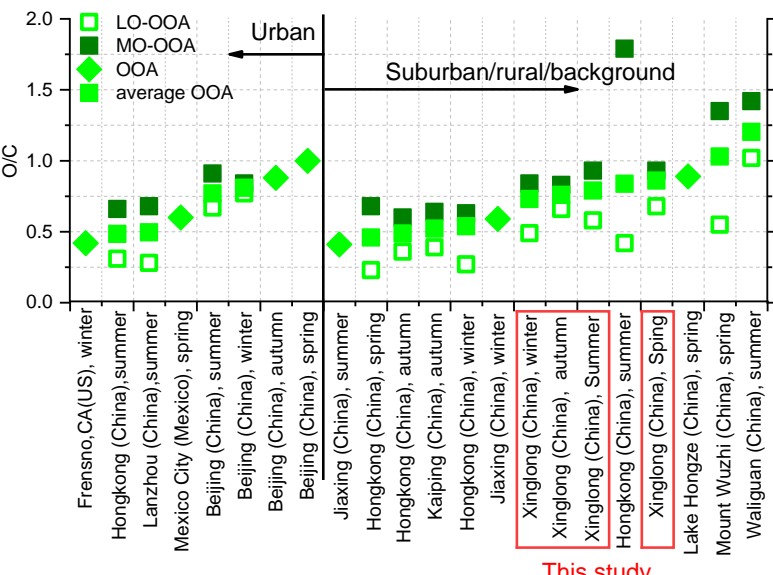

**Figure 6.** Estimated O/C ratios of OOA, LO-OOA and MO-OOA at each site. The mass-weighted average OOA component

is also shown for sites in which LO-OOA and MO-OOA are both resolved. (Note: MO-OOA and LO-OOA are identified as

LV-OOA and SV-OOA, respectively).

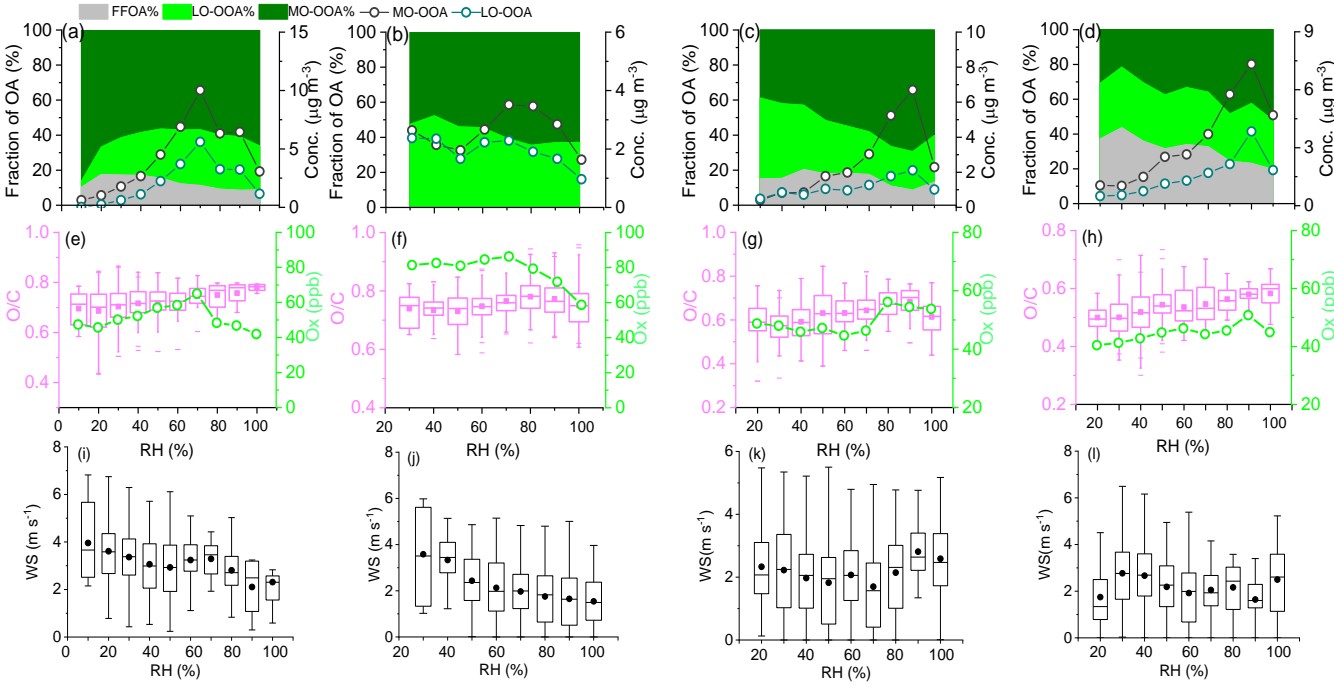

**Figure 7.** Variations in the mass concentrations of FFOA, LO-OOA and MO-OOA and the fractions of OA factors in OA, O/C,

Ox (odd oxygen, $O_3 + NO_2$), and WS (wind speed) as a function of RH in (a, e, i) spring, (b, f, j) summer, (c, g, k) autumn,

and (d, h, l) winter. The data were binned according to the RH (10% increments).

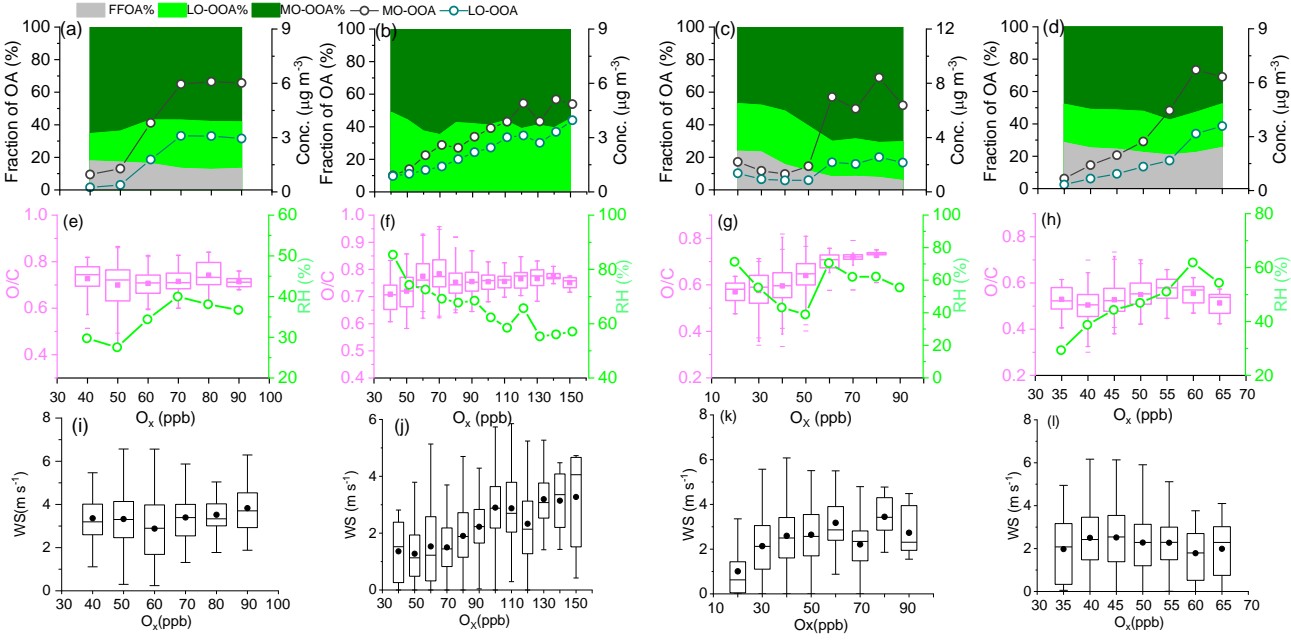

**Figure 8.** Variations in the mass concentrations of FFOA, LO-OOA and MO-OOA and the fractions of OA factors in OA,

RH, and WS (wind speed) as a function of Ox (odd oxygen, $O_3 + NO_2$) in (a, e, i) spring, (b, f, j) summer, (c, g, k) autumn, and (d, h, l) winter. The data were binned according to the Ox (10 ppb increments in spring, summer, and autumn; 5ppb increments in winter).

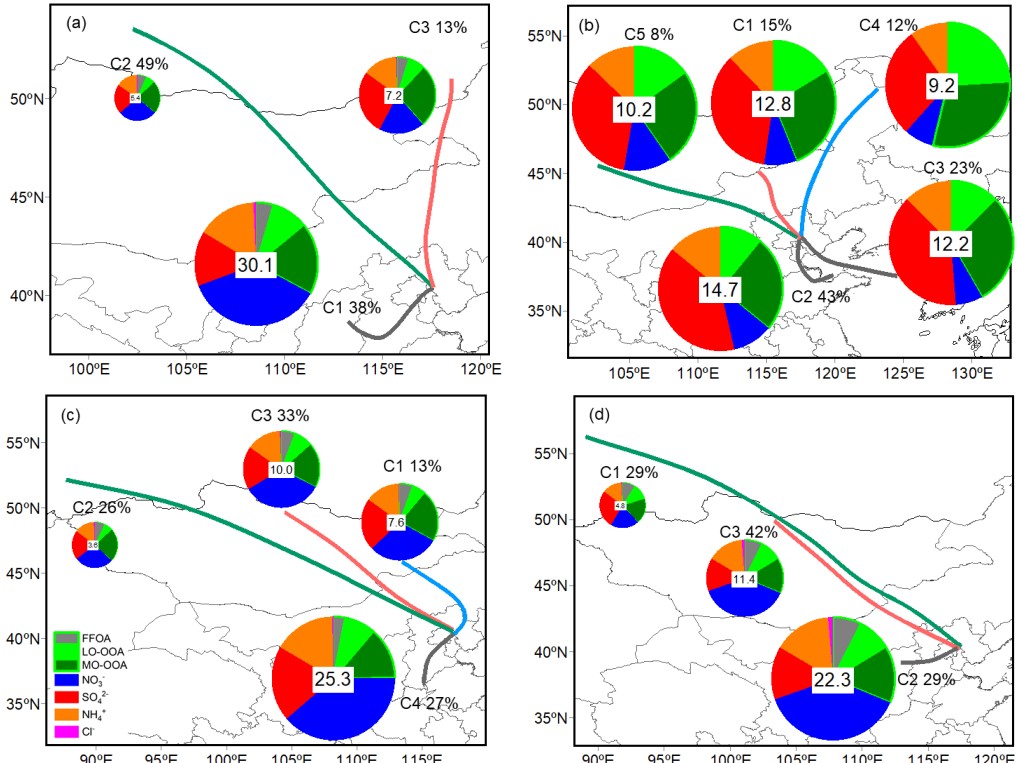

**Figure 9.** Average 48-h backward airmass trajectory clusters calculated at 1-h intervals in (a) spring, (b) summer, (c) autumn, and (d) winter. The percentages of $PM_1$ species in $PM_1$ in each airmass trajectory cluster are shown in the pie charts with the average $PM_1$ concentrations marked in the center of the pie charts.