# Peer review of "Seasonal variations of the highly time-resolved aerosol composition, sources, and chemical processes of background submicron particles in North China Plain"

_Atmospheric Chemistry and Physics, 2020_

## Referee Comment (RC1) · Anonymous Referee #1 · 26 May 2020

This study was conducted at a regional background site of NCP for four seasons using a high-resolution aerosol mass spectrometer. Highly time-resolved chemistry and sources of submicron aerosols were investigated. The authors found that nitrate was the most abundant inorganic species and submicron particles were almost neutralized by excess ammonium in all seasons except summer. Source apportionment of organic aerosol (OA) identified two oxidized OAs, more-oxidized oxygenated OA (MO-OOA) and less-oxidized oxygenated OA (LO-OOA) in all seasons. Significant contributions of aged secondary organic aerosol in OA were observed in all seasons, especially in

summer. The oxidation degree and evolution process of OAs in the four seasons and the comparison with urban studies were further investigated. Overall, the long-term dataset provided in this study is a valuable addition to the literature. I recommend publication after the following issues are addressed. Specific comments: 1. In the introduction, the importance of background site should be emphasized, which is important for highlighting the significance of this paper. 2. Two OOA factors resolved in this study are less-oxidized OOA (LO-OOA) and more-oxidized OOA (MO-OOA), rather than semi-volatile OOA (SV-OOA) and low-volatile OOA (LV-OOA). Please provide more info about the definition for LO-OOA and MO-OOA? 3. The names of submicron aerosol species in the figures should be consistent, e.g., "Cl-" vs Chl, "NO3" vs. "NO3-". 4. Line 77-80: please provide more discussion and reference(s) about the higher atmospheric oxidizing capacity in the background site. 5. Line 153: Please add the standard deviation of the average PM1 concentrations of all seasons here. 6. Line 154: The full name is needed when abbreviation is shown at the first place, such as "SIA" here, 7. Line 337-339: "These characteristics were similar to the results found in previous researches conducted in urban Beijing showing that...... please add references. 8. Line 328: "in the each season" should be "in each season". Line 309: "high oxidized ability" should be "high atmospheric oxidizing capacity". Please go through the manuscript for similar typos.

---

## Referee Comment (RC2) · Anonymous Referee #2 · 18 Jun 2020

This paper reports the seasonal variations of submicron particles and its chemical composition at a background station using HR-ToF-AMS. Using ME-2 analysis, the authors identified different sources of organic aerosol and explored the oxidation degree and evolution process of different OA in four seasons. Backward trajectory analysis was conducted to see the influence of air mass transport. The seasonal dataset of HR-ToF-AMS measurements at a background station in north China is valuable and suitable for a measurement report. But to publish as a scientific article in ACP, this reviewer did not find good novelty or significance of this study compared to previous findings.

[Figure]

Also, many conclusions drawn in this study are not well supported by the observations or interpretations. Overall, this reviewer do not think the scientific significance of the paper meet the scope of ACP scientific articles and cannot recommend the publication of the paper.

Major concerns:

1. With similar data analysis and similar results from this study compared to so many published AMS papers, it is difficult to see the novelty and significance of this paper. Regarding the background atmosphere in NCP, there have also been many studies focusing on air quality and particle chemical composition, including AMS studies. The authors should make it clear what is the specific values of this study. It should not be because you did measurements in a different location, or your measurement period is longer. Instead, the authors should state clearly the scientific questions or valuable findings from these measurements that can improve current understanding of aerosol chemistry.

2. The reviewer cannot be convinced by the PMF analysis and result evaluation in this study, and details are lacking. (1) Regarding the first PMF run with PET, the authors showed in Fig. S2 that OOA was over split in a 5-factor solution without only showing the similar mass profiles. But how about the time series, diurnal variations and O/C ratios of different OOA factors? The authors should not conclude this by only checking the mass profiles. Are they representative of other OOA rather than LO-OOA and MO-OOA? For example, as the authors have been emphasizing the significance of aqueous-phase processing, is there any single OOA factor related to aqueous phase chemistry? Please provide these details either in the manuscript or SI. (2) To perform ME-2 analysis, the authors constrain the FFOA profiles with the POA factor resolved in the five-factor solution of PMF analysis in spring and apply it to all seasons. One concern here is that how do the authors believe the POA factor from the five-factor solution is good enough to represent the primary sources? How does the profile look like when performing PMF analysis to six or seven factor? Does the POA factor in this

study comparable to those resolved from AMS studies previously, especially those in NCP area? Another big concern is that how robust it is to apply this factor from spring to all the other seasons. To check this, the author should perform PET PMF analysis for the other season and see if a POA factor will appear when go to more factors, and then check if the mass profile of POA are comparable in all seasons. As shown in Fig. 5, the mass contribution of FFOA in total OA is 5%, which is in the uncertainty range of PMF analysis. The authors should prove if it is reasonable to manually constrain the ME-2 to separate a single FFOA factor in summer. (3) Details about the evaluation of PMF and ME-2 results are lacking. To interpret PMF/ME-2 results, the authors should carefully follow the procedures proposed by Ulbrich et al. (2009) and Zhang et al. (2011). For example, evaluation of Q/Qexp as a function of factor number and Q/Qexp or scaled residual as a function of m/z should be provided in the manuscript or SI.

3. It is questionable that the authors solely based on the ratio of measured NH4+ to predicted NH4+ to evaluate aerosol acidity. While this ratio has been used as an indicator of aerosol acidity by some previous AMS studies, it has been proved recently that this ion balance method is not reliable to evaluate particle acidity (Guo et al., 2015; Song et al., 2018). Thermodynamic models, i.e., E-AIM and ISORROPIA, should be used. With thermodynamic models, Guo et al. (2017) found that aerosol is always acidic in NPC region, which is contradictory to results from this study.

4. Many interpretations or conclusions from this study cannot be well supported by the observations. For example, the authors conclude in the end of the abstract that the neutralized state of submicron particles highlight the significance of NOx and ammonia reduction (also in conclusion part, Line 486 and Line 510). But these two do not have causal relationship. Line 215, the authors should not make a conclusion that the high NOx concentration is solely due to strong influence of traffic. Line 280, the authors conclude that POA factor was closely related to traffic emission without evaluating the characteristic of POA. While POA showed a good correlation with NOx, does its diurnal profile show morning and evening peaks, which is a typical feature of traffic-related

pollutant? It is not correct to conclude that the high concentration of FFOA at night indicate the high primary emissions. The variations in boundary layer height play a significant role. Similar issues happened again in Sect. 3.3. The authors draw conclusions regarding species formation mechanisms during daytime and nighttime based on the variations in their concentrations without considering the major influence of boundary layer height. Line 323, why the authors think the increase in nitrate concentration is caused by regional transport? The observations cannot support this. In the end of this paragrath, the authors emphasize the strong effects of local chemical production and regional transport on nitrate diurnal pattern. But the reviewer did not find any related discussion about the relative contribution of local production and regional transport. Again, Line 336, why the increased concentration from noon to evening indicate the regional characteristics of MO-OOA? Overall, these conclusions without detailed interpretation and well supported observations would confuse the audience a lot. Line 426, from the observations, the authors can only conclude that aqueous-phase and photochemical processing both play roles. How do they evaluate which is more important? Line 432, how do the authors conclude that photochemical processing enhanced during regional transport without any observation or interpretation regarding this (Also in the conclusion part, Line 499)? Line 434, the authors said that the impact of photochemical processing on MO-OOA production was limited in summer. Then what is the major formation mechanisms of MO-OOA in summer? Do they provide any evidence regarding different formation mechanism of LO-OOA and MO-OOA from this study? Line 505, the longer transport distance of air masses in summer does not mean that the influence of regional transport is strongest in summer. Not proper quantification.

Specific comments:

(1) In the introduction, Line 42, it's not proper to summarize that sulfate dominated in the south of NPC and nitrate dominate in the north of NCP. It is not determined by the location, but more by the emission characteristics.

(2) Line 55, the authors said that previous studies at background NCP site are limited by the low resolution. Any new findings do they draw from their high-resolution measurements?

(3) Line 105, to quantify aerosol concentrations, what are the RIE values of sulfate and ammonium according to the standard calibration?

(4) Since the measurement station is 960 meters above sea level, why do the authors choose the height of 500 m to permafrost back trajectories? How does it compare to 1000m or 1500m?

(5) Line 114, the AMS does not measure chlorine but chloride. Please revise the whole manuscript accordingly.

(6) Line 162, are high RH and high PM1 concentration correspond to air masses from the south?

(7) Line 170, it should be clearly noted that the frequency distribution of PM1 is shown in Fig. 1 using the white curve.

(8) Line 221, how do the authors define the wind dilution ratio? Please make it clear.

(9) In Fig. 3, how do the authors average WD? Do they follow the vector average wind direction?

(10) Line 280, please provide the number of correlation coefficient of POA vs. NOx and POA vs. chloride.

(11) Line 300, how do their correlations look like? According to previous studies, LO-OOA correlates better with nitrate, while MO-OOA better with sulfate.

(12) In Fig. 5, the nitrate time series is missing in Fig. 5b.

(13) Line 360, the higher O/C ratio in Xinglong should be due to the weak influence of primary emissions.

(14) Line 390, please define the Ox.

(15) Figure 9, the plots as a function of Ox not RH.

(16) Please define what is "dva" in the manuscript.

(17) Many grammatical mistakes. The reviewer recommends to do editing service. For example, in the title, it should be "highly time-resolved". Line 309, should be "be attributed to". Line 83, should be "a HR-ToF-AMS was deployed......with collocated measurements of meteorological parameters and gaseous species. " Line 89, should change "of" to "on".

References

Ulbrich, I. M., Canagaratna, M. R., Zhang, Q., Worsnop, D. R., and Jimenez, J. L.: Interpretation of organic components from Positive Matrix Factorization of aerosol mass spectrometric data, Atmos. Chem. Phys., 9, 2891-2918, 10.5194/acp-9-2891-2009, 2009.

Zhang, Q., Jimenez, J. L., Canagaratna, M. R., Ulbrich, I. M., Ng, N. L., Worsnop, D. R., Sun, Y. J. A., and Chemistry, B.: Understanding atmospheric organic aerosols via factor analysis of aerosol mass spectrometry: a review, 401, 3045-3067, 10.1007/s00216-011-5355-y, 2011.

Guo, H., Xu, L., Bougiatioti, A., Cerully, K. M., Capps, S. L., Hite Jr., J. R., Carlton, A. G., Lee, S.-H., Bergin, M. H., Ng, N. L., Nenes, A., and Weber, R. J.: Fine-particle water and pH in the southeastern United States, Atmos. Chem. Phys., 15, 5211–5228, https://doi.org/10.5194/acp-15-5211-2015, 2015.

Song, S., Gao, M., Xu, W., Shao, J., Shi, G., Wang, S., Wang, Y., Sun, Y., and McElroy, M. B.: Fine-particle pH for Beijing winter haze as inferred from different thermodynamic equilibrium models, Atmos. Chem. Phys., 18, 7423–7438, https://doi.org/10.5194/acp-18-7423-2018, 2018.

Guo, H., Weber, R. J., and Nenes, A.: High levels of ammonia do not raise fine particle pH sufficiently to yield nitrogen oxide-dominated sulfate production, Scientific Reports,

7, 12109, 10.1038/s41598-017-11704-0, 2017.

---

## Author Comment (AC1) · 25 Aug 2020

Response to referees' comments on "Seasonal variations in the highly time-resolved aerosol composition, sources, and chemical process of background submicron particles in North China Plain"

We highly appreciate the detailed valuable comments of the two referees on our manuscript. The suggestions are quite helpful for us to improve the quality of our paper. Please see the detailed point-by-point response below. We list the comments

in black, our replies in blue, and the changes in revised MS in red.

Anonymous Referee #1

This study was conducted at a regional background site of NCP for four seasons using a high-resolution aerosol mass spectrometer. Highly time-resolved chemistry and sources of submicron aerosols were investigated. The authors found that nitrate was the most abundant inorganic species and submicron particles were almost neutralized by excess ammonium in all seasons except summer. Source apportionment of organic aerosol (OA) identified two oxidized OAs, more-oxidized oxygenated OA (MO-OOA) and less-oxidized oxygenated OA (LO-OOA) in all seasons. Significant contributions of aged secondary organic aerosol in OA were observed in all seasons, especially in summer. The oxidation degree and evolution process of OAs in the four seasons and the comparison with urban studies were further investigated. Overall, the long-term dataset provided in this study is a valuable addition to the literature. I recommend publication after the following issues are addressed.

[Response] Thank you very much for your helpful comments and we have taken all of them into account in the revised version of the manuscript. Please see the detailed response marked blue below and the changes marked red in the revised manuscript.

Specific comments:

1. In the introduction, the importance of background site should be emphasized, which is important for highlighting the significance of this paper. [Response] Thanks for your suggestion. It is suggested that investigate the air quality and the chemical compositions of particles at the background sites could reflect the characteristics of regional air pollution. We have emphasized the importance of background site and highlighted the significance of our study. The revised part of the introduction section are as follows: "The high-resolution time-of-flight aerosol mass spectrometry (HR-ToF-AMS) has been widely used to characterize nonrefractory submicron particles (NR-PM1) at numerous urban sites and a few background sites on the Qinghai–Tibet Plateau (QTP) in western China, the Lake Hongze site in northern China, the Mount Wuzhi site in southern China, and the Mount Tai and Xinglong station in the NCP (Zhang et al., 2019;Xu et al., 2018;Du et al., 2015;Zhu et al., 2016;Zhang et al., 2020;Li et al., 2019). These previous studies on air quality and aerosol chemical composition in the background area of North China indicated that the aerosol species were well-mixed and highly aged from regional transport. Meanwhile, the background atmosphere had strong atmospheric oxidizing capacity and the organics were highly aged (Wang et al., 2013;Li et al., 2019). However, the evolution and formation mechanisms of SOA in the background area in the NCP are still unclear. Currently, researches focused on the evolution and formation mechanism of SOA is mainly concentrated in urban areas and the results varied in different places and seasons. For example, photochemical processing dominated the oxidized degree of OA in haze events, whereas aqueous-phase processing was the main reason that affected the oxidized degree of OA in foggy events in Hong Kong (Li et al., 2013;Qin et al., 2016). In urban Beijing, Xu et al. (2017a) found that aqueous-phase processing dominated MO-OOA formation in all seasons. While in Li's et al (2020) study, the impact of photochemistry on MO-OOA formation enhanced as the photochemical age increased in early autumn in Beijing. Due to the stronger atmospheric oxidizing capacity and higher oxidized degree of organics in the background atmosphere than in urban atmosphere in the NCP, the evolution and formation mechanisms of SOA must be different from those of urban areas and is of great significant to investigate, but the research in this field in background areas is limited. Also, there are only very limited studies investigating the seasonal difference so far. The formation and evolution of SOA vary greatly in different areas and seasons, mainly due to the complex interaction of local emissions, chemical reactions, and meteorological influences. Therefore, we presented four season measurements and discussed the seasonal difference in aerosol sources and formation processes. In particular, based on robust data analyses, we evaluated the influence of photochemical and aqueous-phase processing on different SOA productions (LO-OOA and MO-OOA) in different seasons in the background atmosphere."
2. Two OOA factors resolved in this study are less-oxidized OOA (LO-OOA) and more-oxidized OOA (MO-OOA), rather than semi-volatile OOA (SV-OOA) and low-volatile OOA (LV-OOA). Please provide more information about the defi̧nition for LO-OOA and MO-OOA? [Response] Thanks for your comments. OOA is usually identified in two categories, including low-volatility (LV-OOA) and semi-volatile OOA (SV-OOA), based on their correlations with sulfate and nitrate, respectively, and inferred volatilities. Previous studies confirmed the relative volatility characteristics of SV-OOA and LV-OOA by thermodenuder measurements (Huffman et al., 2009;Cappa and Jimenez, 2010). However, the terminology LO-OOA and MO-OOA for less and more oxidized OOA, respectively, is also appropriate, especially for datasets for which volatility dates are not available (Zhang et al., 2011). In this study, the high f44 values were permanent in the MS of both LO-OOA and MO-OOA. The f44 values for MO-OOA and LO-OOA in the four seasons ranged from 16.3 to 23.5%, 8.1 to 13.8%, respectively. The f43 values for MO-OOA and LO-OOA ranged from 4.8 to 5.2%, 6.8 to 9.1%, respectively. These behaviors indicated that MO-OOA had a higher oxidation degree than LO-OOA. The O/C ratios of the MO-OOA factors in the four seasons were 0.93, 0.93, 0.84, and 0.83, respectively, higher than those in the corresponding LO-OOA (0.69, 0.58, 0.67, and 0.49). More detailed information of PMF analysis can be seen in the revised supplementary materials.

3. The names of submicron aerosol species in the fi̧gures should be consistent, e.g., "ClâËŸ AËĞR" vs Chl, "NO3" vs. "NO3-". [Response] Thanks for your suggestion and the names of submicron aerosol species have been changed to be consistent in the revised manuscript.

4. Line 77-80: please provide more discussion and reference(s) about the higher atmospheric oxidizing capacity in the background site. [Response] Thanks for your suggestion and the related references have been added in the revised manuscript.

5. Line 153: Please add the standard deviation of the average PM1 concentrations of all seasons here. [Response] Thanks for your suggestion and the standard deviation of

the average PM1 concentrations were added in the revised manuscript in both Section 3.1.1 and conclusion (Section 4)

6. Line 154: The full name is needed when abbreviation is shown at the first place, such as "SIA" here. [Response] Thanks for your reminding. We have checked carefully and defined the abbreviation in the revised manuscript.

7. Line 337-339: "These characteristics were similar to the results found in previous researches conducted in urban Beijing showing that......" please add references. [Response] Thanks for your reminding. The related references have been added.

8. Line 328: "in the each season" should be "in each season". Line 309: "high oxidized ability" should be "high atmospheric oxidizing capacity". Please go through the manuscript for similar typos. [Response] We are so sorry for making the grammatical mistakes to make trouble to your review work. We have carefully checked and corrected the errors sentence by sentence.

––––––––––––––––––––––––––––––––

---

## Author Comment (AC2) · 25 Aug 2020

We highly appreciate the detailed valuable comments of the two referees on our manuscript. The suggestions are quite helpful for us to improve the quality of our paper. Please see the detailed point-by-point response below. We list the comments in black, our replies in blue, and the changes in revised MS in red.

Anonymous Referee #2

This paper reports the seasonal variations of submicron particles and its chemical

composition at a background station using HR-ToF-AMS. Using ME-2 analysis, the authors identified different sources of organic aerosol and explored the oxidation degree and evolution process of different OA in four seasons. Backward trajectory analysis was conducted to see the influence of air mass transport. The seasonal dataset of HR-TOF-AMS measurements at a background station in north China is valuable and suitable for a measurement report. But to publish as a scientific article in ACP, this reviewer did not find good novelty or significance of this study compared to previous findings. Also, many conclusions drawn in this study are not well supported by the observations or interpretations. Overall, this reviewer do not think the scientific significance of the paper meet the scope of ACP scientific articles and cannot recommend the publication of the paper.

[Response] Thank you very much for the detailed comments and the very careful reading of our manuscript. The suggestions are quite helpful for us. We have done our best to incorporate them in the revised manuscript to improve the quality of our paper. Please see the detailed point-by-point response below. We list the comments in black, our replies in blue, and the changes in revised MS in red.

Firstly, we are grateful for your recognition of the significant of our experiment work in the background station in Northern China.

Secondly, we agree with you that there are an increasing number of ACSM or AMS studies in Northern China, including a few of them conducted in background areas. These previous AMS studies indicated that the aerosol species in the background area of North China tend to be well-mixed from regional transport. Meanwhile, the organic aerosols at the background atmosphere are reported to be highly aged due to the strong atmospheric oxidizing capacity. It should be pointed out that although the long-term measurements and source analyses of fine particles have been intensely conducted in the urban areas of NCP (Liu et al., 2015; Sun et al., 2018), there still few of them were conducted in the background area in this highly polluted region, which usually lasting for a short period in one or two seasons (Zhang et al., 2017; Li et al.,

**ACPD**
2019b). Sources of fine particles reported to be varied greatly among the different seasons in NCP; for instance, coal combustion during periods requiring more domestic heating, biomass burning in harvest seasons, and dust storms in spring (Zhang et al., 2013; Huang et al., 2017). Thus, our understanding of seasonal variations of aerosol species and sources in the background areas of NCP remains guite poor. Furthermore, previous long-term aerosol studies in the background areas of NCP focused on limited aerosol species with daily sampling resolutions. No systematic measurements with high time resolution of the mass-size distributions of chemical components in fine aerosol particles, covering four seasons, have yet been reported. Thus, it is essential to accurately and objectively assess the physiochemistry characterization of various chemical components in the background areas of NCP, which would improve our understanding on the formation mechanism and aging processes of secondary aerosol on a regional scale. Therefore, in the present study, we present one year-round measurements of submicron aerosols (i.e., all four seasons) at a regional background site in NCP, to explore the seasonal variations in aerosol sources and formation processes. In particular, based on robust data analyses, we evaluated the influence of photochemical and aqueous-phase processing on SOA productions [LO-OOA (less oxidized OA) and MO-OOA], which has critical relevance in reflecting the general picture of anthropogenic emissions in similarly polluted regions. We believe that our present study provides essential information to the scientific community to improve our understanding of aerosol chemistry in background atmosphere.

Thirdly, the discussions and interpretations in section 3.3 and 3.4 about the evolution of secondary aerosol especially for SOA have been improved to show the conclusions clearly. The discussions in the introduction have been also improved to clearly show the scientific goals for this work. Please see detailed replies below.

Major concerns:

1. With similar data analysis and similar results from this study compared to so many published AMS papers, it is difficult to see the novelty and significance of this paper.
Regarding the background atmosphere in NCP, there have also been many studies focusing on air quality and particle chemical composition, including AMS studies. The authors should make it clear what is the specific values of this study. It should not be because you did measurements in a different location, or your measurement period is longer. Instead, the authors should state clearly the scientific questions or valuable findings from these measurements that can improve current understanding of aerosol chemistry.

[Response] Thanks for your comments. High-resolution time-of-flight aerosol mass spectrometry (HR-ToF-AMS) has been widely used to characterize nonrefractory submicron particles (NR-PM1) at numerous urban sites and a few background sites on the Tibetan Plateau in western China (Zhang et al., 2019), the Lake Hongze site in northern China, the Mount Wuzhi site in southern China (Zhu et al., 2016), the Mount Tai in central east China (Zhang et al., 2014) and the Mount Xinglong (Li et al., 2019b) and Shangdianzi station in the NCP (Zhang et al., 2017; Li et al., 2019b). These previous studies in the background area of North China indicated that the aerosol species tend to be well-mixed from regional transport. Meanwhile, the organic aerosols at the background atmosphere are also highly aged due to the strong atmospheric oxidizing capacity (Wang et al., 2013; Li et al., 2019b). It should be pointed out that although the long-term measurements and source analyses of fine particles have been intensely conducted in the urban areas of NCP (Liu et al., 2015; Sun et al., 2018), there still few of them were conducted in the background area in this highly polluted region, which usually lasting for a short period in one or two seasons (Zhang et al., 2017; Li et al., 2019b). Sources of fine particles reported to be varied greatly among the different seasons in NCP; for instance, coal combustion during periods requiring more domestic heating, biomass burning in harvest seasons, and dust storms in spring (Zhang et al., 2013; Huang et al., 2017). Thus, our understanding of seasonal variations of aerosol species and sources in the background areas of NCP remains guite poor. Furthermore, previous long-term aerosol studies in the background areas of NCP focused on limited aerosol species with daily sampling resolutions. No systematic measurements
with high time resolution of the mass-size distributions of chemical components in fine aerosol particles, covering four seasons, have yet been reported, which would hinder our understanding on the evolution and formation mechanisms of secondary aerosol on a regional scale. For example, photochemical processing was found to dominate the oxidation state of organic aerosol (OA) in haze events, whereas aqueous-phase processing was the main reason during foggy events in Hong Kong (Li et al., 2013; Qin et al., 2016). In urban Beijing, Xu et al. (2017) found that aqueous-phase processing dominated MO-OOA (more oxidized OA) formation in all seasons. Meanwhile. more recently, Li et al (2020) found that the impact of photochemistry on MO-OOA formation enhanced as the photochemical age increased in early autumn in Beijing. Due to the stronger atmospheric oxidizing capacity and higher oxidation state of organics in the background atmosphere than in the urban atmosphere over the NCP, the evolution and formation mechanisms of SOA would be largely different from those of urban areas, mainly due to the complex interactions of local emissions, chemical reactions, and meteorological influences. Whereas, these kinds of studies were still limited and hinder our understanding of background aerosol chemistry in polluted regions such as the NCP. Therefore, we present one year-round measurements of submicron aerosols at a regional background station in NCP, to explore the seasonal variations in aerosol sources and formation processes. Especially, the influence of photochemical and aqueous-phase processing on SOA productions [LO-OOA (less oxidized OA) and MO-OOA] were evaluated based on robust data analyses.

Our results suggested the dominant role of aqueous-phase processing on SOA formation in winter, while both of photochemical and aqueous-phase processing contribute to LO-OOA and MO-OOA production in spring and autumn. In summer, the photochemical processing dominant MO-OOA formation, but the role of aqueous-phase processing under moderate RH (40%

Ox associated with the increases of wind speed may imply important role of regional transport in SOA formation in summer. These results showed a clearer picture on the evolution of SOA at the background atmosphere, and could improve our understanding on the formation mechanism of SOA under the high atmospheric oxidizing capacity of NCP. The related discussion had been added in the introduction part and the conclusion part.

2. The reviewer cannot be convinced by the PMF analysis and result evaluation in this study, and details are lacking. (1) Regarding the first PMF run with PET, the authors showed in Fig. S2 that OOA was over split in a 5-factor solution without only showing the similar mass profiles. But how about the time series, diurnal variations and O/C ratios of different OOA factors? The authors should not conclude this by only checking the mass profiles. Are they representative of other OOA rather than LO-OOA and MO-OOA? For example, as the authors have been emphasizing the significance of aqueous-phase processing, is there any single OOA factor related to aqueous phase chemistry? Please provide these details either in the manuscript or SI.

[Response] Thanks for your comments. We have added the details of the PMF analysis in the revised manuscript.

As shown in Fig. S4 in the supplementary material, in the 5-factor solution in spring, factor 1, factor 2 and factor 4 had similar mass spectra, time series and O/C ratios (0.87-0.96). It was unclear if the three OOA components represent distinct sources or chemical types. Therefore, it was over split by one OOA factor. Another OOA factor (factor 5) had different mass spectra and lower O/C ratio, suggesting different formation mechanism.

Previous studies showed aqueous-phase processing plays an important role in formation of nitrogen-containing compounds. C2O2+, C2H2O2+ are typical fragment ions of methylglyoxal and glyoxal, that are precursors of SOA via cloud processing. CH2SO2+, CH3SO2+, and CH3SO+ are three typical fragment ions of methane
sulfonic acid, which are products mainly from the oxidation of dimethyl sulfide and can be strongly enhanced by aqueous-phase processing (Xu et al., 2017a;Xu et al., 2017b;Xu et al., 2017b;Xu et al., 2019). In this study, factor 2 and factor 4 were poorly correlated with aqueous-processing related fragment ions (e.g., C2O2+, C2H2O2+, CH2SO2+, CH3SO2+, and CH3SO+; R2=0.38-0.48). Factor 1 and Factor 5 were poorly correlated with N-containing ions (e.g., CH4N+, C2H6N+, and C3H8N+, R2=0.42-0.53). Therefore, it was lack of sufficient evidence to identify a single OOA factor related to aqueous phase chemistry.

Because of the over split of the OOA factors in the 5-factor solution and the difficulty to separate POA from OOA by PMF analysis, we constrained the FFOA profiles separated by the 5-factor solution of PMF analysis in spring to better separate FFOA from OOA. As a result, three OA factors, including FFOA, LO-OOA and MO-OOA, were identified with ME-2 analysis in spring.

(2) To perform ME-2 analysis, the authors constrain the FFOA profiles with the POA factor resolved in the five-factor solution of PMF analysis in spring and apply it to all seasons. One concern here is that how do the authors believe the POA factor from the five-factor solution is good enough to represent the primary sources? How does the profile look like when performing PMF analysis to six or seven factors? Does the POA factor in this study comparable to those resolved from AMS studies previously, especially those in NCP area?

[Response] Thanks for your comments. The mass spectra of the POA from the 5-factor solution was compared with those resolved from AMS studies previously in Xi'an and Beijing according to the method in Dou's et al (2009) research:  $\cos(\theta)=(MSAMSB)/(|MSA||MSB|)$  where MSA and MSB are the two AMS mass spectra. The correlation coefficient R is equal to the cosine of the angle  $\theta$  that we will use in this work for mass spectra comparison (Dou et al., 2009).

As shown in Table 1, the POA factor resolved in the 5-factor solution had some similarity
with HOA and CCOA, but not exactly the same as HOA or CCOA. In addition, the correlation coefficient between POA and NOx was 0.58, and that between POA and chloride was 0.78, suggesting a significant contribution of coal combustion and traffic-related sources to the POA factor in Xinglong. Previous studies found that HOA and CCOA showed remarkably similar mass spectrum patterns when m/z is below 120 (Sun et al., 2016; Sun et al., 2018), which is sometimes difficult to be separated by PMF analysis, so the POA factor resolved in this study can be considered as a combined factor of HOA and CCOA. Therefore, the POA factor was identified as fossil fuel OA (FFOA).

The POA factor from the 5-factor solution in spring was compared with the POA factor from 6- and 7-factor solutions of PMF analysis. In the 6- or 7-factor solutions (Fig. S5-S6), two POA factors appeared, but none of them was good enough to represent the primary sources in Xinglong. The two POA factors were more likely to be over split of FFOA. Take 6-factor solution as an example, as shown in table 1, the factor 4 resolved in the 6-factor solution was different with HOA or CCOA. Although the similarity between factor 1 resolved in the 6-factor solution and HOA and CCOA were not significantly weaker than those of the POA factor resolved in the 5-factor solution, the mass fraction of factor 1 in total OA was only 5%, which is in the uncertainty range of PMF analysis. Meanwhile, the correlation coefficient between the factor 4 and NOx and chloride were lower than that of the POA factor resolved in the 5-factor solution. Therefore, the POA factor resolved in the 5-factor solution. Therefore, the POA factor resolved in the 5-factor solution.

In conclusion, the POA factor resolved from the 5-factor solution in spring was compared with the POA factors from 6- and 7-factor solutions of PMF analysis. Results showed that it was good enough for the POA factor to represent the primary source. The POA factor was also compared with primary sources resolved from previous studies in Beijing and Xi'an. Results showed that the POA factor had some similarity with both HOA and CCOA. Therefore, it was identified as FFOA.
Another big concern is that how robust it is to apply this factor from spring to all the other seasons. To check this, the author should perform PET PMF analysis for the other season and see if a POA factor will appear when go to more factors, and then check if the mass profile of POA are comparable in all seasons. As shown in Fig. 5, the mass contribution of FFOA in total OA is 5%, which is in the uncertainty range of PMF analysis. The authors should prove if it is reasonable to manually constrain the ME-2 to separate a single FFOA factor in summer.

[Response] Thanks for your comments. To check if the POA factor resolved in spring is comparable in all seasons, we performed PMF analysis in all seasons. In autumn, the solution of the PMF analysis was similar to that for the spring observation. A POA factor appeared until the 7-factor solution and OOA was over-split. The POA factor was similar with the POA factor resolved in spring (the angle  $\theta$  between the two mass spectra is 9 degree). To further improve the accuracy of ME-2 analysis in autumn, we constrained the POA profile separated by the 7-factor solution of PMF analysis in autumn for ME-2 analysis in the revised manuscript.

In summer, ME-2 was used in the previous manuscript to separated POA from OOA by constrained the POA factor resolved in spring. However, no POA factor appeared in the 2- to 9-factor solutions of the PMF analysis, suggesting the faction of POA in OA in summer was too low to be identified. Meanwhile, the mass fraction of POA in total OA was only 5%, which was in the uncertainty range of PMF analysis. Therefore, the PMF analysis results of two OA factors (LO-OOA and MO-OOA) were used in the revised manuscript in summer.

In winter, the 3-factor solution (FFOA, LO-OOA and MO-OOA) from PMF analysis was good enough, so it was not necessary to use ME-2 analysis to separate POA from OOA. The POA factor resolved by PMF analysis in winter was similar with the POA factor resolved in spring (the angle  $\theta$  between the two mass spectra is 10 degree) by ME-2 analysis and the results of the two methods were similar. To make the result more accurate, the 3-factor solution of PMF analysis in the revised manuscript was
used. Please see detailed information on how to select the optimum PMF solution in each season in FigureS18-S20 and Table S2 in the supplementary material:

Determination of the PMF and ME-2 solution Springiii Z Factor number from 1 to 8 were selected to run in the PMF model. For the spring observation, there was no POA factor appeared in the 2- to 4- factor solution. A POA factor appeared in the 5-factor solution and diagnostic plots of the PMF analysis were shown in Fig. S2. The mass spectra of the POA factor had some similarity with HOA and CCOA. The correlation coefficient between POA and NOx was 0.58, and that between POA and chloride was 0.78. suggesting a significant contribution of coal combustion and traffic-related sources to the POA factor in Xinglong. Previous studies found that HOA and CCOA showed remarkably similar mass spectrum patterns when m/z is below 120 (Sun et al., 2016; Sun et al., 2018), which was sometimes difficult to be separated by PMF analysis, so FFOA could be considered as a combined factor of HOA and CCOA (Sun et al., 2018). Therefore, the POA factor was identified as FFOA. As shown in Fig. S4, in the 5factor solution, factor 1, factor 2 and factor 4 had similar mass spectra, time series and O/C ratios (0.87-0.96). It was unclear if the three OOA components represent distinct sources or chemical types. Therefore, it was over split by one OOA factor. Another OOA factor (factor 5) had different mass spectra and lower O/C ratio, suggesting different formation mechanism. Therefore, we constrained the FFOA profiles separated by the 5-factor solution of PMF analysis in spring. As a result, three OA factors, including FFOA, LO-OOA and MO-OOA, were identified with ME-2 analysis in spring. The mass spectra, time series, and diurnal variations of ME-2 result were shown in Fig. S7.

Figure S2. Diagnostic plots of the PMF analysis on OA mass spectral matrix for the spring observation.

Summer: For the summer observation, the 2-factor, fpeak=0 solution was selected as the optimum solution. The diagnostic plots of the PMF analysis were shown in Fig. S8. The two OA factors are more oxidized (MO-OOA) and less oxidized OOA (LO-OOA). The mass spectrum, time series and diurnal variations of OA factors were different.
The O/C of the two factors were 0.58 and 0.93, respectively. No POA factor appeared in the 2- to 9-factor solutions. OOA was over spilt in the 3- to 9-factor solutions. The detailed information on how to select the optimum PMF solution can be found in Figure S9-S12 and Table S1.

Autumn: The solution of the PMF analysis for the autumn observation is similar to that for the spring observation. A POA factor appeared until the 7-factor solution and OOA was over-split. The diagnostic plots of the PMF analysis were shown in Fig. S13. The correlation coefficient of the POA vs. NOx is 0.61. The mass spectra of the POA factor is similar with the FFOA factor resolved in spring (the angle  $\theta$  between the two mass spectra is 9 degree). We constrained the POA profile separated by the 7-factor solution of PMF analysis in autumn to better separate POA from OOA. As a result, three OA factors, including FFOA, LO-OOA and MO-OOA, were identified with ME-2 analysis in autumn. The mass spectra, time series, and diurnal variations of ME-2 result were shown in Fig. S16.

Winter: For the winter observation, the 3-factor, fpeak=0 solution was selected as the optimum solution. When OA was separated into four factors, OOA was also split into three factors (Fig. S19). In the 4-factor solution, factor 2 was similar to factor 2 which was resolved in the 3-factor solution with similar mass spectra, time series, diurnal variation and O/C ratios. However, factor 3 and factor 4 in the four-factor solution had similar O/C ratios, time series and diurnal variation. It was unclear if the two OOA components represent distinct sources or chemical types. When more than 5 factors, OOA decomposed into three or more factors. Thus, two OOA factors were combined into total OOA for further analysis. The 3-factor solution (FFOA, LO-OOA and MO-OOA) from PMF analysis was good enough, so it was not necessary to use ME-2 analysis to separate POA from OOA. The detailed information on how to select the optimum PMF solution can be found in FigureS18-S20 and Table S2.

(3) Details about the evaluation of PMF and ME-2 results are lacking. To interpret PMF/ME-2 results, the authors should carefully follow the procedures proposed by

**ACPD**
Ulbrich et al. (2009) and Zhang et al. (2011). For example, evaluation of Q/Qexp as a function of factor number and Q/Qexp or scaled residual as a function of m/z should be provided in the manuscript or SI. [Response] Thanks for your suggestion. We added more details about the evaluation of PMF and ME-2 results in the revised supplementary materials, as showed above.

3. It is questionable that the authors solely based on the ratio of measured NH4+ to predicted NH4+ to evaluate aerosol acidity. While this ratio has been used as an indicator of aerosol acidity by some previous AMS studies, it has been proved recently that this ion balance method is not reliable to evaluate particle acidity (Guo et al., 2015; Song et al., 2018). Thermodynamic models, i.e., E-AIM and ISORROPIA, should be used. With thermodynamic models, Guo et al. (2017) found that aerosol is always acidic in NPC region, which is contradictory to results from this study.

[Response] Thanks for your suggestion. We agree with you that the ion balance method is not reliable to evaluate particle acidity and we reevaluate the particle acidity by the thermodynamic model. Liu et al. (2017) and Song et al. (2019) found that pH values in ISORROPIA were on average 0.3-0.4 units higher than E-AIM under winter haze conditions. In addition, the ISORROPIA model is more widely used and has more results to compare, so that it was used in the revised manuscript. The forward mode was used with just aerosol-phase data and NH3 data input in this study, to avoid measurement error (Song et al., 2018;Guo et al., 2017). An ammonia (NH3) analyzer (NH3-H2O, Model 911-0016, LGR) was also used to simultaneously measure NH3. Notably, data for RH < 30% and RH > 95% were excluded because of the large uncertainty in LWC and pH values (Ding et al., 2019; Guo et al., 2015). Results showed that the aerosol in Xinglong were acidity in summer (PH: 2.7 iCs 0.6) and moderate acidity in spring (4.2 iCs 0.7), autumn (3.5 iCs 0.5) and winter (3.7 iCs 0.6), consistent with previous studies that although NH3 in the NCP was abundant, the aerosol was far from neutral (Ding et al., 2019). Please see detailed analysis as follows: "The acidity of PM1 was evaluated in each season using the thermodynamic model ISORROPIA-II (Table
2). PM1 in Xinglong showed moderate acidity in spring, autumn and winter, with average pH values of 4.2 ïĆś 0.7, 3.5 ïĆś 0.5 and 3.7 ïĆś 0.6, respectively. Comparatively, the pH value in summer was the lowest (2.7 iCs 0.6) among all seasons, similar to the findings of previous studies (Ding et al., 2019; Liu et al., 2017). The seasonal variation in the pH at Xinglong was similar to results reported in urban Beijing, except for spring and winter. The pH value in urban Beijing was highest in winter, followed by spring (4.4 ïĆś 1.2), autumn (4.3 ïĆś 0.8), and summer (3.8 ïĆś 1.2) (Ding et al., 2019). The seasonal variation of pH in this study was strongly related to the chemical composition of aerosols in each season. Previous studies show that, compared to an elevated nitrate concentration, an elevated sulfate concentration can result in higher acidity because of the low volatility of sulfate (Tan et al., 2018; Xu et al., 2019). In this study, the mass fraction of sulfate in PM1 was highest in summer (37%) and lowest in spring (16%). Similarly, the nitrate-to-sulfate ratio was highest in spring (2.13) and lowest in summer (0.26). Recent studies have shown that sulfate has been effectively reduced in Beijing because of the strict emission control measures, and the mass fraction of nitrate in PM has increased significantly, with an increased NO3/SO4 ratio in Beijing ubiquitously observed (Xu et al., 2019; Song et al., 2019). Notably, the pH in spring (4.2 ïĆś 0.7) was similar to the value found in urban Beijing (4.4 ïĆś 1.2) in the same season. In comparison, the pH values in autumn and winter were 0.5 to 1.1 lower than those found in urban areas of northern China, such as 4.3 ïCś 0.8 in autumn in Beijing, 4.5 (3.8-5.2) in winter in Zhengzhou, and 4.8 (3.9-5.9) in winter in Anyang (Ding et al., 2019; Wang et al., 2020). The higher LWC in urban areas may be one of the important reasons for its slightly lower acidity in autumn and winter compared to that in background areas in northern China. Aerosol acidity is closely related to LWC, with higher LWC usually accompanied by higher aerosol pH according to previous studies (Guo et al., 2015; Liu et al., 2017). In this study, the LWC in autumn and winter was 18 ïĆś 38 and 12 ïĆś 26, respectively, which was obviously lower than that in urban areas, such as 109 ïĆś 160 ïA∎g m-3 in autumn in Beijing (Ding et al., 2019), 220 (28-711)iĂăiA∎g m-3 in winter in Beijing (Liu et al., 2017), and 95 iA∎g m-3 in winter

**ACPD**
in Zhengzhou (Wang et al., 2020). The higher LWC in urban areas is mainly due to the high aerosol concentrations, which can enhance aerosol water uptake (Liu et al., 2017). In comparison, the Hair+ concentration was 3.7ïĂăïA∎g m-3 in autumn and 1.2 iA=g m-3 in winter, which were comparable with those observed in urban areas. Therefore, the lower LWC/Hair+ ratio in Xinglong favored the slightly lower pH values, according to equation (1). Moreover, the pH values in this study and previous studies in urban areas of the NCP were 0.8 to 3.5 units higher than those observed in the U.S. and Europe, such as 1.2 iCs 1.1 in Crete, Greece, in winter (Bougiatioti et al., 2016); 0.9 ïĆś 0.6 in Alabama, southeastern U.S., in summer (Guo et al., 2015); and 2.2 ïĆś 0.6 in Yorkville, southeastern U.S., in autumn (Nah et al., 2018). The excessive NH3 emissions in the NCP play an important role in the large gap (Song et al., 2019). In this study, the average NH3 concentrations were 12, 19, 8 and 4 ppb in each season, with a maximum value of 39 ppb, while in the southeastern U.S., NH3 generally ranged between 0.1 and 3.0 ppb (Weber et al., 2016). Another explanation might be the changes in chemical composition of aerosols over the NCP. The NO3/SO4 ratios in this study and urban areas of the NCP were obviously higher than those in other countries (Table 2). Thus, the relatively lower aerosol acidity in the NCP might be attributable to the excessive NH3 and high NO3/SO4 ratios on a regional scale in this region."

4. Many interpretations or conclusions from this study cannot be well supported by the observations. For example, the authors conclude in the end of the abstract that the neutralized state of submicron particles highlight the significance of NOx and ammonia reduction (also in conclusion part, Line 486 and Line 510). But these two do not have causal relationship.

[Response] Thanks for pointing this out. We now admit that the mentioned conclusion in the abstracts is not accurate, and we revised this sentence as showed below: "Our results illustrate that the background particles in the NCP are influenced significantly by aging processes and regional transport, and the increased contribution of aerosol nitrate highlights how regional reductions in emissions of nitrogen oxide are critical for
remedying occurrence of nitrate-dominated haze events over the NCP."

Line 215, the authors should not make a conclusion that the high NOx concentration is solely due to strong influence of traffic.

[Response] Thanks for pointing this out. We agree that the high NOx concentration in winter would not be mainly contributed from the traffic emissions. NOx exhibited its highest concentration in winter and correlated well with chloride (R2 = 0.6), suggesting a strong influence of fossil fuel combustion, such as coal combustion. Therefore, the high NOx concentration in winter was mainly due to the coal combustion, while regional transport from heavily polluted regions may also contribute partly.

Line 280, the authors conclude that POA factor was closely related to traffic emission without evaluating the characteristic of POA. While POA showed a good correlation with NOx, does its diurnal profile show morning and evening peaks, which is a typical feature of traffic-related pollutant? It is not correct to conclude that the high concentration of FFOA at night indicate the high primary emissions. The variations in boundary layer height play a significant role. Similar issues happened again in Sect. 3.3. The authors draw conclusions regarding species formation mechanisms during daytime and nighttime based on the variations in their concentrations without considering the major influence of boundary layer height.

[Response] Thanks for your suggestions. The mass spectra of the POA factor had some similarity with HOA and CCOA (Hu et al., 2017;Elser et al., 2016), which was detailed compared in the response to the second major question. The correlation coefficient between POA and NOx was 0.58, and that between POA and chloride was 0.78, in spring, suggesting a significant contribution of coal combustion and traffic-related sources to the POA factor in Xinglong. Therefore, the POA factor in spring could be considered as FFOA. The concentrations of FFOA were obviously higher at night than during the daytime during the four seasons, which was mainly due to the variations in PBL. The diurnal profile of FFOA showed evening peak in autumn and winter, while

ACPD
showed no morning peaks. In spring, the diurnal profile of FFOA showed no morning and evening peaks. This diurnal profile of FFOA may be related to the observation position of Xinglong Station, which is located on a mountain with an altitude of 960 m, surrounded by forests and farmland, and more than 100 kilometers away from the urban site. The distance from urban area to the site and the altitude of the site might allow time for substantial vertical mixing and dilution. Therefore, the traffic emissions of diurnal profile of FFOA showed no obvious morning and evening peaks like Beijing and other urban areas. Similar diurnal profile of HOA was also observed in in suburbs site located in the downwind of the urban site (de Sá et al., 2018).

According to your suggestion, we considered the major influence of the variations in boundary layer height on the diurnal variation of PM1 species. The concentrations of FFOA were higher at night than during the daytime during the four seasons, which was mainly due to the variations in the PBL. The lower PBL at night suppressed the diffusion of pollutants. Meanwhile, higher primary emissions, such as coal-burning emissions, at night than during the daytime, also partly contributed. Nitrate exhibited drastic diurnal variation in each season, with a high concentration at night and low concentration during the daytime. This behavior was closely related to the variation of the PBL, which reduced the concentration of nitrate during the daytime and suppressed the diffusion of nitrate at night. MO-OOA exhibited similar drastic diurnal profiles in spring, autumn and winter, peaking in the afternoon and at night. The high concentration of MO-OOA at night was likely due to the co-effect of the low PBL and aqueous chemistry under high RH conditions at night (Hu et al., 2017; Sun et al., 2018). Although the PBL expanded, the concentrations of MO-OOA increased significantly from 12:00 to 18:00, indicating an important role played by photochemical processes in MO-OOA production during the daytime in these three seasons. Please see detailed analysis in line 369 to 380 in the revised manuscript.

Line 323, why the authors think the increase in nitrate concentration is caused by regional transport? The observations cannot support this. In the end of this paragraph,
the authors emphasize the strong effects of local chemical production and regional transport on nitrate diurnal pattern. But the reviewer did not find any related discussion about the relative contribution of local production and regional transport. Again, Line 336, why the increased concentration from noon to evening indicate the regional characteristics of MO-OOA? Overall, these conclusions without detailed interpretation and well supported observations would confuse the audience a lot.

[Response] Thanks for pointing this out. We now admit that the increase in nitrate concentration cannot be solely attributed to regional transport. As shown in Fig. 5, nitrate exhibited drastic diurnal variation in each season, with a high concentration at night and low concentration during the daytime. This behavior was closely related to the variation of the PBL, which reduced the concentration of nitrate during the daytime and suppressed the diffusion of nitrate at night. Heterogeneous/aqueous-phase reactions and gas-to-particle condensation processes are the main pathways to forming finemode nitrate (Sun et al., 2018; Hu et al., 2017). The high concentration of nitrate in each season suggested the pathway of the hydrolysis of dinitrogen pentoxide (N2O5) to nitrate formation at night in Xinglong might be strong due to low NO concentrations and high O3 concentrations, even at night. The NO concentrations at Xinglong Station in the four seasons were as low as 0.2 to 0.7 ppb (Table 1). Because of the low concentration of NO, it would be difficult for NO to react with O3 and thus deplete O3 so that O3 could accumulate, even at night. O3 concentrations at night in the four seasons were about 45, 70, 35 and 25 ppb, respectively, which showed that the background atmosphere exhibited high atmospheric oxidation capacity, even at night, especially in summer. The diurnal variation of nitrate showed an obvious increase from noon through the afternoon in each season, suggesting the increased nitrate concentrations were influenced by photochemical production. Nitrate exhibited its lowest concentration in summer, which can be attributed to the evaporation of NH4NO3 due to the high temperatures (Fig. 5). Therefore, the nitrate diurnal pattern might be influenced by the variation of PBL and local chemical production, including the hydrolysis of N2O5 to nitrate formation at night and photochemical processes during the daytime.
Similar with nitrate, MO-OOA also exhibited drastic diurnal profiles in spring, autumn and winter, peaking in the afternoon and at night. The high concentration of MO-OOA at night was likely due to the co-effect of the low PBL and aqueous chemistry under high RH conditions at night (Hu et al., 2017; Sun et al., 2018). Although the PBL expanded in the afternoon, the concentrations of MO-OOA increased significantly from 12:00 to 18:00, indicating an important role played by photochemical processes in MO-OOA production during the daytime in these three seasons. In summer, the concentration of MO-OOA showed its greatest increase rate (0.18  $\mu$ gÂům–3Âůh–1) from 09:00 to 18:00, implying stronger photochemical production of MO-OOA than in other seasons. Note that the wind speed in summer also increased rapidly from 09:00 to 16:00, along with the increase of MO-OOA, which may suggest regional transport also partly contributed to the rapid increase in MO-OOA during the daytime in summer. The related discussion was added in the revised manuscript.

"MO-OOA exhibited similar drastic diurnal profiles in spring, autumn and winter, peaking in the afternoon and at night. The high concentration of MO-OOA at night was likely due to the co-effect of the low PBL and aqueous chemistry under high RH conditions at night (Hu et al., 2017; Sun et al., 2018). Although the PBL expanded, the concentrations of MO-OOA increased significantly from 12:00 to 18:00, indicating an important role played by photochemical processes in MO-OOA production during the daytime in these three seasons. In summer, the concentration of MO-OOA showed its greatest increase rate (0.18  $\mu$ gÂům-3Âůh-1) from 09:00 to 18:00, implying stronger photochemical production of MO-OOA than in other seasons. Note that the wind speed in summer also increased rapidly from 09:00 to 16:00, along with the increase of MO-OOA, which may suggest regional transport also partly contributed to the rapid increase in MO-OOA during the daytime in summer. The diurnal profiles of LO-OOA in each season were flatter than those of MO-OOA. The decreased PBL mainly resulted in a higher concentration of LO-OOA at night. The increased concentration of LO-OOA from noon through the afternoon was mainly due to photochemical processes. The highest concentration of LO-OOA in summer suggested a stronger photochemical production of Interactive comment

LO-OOA than in other seasons."

Line 426, from the observations, the authors can only conclude that aqueous-phase and photochemical processing both play roles. How do they evaluate which is more important?

[Response] Thanks for pointing this out. Yes, we agree that it is hard to say which is more important but can only conclude that aqueous-phase and photochemical processing both play roles in spring, autumn and winter, as the two OOA factors increased with the elevation of both RH and Ox in these three seasons (Fig.7 and 8). Whereas, as showed in Fig. 7 and Fig.8, the mass fraction of MO-OOA in OA did not increase as Ox elevated in winter, while it increased from 30% to 40% as RH increased from 30 to 90%. This characteristic suggested a more important role of aqueous-phase processing on SOA formation than photochemical processing in winter. The related discussion was added in the revised manuscript. "Note that the mass fraction of MO-OOA did not increase as Ox elevated in winter, while it increased ~from 30% to 40% as RH increased from 30 to 90%. This characteristic suggested a more emportant role of aqueous-phase processing on SOA formation than photochemical processing in winter. The related discussion was added in the revised manuscript. "Note that the mass fraction of MO-OOA did not increase as Ox elevated in winter, while it increased ~from 30% to 40% as RH increased from 30 to 90%. This characteristic suggested a more dominant important role of aqueous-phase processing on SOA formation than photochemical processing in this season."

Line 432, how do the authors conclude that photochemical processing enhanced during regional transport without any observation or interpretation regarding this (Also in the conclusion part, Line 499)?

[Response] Thanks for your reminding, and we now admit that it cannot conclude that photochemical processing enhanced during regional transport based on the present results. In fact, both LO-OOA and MO-OOA showed overall increasing trends as Ox increased in summer. In addition, increases of LO-OOA and MO-OOA as functions of Ox were clearly associated with the increases of wind speed, which was more significant in summer than in other seasons. In comparison, in urban areas, such as Beijing, increase of LO-OOA were associated with the decreases of wind speed, that facilitated
the accumulation of air pollutants (Xu et al., 2017). Such a difference between urban and background areas may be due to the influence of regional transport on the Ox and SOA concentrations in background areas. We revised this part and the related discussion was provided as below:

"In summer, both LO-OOA and MO-OOA showed overall increasing trends as Ox increased, while RH showed a corresponding overall decreasing trend. This behavior indicates a strong influence of photochemical processing on both LO-OOA and MO-OOA production. Meanwhile, LO-OOA showed a continuously decreasing trend as RH increased in summer, except for a slightly increasing trend when RH increased from 40 to 60%, indicating photochemical processing dominated LO-OOA formation. MO-OOA increased significantly with Ox, while it increased slightly with RH (40%

production was limited in summer. Then what is the major formation mechanisms of MO-OOA in summer? Do they provide any evidence regarding different formation mechanism of LO-OOA and MO-OOA from this study?

[Response] Sorry for the misunderstanding. The line 434 in previous manuscript referred to the impact of processing on MO-OOA production was limited in summer in the urban site in Beijing. According to Xu's et al (2017) research, MO-OOA had no obvious changes as Ox increased, while increased continuously as RH increased, indicating aqueous-phase processing dominant MO-OOA formation in summer. In comparison, in background Xinglong, MO-OOA showed overall increasing trends as Ox increased, while RH showed a corresponding overall decreasing trend. This behavior indicates the strong influence of photochemical processing on MO-OOA production. Meanwhile, MO-OOA increased slightly with RH (40%

and by 12% in winter when RH increased from 20 to 90%. Corresponding, the mass fraction of LO-OOA decreased by 15% in autumn. These characteristics indicated that aqueous-phase processing plays more important role in MO-OOA formation than that in LO-OOA in these two seasons. Detailed discussion can be seen in Line 405 to 417.

Line 505, the longer transport distance of air masses in summer does not mean that the influence of regional transport is strongest in summer. Not proper quantification.

[Response] Thanks for your reminding. We admit that the previous conclusion is not proper and we deleted it, and rewrite as follows: "The air masses from the southern regions (clusters 2 and 3) accounted for 56% of all the air masses in summer, which was obviously higher than the percentage in other seasons (27–38%). Cluster 3 in summer started at Bohai Bay and passed through the Shandong Peninsula and over Bohai Bay. The PM1 concentrations for clusters 2 (14.7 ïAmg m-3) and 3 (12.2 ïAmg m-3) were both high. These results suggest a dominant role played by southern transport in submicron aerosol concentrations over the NCP in summer."

Specific comments: (1) In the introduction, Line 42, it's not proper to summarize that sulfate dominated in the south of NPC and nitrate dominate in the north of NCP. It is not determined by the location, but more by the emission characteristics.

[Response] Thanks for your reminding. This sentence was revised to "sulfate dominates the secondary inorganic aerosols (SIAs) in heavy-industry cities such as Shijiazhuang and Handan, while in recent years nitrate has dominated those in Beijing because of the strict emissions reduction measures for coal combustion."

(2) Line 55, the authors said that previous studies at background NCP site are limited by the low resolution. Any new findings do they draw from their high-resolution measurements?

[Response] Thanks for your comment. The HR-ToF-AMS can provide elemental information, such as hydrogen-to-carbon (H/C), organic-mass-to-organic-carbon (OM/OC),
and oxygen-to-carbon (O/C), which can help to quantify the oxidation degree of OA (Jimenez, 2003). OOA can also be separated as more-oxidized OOA (MO-OOA) and less-oxidized OOA (LO-OOA) due to the different O/C ratios (Zhang et al., 2011). According to the high-resolution measurements, our results suggested the dominant role of aqueous-phase processing on SOA formation than photochemical processing in winter. Aqueous-phase processing plays more important role in MO-OOA formation than that in LO-OOA in autumn and winter. Both of photochemical and aqueous-phase processing contribute to LO-OOA and MO-OOA production in spring. In summer, the photochemical processing dominant MO-OOA formation, but the role of aqueous-phase processing under moderate RH (40%

long could reduce to 100-200m during haze episode (Li, et al., 2020). Thus, we choose the height of 200m (AGL) and recalculate the air mass trajectories in the revised MS. According to the suggestion of comparing to 1000m and 1500m, we found these two heights was not suitable for our observation sites. If they are refer to the height above sea level, then it was too low for the height of 1000m which was only 40m above ground level that the air mass may be blocked by mountains on the moving ways, while for the height of 1500m (refer to 540m AGL), it would sometimes above the PBL and thus could not arrive to the receptor site. If they are refer to the height above ground level, then these two heights are further above the PBL and thus could not arrive to the receptor site. To make it clear, we revised this part. "The 48-h back trajectories were calculated every hour at a height of 200 m (above ground level) using the HYSPLIT (Hybrid Single-Particle Lagrangian Integrated Trajectories) model"

(5) Line 114, the AMS does not measure chlorine but chloride. Please revise the whole manuscript accordingly.

[Response] Thanks for your reminding. We have carefully corrected similar mistakes in the revised manuscript.

(6) Line 162, are high RH and high PM1 concentration correspond to air masses from the south?

[Response] Thanks for your comment. As shown in Fig. 9, high RH and high PM1 concentration correspond to air masses from the south in all seasons. In summer, the RH and PM1 concentration were also at high levels of the air massed from the north and northwest regions of Xinglong. Detailed discussion can be seen in Section 3.4.

(7) Line 170, it should be clearly noted that the frequency distribution of PM1 is shown in Fig. 1 using the white curve.

[Response] Thanks for your reminding. We have clearly noted that the frequency distribution of PM1 was shown in Fig. 1 using the white curve in the revised manuscript.
(8) Line 221, how do the authors define the wind dilution ratio? Please make it clear.

[Response] Thanks for your suggestion. The wind dilution ratio was defined as the percentage decrease in the concentration of the aerosol species for every 1 m s-1 decrease in wind speed (Sun et al., 2013). We have made it clear in the revised manuscript.

(9) In Fig. 3, how do the authors average WD? Do they follow the vector average wind direction?

[Response] Thanks for your comment. In the previous manuscript, we divided the data into 8 directions (0-45°, 45-90°, 90-135°, 135-180°, 180-225°, 225-270°, 270-315°, 315-360°) according to the wind direction and calculated the average wind direction by arithmetic average. However, it's more reasonable to divided the data according to the vector average wind direction. Therefore, we divided the data into 8 directions (N, NE, E, SE, S, SW, W, NW) in the revised manuscript and now the revised Fig.2 (Fig. 3 in previous manuscript) is as follows:

(10) Line 280, please provide the number of correlation coefficient of POA vs. NOx and POA vs. chloride.

[Response] Thanks for your suggestion. The correlation coefficient between POA and NOx was 0.58, and that between POA and chloride was 0.78, in spring. The correlation coefficient between POA and NOx was 0.78, and that between POA and chloride was 0.61, in winter. The correlation coefficient between POA and NOx was 0.61 in autumn. Detailed information can be seen in the supplementary materials.

(11) Line 300, how do their correlations look like? According to previous studies, LO-OOA correlates better with nitrate, while MO-OOA better with sulfate.

[Response] Thanks for your suggestion. The correlation coefficient between LO-OOA and nitrate (sulfate) was 0.79 (0.58) in autumn and 0.71 (0.44) in winter. Therefore, LO-OOA correlated well with nitrate in autumn and winter. MO-OOA, meanwhile, corre-
lated well with both nitrate and sulfate in autumn (MO-OOA vs. nitrate: 0.86; MO-OOA vs. sulfate: 0.87) and winter (MO-OOA vs. nitrate: 0.79; MO-OOA vs. sulfate: 0.72), similar to the findings of previous studies in wintertime in Beijing (Hu et al., 2017). In comparison, LO-OOA had a low correlation coefficient with nitrate or sulfate, and MO-OOA had a correlation coefficient of 0.65 with sulfate, in summer. The poor correlation between LO-OOA and secondary inorganic species has also been found in a previous study (Sun et al., 2018). Both of LO-OOA and MO-OOA correlated well with nitrate and sulfate in spring.

(12) In Fig. 5, the nitrate time series is missing in Fig. 5b.

[Response] Thanks for your reminding. The time series of nitrate was added in Fig. 5b

(13) Line 360, the higher O/C ratio in Xinglong should be due to the weak influence of pimary emissions.

[Response] Thanks for your comment and we have corrected the sentence as follows: "The O/C ratios in Xinglong in all seasons (0.54–0.75) were slightly higher than those in urban Beijing (0.47–0.53), mainly due to the weak influence of primary emissions."

(14) Line 390, please define the Ox.

[Response] Thanks for your reminding. Ox was defined in the revised manuscript.

(15) Figure 9, the plots as a function of Ox not RH.

[Response] Thanks for your reminding. The caption of Fig. 9 was corrected in the revised manuscript.

(16) Please define what is "dva" in the manuscript.

[Response] Thanks for your reminding. "dva" (vacuum dynamic diameter) was defined in the revised manuscript.

(17) Many grammatical mistakes. The reviewer recommends to do editing service.

**ACPD**
For example, in the title, it should be "highly time-resolved". Line 309, should be "be attributed to". Line 83, should be "a HR-ToF-AMS was deployed. . . . . . with collocated measurements of meteorological parameters and gaseous species." Line 89, should change "of" to "on".

[Response] We are so sorry for making the grammatical mistakes to make trouble to your review work. Thanks for your useful comments and suggestions to improve the manuscript. We have done editing service and carefully checked and corrected the errors sentence by sentence.

References

Bougiatioti, A., Nikolaou, P., Stavroulas, I., Kouvarakis, G., Weber, R., Nenes, A., Kanakidou, M., and Mihalopoulos, N.: Particle water and pH in the eastern Mediterranean: source variability and implications for nutrient availability, Atmospheric Chemistry and Physics, 16, 4579-4591, 10.5194/acp-16-4579-2016, 2016.

Cappa, C. D., and Jimenez, J. L.: Quantitative estimates of the volatility of ambient organic aerosol, Atmospheric Chemistry and Physics, 10, 5409-5424, 10.5194/acp-10-5409-2010, 2010.

de Sá, S. S., Palm, B. B., Campuzano-Jost, P., Day, D. A., Hu, W., Isaacman-VanWertz, G., Yee, L. D., Brito, J., Carbone, S., Ribeiro, I. O., Cirino, G. G., Liu, Y., Thalman, R., Sedlacek, A., Funk, A., Schumacher, C., Shilling, J. E., Schneider, J., Artaxo, P., Goldstein, A. H., Souza, R. A. F., Wang, J., McKinney, K. A., Barbosa, H., Alexander, M. L., Jimenez, J. L., and Martin, S. T.: Urban influence on the concentration and composition of submicron particulate matter in central Amazonia, Atmospheric Chemistry and Physics, 18, 12185-12206, 10.5194/acp-18-12185-2018, 2018.

Ding, J., Zhao, P., Su, J., Dong, Q., Du, X., and Zhang, Y.: Aerosol pH and its driving factors in Beijing, Atmospheric Chemistry and Physics, 19, 7939-7954, 10.5194/acp-19-7939-2019, 2019.

**ACPD**
Du, W., Sun, Y., Xu, Y., Jiang, Q., Wang, Q., Yang, W., Wang, F., Bai, Z., Zhao, X., and Yang, Y.: Chemical characterization of submicron aerosol and particle growth events at a national background site (3295 m asl) on the Tibetan Plateau, Atmospheric Chemistry and Physics, 15, 10811-10824, 2015.

Duan, J., Huang, R.-J., Li, Y., Chen, Q., Zheng, Y., Chen, Y., Lin, C., Ni, H., Wang, M., Ovadnevaite, J., Ceburnis, D., Chen, C., Worsnop, D. R., Hoffmann, T., amp, apos, Dowd, C., and Cao, J.: Summertime and wintertime atmospheric processes of secondary aerosol in Beijing, Atmospheric Chemistry and Physics, 20, 3793-3807, 10.5194/acp-20-3793-2020, 2020.

Elser, M., Huang, R.-J., Wolf, R., Slowik, J. G., Wang, Q., Canonaco, F., Li, G., Bozzetti, C., Daellenbach, K. R., Huang, Y., Zhang, R., Li, Z., Cao, J., Baltensperger, U., El-Haddad, I., and Prévôt, A. S. H.: New insights into PM2.5 chemical composition and sources in two major cities in China during extreme haze events using aerosol mass spectrometry, Atmospheric Chemistry and Physics, 16, 3207-3225, 10.5194/acp-16-3207-2016, 2016.

Guo, H., Xu, L., Bougiatioti, A., Cerully, K. M., Capps, S. L., Hite, J. R., Carlton, A. G., Lee, S. H., Bergin, M. H., Ng, N. L., Nenes, A., and Weber, R. J.: Fine-particle water and pH in the southeastern United States, Atmospheric Chemistry and Physics, 15, 5211-5228, 10.5194/acp-15-5211-2015, 2015.

Guo, H., Weber, R. J., and Nenes, A.: High levels of ammonia do not raise fine particle pH sufficiently to yield nitrogen oxide-dominated sulfate production, Sci Rep, 7, 12109, 10.1038/s41598-017-11704-0, 2017.

Hu, W., Hu, M., Hu, W.-W., Zheng, J., Chen, C., Wu, Y., and Guo, S.: Seasonal variations in high time-resolved chemical compositions, sources, and evolution of atmospheric submicron aerosols in the megacity Beijing, Atmospheric Chemistry and Physics, 17, 9979-10000, 10.5194/acp-17-9979-2017, 2017.
Huffman, J., Docherty, K., Aiken, A., Cubison, M., Ulbrich, I., DeCarlo, P., Sueper, D., Jayne, J., Worsnop, D., and Ziemann, P.: Chemically-resolved aerosol volatility measurements from two megacity field studies, Atmospheric Chemistry and Physics, 9, 7161-7182, 2009.

Jimenez, J. L.: Ambient aerosol sampling using the Aerodyne Aerosol Mass Spectrometer, Journal of Geophysical Research, 108, 10.1029/2001jd001213, 2003.

Li, J., Liu, Z., Cao, L., Gao, W., Yan, Y., Mao, J., Zhang, X., He, L., Xin, J., Tang, G., Ji, D., Hu, B., Wang, L., Wang, Y., Dai, L., Zhao, D., Du, W., and Wang, Y.: Highly time-resolved chemical characterization and implications of regional transport for submicron aerosols in the North China Plain, Science of The Total Environment, 135803, https://doi.org/10.1016/j.scitotenv.2019.135803, 2019.

Li, J., Liu, Z., Gao, W., Tang, G., Hu, B., Ma, Z., and Wang, Y.: Insight into the formation and evolution of secondary organic aerosol in the megacity of Beijing, China, Atmospheric Environment, 220, 117070, https://doi.org/10.1016/j.atmosenv.2019.117070, 2020.

Li, Y. J., Lee, B. Y. L., Yu, J. Z., Ng, N. L., and Chan, C. K.: Evaluating the degree of oxygenation of organic aerosol during foggy and hazy days in Hong Kong using high-resolution time-of-flight aerosol mass spectrometry (HR-ToF-AMS), Atmospheric Chemistry and Physics, 13, 8739-8753, 10.5194/acp-13-8739-2013, 2013.

Liu, M., Song, Y., Zhou, T., Xu, Z., Yan, C., Zheng, M., Wu, Z., Hu, M., Wu, Y., and Zhu, T.: Fine particle pH during severe haze episodes in northern China, Geophysical Research Letters, 44, 5213-5221, 10.1002/2017gl073210, 2017.

Liu, Z., Hu, B., Wang, L., Wu, F., Gao, W., and Wang, Y.: Seasonal and diurnal variation in particulate matter (PM10 and PM2:5/ at an urban site of Beijing: analyses from a 9-year study, Environ. Sci. Pollut. Res., 22, 627–642, https://doi.org/10.1007/s11356-014-3347-0, 2015.

ACPD
Nah, T., Guo, H., Sullivan, A. P., Chen, Y., Tanner, D. J., Nenes, A., Russell, A., Ng, N. L., Huey, L. G., and Weber, R. J.: Characterization of aerosol composition, aerosol acidity, and organic acid partitioning at an agriculturally intensive rural southeastern US site, Atmospheric Chemistry and Physics, 18, 11471-11491, 10.5194/acp-18-11471-2018, 2018.

Qin, Y. M., Li, Y. J., Wang, H., Lee, B. P. Y. L., Huang, D. D., and Chan, C. K.: Particulate matter (PM) episodes at a suburban site in Hong Kong: evolution of PM characteristics and role of photochemistry in secondary aerosol formation, Atmospheric Chemistry and Physics, 16, 14131-14145, 10.5194/acp-16-14131-2016, 2016. Song, S., Gao, M., Xu, W., Shao, J., Shi, G., Wang, S., Wang, Y., Sun, Y., and McElroy, M. B.: Fine-particle pH for Beijing winter haze as inferred from different thermodynamic equilibrium models, Atmospheric Chemistry and Physics, 18, 7423-7438, 10.5194/acp-18-7423-2018, 2018.

Song, S., Nenes, A., Gao, M., Zhang, Y., Liu, P., Shao, J., Ye, D., Xu, W., Lei, L., Sun, Y., Liu, B., Wang, S., and McElroy, M. B.: Thermodynamic Modeling Suggests Declines in Water Uptake and Acidity of Inorganic Aerosols in Beijing Winter Haze Events during 2014/2015–2018/2019, Environmental Science & Technology Letters, 6, 752-760, 10.1021/acs.estlett.9b00621, 2019. Sun, Y., Du, W., Fu, P., Wang, Q., Li, J., Ge, X., Zhang, Q., Zhu, C., Ren, L., and Xu, W.: Primary and secondary aerosols in Beijing in winter: sources, variations and processes, Atmospheric Chemistry and Physics, 16, 8309-8329, 2016.

Sun, Y., Xu, W., Zhang, Q., Jiang, Q., Canonaco, F., Prévôt, A. S., Fu, P., Li, J., Jayne, J., and Worsnop, D. R.: Source apportionment of organic aerosol from 2-year highly time-resolved measurements by an aerosol chemical speciation monitor in Beijing, China, Atmospheric Chemistry and Physics, 18, 8469-8489, 2018.

Sun, Y. L., Wang, Z. F., Fu, P. Q., Yang, T., Jiang, Q., Dong, H. B., Li, J., and Jia, J. J.: Aerosol composition, sources and processes during wintertime in Beijing, China, At-
mospheric Chemistry and Physics, 13, 4577-4592, 10.5194/acp-13-4577-2013, 2013.

Tan, T., Hu, M., Li, M., Guo, Q., Wu, Y., Fang, X., Gu, F., Wang, Y., and Wu, Z.: New insight into PM2.5 pollution patterns in Beijing based on one-year measurement of chemical compositions, Sci Total Environ, 621, 734-743, 10.1016/j.scitotenv.2017.11.208, 2018.

Wang, S., Wang, L., Li, Y., Wang, C., Wang, W., Yin, S., and Zhang, R.: Effect of ammonia on fine-particle pH in agricultural regions of China: comparison between urban and rural sites, Atmospheric Chemistry and Physics, 20, 2719-2734, 10.5194/acp-20-2719-2020, 2020.

Wang, Y., Hu, B., Tang, G., Ji, D., Zhang, H., Bai, J., Wang, X., and Wang, Y.: Characteristics of ozone and its precursors in Northern China: A comparative study of three sites, Atmospheric research, 132, 450-459, 2013.

Weber, R. J., Guo, H., Russell, A. G., and Nenes, A.: High aerosol acidity despite declining atmospheric sulfate concentrations over the past 15 years, Nature Geoscience, 9, 282-285, 10.1038/ngeo2665, 2016.

Xu, J., Zhang, Q., Shi, J., Ge, X., Xie, C., Wang, J., Kang, S., Zhang, R., and Wang, Y.: Chemical characteristics of submicron particles at the central Tibetan Plateau: insights from aerosol mass spectrometry, Atmospheric Chemistry and Physics, 18, 2018.

Xu, W., Han, T., Du, W., Wang, Q., Chen, C., Zhao, J., Zhang, Y., Li, J., Fu, P., Wang, Z., Worsnop, D. R., and Sun, Y.: Effects of Aqueous-Phase and Photochemical Processing on Secondary Organic Aerosol Formation and Evolution in Beijing, China, Environ Sci Technol, 51, 762-770, 10.1021/acs.est.6b04498, 2017a.

Xu, W., Sun, Y., Wang, Q., Du, W., Zhao, J., Ge, X., Han, T., Zhang, Y., Zhou, W., and Li, J.: Seasonal Characterization of Organic Nitrogen in Atmospheric Aerosols Using High Resolution Aerosol Mass Spectrometry in Beijing, China, ACS Earth and Space Chemistry, 1, 673-682, 2017b.

**Supplement:**

**Supplementary material of**

**Seasonal variations of the highly time-resolved aerosol composition, sources, and chemical processes of background submicron particles in North China Plain**

Jiayun Li[1, 4], Liming Cao[2], Wenkang Gao[1], Lingyan He[2]; Yingchao Yan[1,4], Dongsheng Ji[1], Zirui Liu[1], Yuesi Wang[1,3,4]

[1]State Key Laboratory of Atmospheric Boundary Layer Physics and Atmospheric Chemistry (LAPC), Institute of Atmospheric Physics, Chinese Academy of Sciences, Beijing 100029, China

[2]Key Laboratory for Urban Habitat Environmental Science and Technology, Peking University Shenzhen Graduate School, Shenzhen, 518055, China

[3]Center for Excellence in Regional Atmospheric Environment, Institute of Urban Environment, Chinese Academy of Sciences, Xiamen 361021, China

[4]University of Chinese Academy of Sciences, Beijing 100049, China

Correspondence: Zirui Liu (liuzirui@mail.iap.ac.cn); Lingyan He (hely@pkusz.edu.cn)

Jiayun Li and Liming Cao contributed equally to this work.

[Figure]

**Figure S1.** Scatter plots of the mass concentration of NR-PM$_1$ vs. total PM$_1$ measured by Sharp-5030 in spring (a), summer (b), autumn (c), and winter (d).

**Determination of the PMF and ME-2 solution**

**Spring:**

Factor number from 1 to 8 were selected to run in the PMF model. For the spring observation, there was no POA factor appeared in the 2- to 4- factor solution. A POA factor appeared in the 5-factor solution and diagnostic plots of the PMF analysis were shown in Fig. S2. The mass spectra of the POA factor had some similarity with HOA and CCOA. The correlation coefficient between POA and NOx was 0.58, and that between POA and chloride was 0.78, suggesting a significant contribution of coal combustion and traffic-related sources to the POA factor in Xinglong. Previous studies found that HOA and CCOA showed remarkably similar mass spectrum patterns when m/z is below 120 (Sun et al., 2016; Sun et al., 2018), which was sometimes difficult to be separated by PMF analysis, so FFOA could be considered as a combined factor of HOA and CCOA (Sun et al., 2018). Therefore, the POA factor was identified as FFOA.

As shown in Fig. S4, in the 5-factor solution, factor 1, factor 2 and factor 4 had similar mass spectra, time series and O/C ratios (0.87-0.96). It was unclear if the three OOA components represent distinct sources or chemical types. Therefore, it was over split by one OOA factor. Another OOA factor (factor 5) had different mass spectra and lower O/C ratio, suggesting different formation mechanism. Therefore, we constrained the FFOA profiles separated by the 5-factor solution of PMF analysis in spring. As a result, three OA factors, including FFOA, LO-OOA and MO-OOA, were identified with ME-2 analysis in spring. The mass spectra, time series, and diurnal variations of ME-2 result were shown in Fig. S7.

[Figure]

Figure S2. Diagnostic plots of the PMF analysis on OA mass spectral matrix for the spring observation.

[Figure]

**Figure S3.** The mass spectra, time series, and diurnal variations of 4-factor solution of PMF analysis for the spring observation.

[Figure]

**Figure S4.** The mass spectra, time series, and diurnal variations of 5-factor solution of PMF analysis for the spring observation.

[Figure]

**Figure S5.** The mass spectra, time series, and diurnal variations of 6-factor solution of PMF analysis for the spring observation.

[Figure]

**Figure S6.** The mass spectra, time series, and diurnal variations of 7-factor solution of PMF analysis for the spring observation.

[Figure]

**Figure S7.** The mass spectra, time series, and diurnal variations of ME-2 analysis for the spring observation.

**Summer:**

For the summer observation, the 2-factor, fpeak=0 solution was selected as the optimum solution. The diagnostic plots of the PMF analysis were shown in Fig. S8. The two OA factors are more oxidized (MO-OOA) and less oxidized OOA (LO-OOA). The mass spectrum, time series and diurnal variations of OA factors were different. The O/C of the two factors were 0.58 and 0.93, respectively. No POA factor appeared in the 2- to 9-factor solutions. OOA was over spilt in the 3- to 9-factor solutions. The detailed information on how to select the optimum PMF solution can be found in Figure S9-S12 and Table S1.

[Figure]

Figure S8. Diagnostic plots of the PMF analysis on OA mass spectral matrix for the summer observation.

[Figure]

**Figure S9.** The mass spectra, time series, and diurnal variations of 2-factor solution of PMF analysis for the summer

observation.

[Figure]

**Figure S10.** The mass spectra, time series, and diurnal variations of 3-factor solution of PMF analysis for the summer

observation.

[Figure]

**Figure S11.** The mass spectra, time series, and diurnal variations of 4-factor solution of PMF analysis for the summer

observation.

[Figure]

**Figure S12.** The mass spectra, time series, and diurnal variations of 5-factor solution of PMF analysis for the summer

observation.

**Table S1** Descriptions of PMF solutions for the summer observation in Beijing.

| Factor number | Fpeak | Q/Qexp | Solution Description |
|---|---|---|---|
| 1 | 0 | 3.83 | Too few factors, large residuals at time periods and key m/z's |
| **2** | **0** | **3.32** | **Optimum solution for the PMF analysis (MO-OOA and LO-OOA). The mass spectrum, time series and diurnal variations of OA factors were different. The O/C of the two factors were 0.58 and 0.93, respectively.** |
| 3-9 | 0 | 3.11 | Factor split. Take 3 factor number solution as an example, factor 2 was similar to the factor 2 which resolved in the 2-factor solution with similar mass spectrum, time series, diurnal variation and O/C ratios. Factor 1 and factor 3 were likely over split with similar time series and different mass spectrum. However, it was difficult to explain if they represent distinct sources or chemical types. |

**Autumn:**

The solution of the PMF analysis for the autumn observation is similar to that for the spring observation. A POA factor appeared until the 7-factor solution and OOA was over-split. The diagnostic plots of the PMF analysis were shown in Fig. S13. The correlation coefficient of the POA vs. NOx is 0.61. The mass spectra of the POA factor is similar with the FFOA factor resolved in spring (the angle θ between the two mass spectra is 9 degree). We constrained the POA profile separated by the 7-factor solution of PMF analysis in autumn to better separate POA from OOA. As a result, three OA factors, including FFOA, LO-OOA and MO-OOA, were identified with ME-2 analysis in autumn. The mass spectra, time series, and diurnal variations of ME-2 result were shown in Fig. S16.

[Figure]

c

Figure S13. Diagnostic plots of the PMF analysis on OA mass spectral matrix for the autumn observation.

[Figure]

**Figure S14.** The mass spectra, time series, and diurnal variations of 6-factor solution of PMF analysis for the autumn

observation.

[Figure]

**Figure S15.** The mass spectra, time series, and diurnal variations of 7-factor solution of PMF analysis for the autumn observation.

[Figure]

**Figure S16.** The mass spectra, time series, and diurnal variations of ME-2 analysis for the autumn observation.

**Winter:**

For the winter observation, the 3-factor, fpeak=0 solution was selected as the optimum solution. When OA was separated into four factors, OOA was also split into three factors (Fig. S19). In the 4-factor solution, factor 2 was similar to factor 2 which was resolved in the 3-factor solution with similar mass spectra, time series, diurnal variation and O/C ratios. However, factor 3 and factor 4 in the 4-factor solution had similar O/C ratios, time series and diurnal variation. It was unclear if the two OOA components represent distinct sources or chemical types. When more than 5 factors, OOA decomposed into three or more

factors. Thus, two OOA factors were combined into total OOA for further analysis. The 3-factor solution (FFOA, LO-OOA and MO-OOA) from PMF analysis was good enough, so it was not necessary to use ME-2 analysis to separate POA from OOA. The detailed information on how to select the optimum PMF solution can be found in FigureS18-S20 and Table S2.

[Figure]

Figure S17. Diagnostic plots of the PMF analysis on OA mass spectral matrix for the winter observation.

[Figure]

**Figure S18.** The mass spectra, time series, and diurnal variations of 3-factor solution of PMF analysis for the winter observation.

[Figure]

**Figure S19.** The mass spectra, time series, and diurnal variations of 4-factor solution of PMF analysis for the winter observation.

[Figure]

**Figure S20.** The mass spectra, time series, and diurnal variations of 5-factor solution of PMF analysis for the winter observation.

**Table S2** Descriptions of PMF solutions for the winter observation in Beijing.

| Factor number | Fpeak | Q/Qexp | Solution Description |
|---|---|---|---|
| 1 | 0 | 2.65 | Too few factors, large residuals at time periods and key m/z's. |
| 2 | 0 | 2.34 | Too few factors, POA was mixed with OOA. |
| **3** | **0** | **2.12** | **Optimum solution for the PMF analysis (FFOA, MO-OOA and LO-OOA). The mass spectrum and time series of the two OOA factors were different. The O/C of the two factors were 0.49 and 0.83, respectively. Thus, two OOA factors were for further analysis. The correlation coefficient between POA and NOx was 0.73, and that between POA and chloride was 0.61. Meanwhile, the mass spectra of the POA factor was similar to HOA and CCOA. Therefore, the POA factor was identified to FFOA.** |

| | | | |
|---|---|---|---|
| 4-7 | 0 | 1.89-1.98 | Factor split. Take 4 factor number solution as an example, factor 2 was similar to the factor 2 which resolved in the 2-factor solution with similar mass spectrum, time series, diurnal variation and O/C ratios. factor 1 and factor 3 were likely over split with similar time series and different mass spectrum. However, it was difficult to explain if they represent distinct sources or chemical types. |

---

## Author Response (AR1)

**Response to referees' comments on "Seasonal variations of the highly time-resolved aerosol composition, sources, and chemical processes of background submicron particles in North China Plain"**

We highly appreciate the detailed valuable comments of the two referees on our manuscript. The suggestions are quite helpful for us to improve the quality of our paper. Please see the detailed point-by-point response below. We list the comments in black, our replies in blue, and the changes in revised MS in red.

**Anonymous Referee #1**

This study was conducted at a regional background site of NCP for four seasons using a high-resolution aerosol mass spectrometer. Highly time-resolved chemistry and sources of submicron aerosols were investigated. The authors found that nitrate was the most abundant inorganic species and submicron particles were almost neutralized by excess ammonium in all seasons except summer. Source apportionment of organic aerosol (OA) identified two oxidized OAs, more-oxidized oxygenated OA (MO-OOA) and less-oxidized oxygenated OA (LO-OOA) in all seasons. Significant contributions of aged secondary organic aerosol in OA were observed in all seasons, especially in summer. The oxidation degree and evolution process of OAs in the four seasons and the comparison with urban studies were further investigated. Overall, the long-term dataset provided in this study is a valuable addition to the literature. I recommend publication after the following issues are addressed.

[Response] Thank you very much for your helpful comments and we have taken all of them into account in the revised version of the manuscript. Please see the detailed response marked blue below and the changes marked red in the revised manuscript.

**Specific comments:**

1. In the introduction, the importance of background site should be emphasized, which is important for highlighting the significance of this paper.

[Response] Thanks for your suggestion. It is suggested that investigate the air quality and the chemical compositions of particles at the background sites could reflect the characteristics of regional air pollution. We have emphasized the importance of background site and highlighted the significance of our study. The revised part of the introduction section

are as follows:

[revised manuscript text omitted]

2. Two OOA factors resolved in this study are less-oxidized OOA (LO-OOA) and more-oxidized OOA (MO-OOA), rather than semi-volatile OOA (SV-OOA) and low-volatile OOA (LV-OOA). Please provide more information about the definition for LO-OOA and MO-OOA?

[Response] Thanks for your comments. OOA is usually identified in two categories, including low-volatility (LV-OOA) and semi-volatile OOA (SV-OOA), based on their correlations with sulfate and nitrate, respectively, and inferred volatilities. Previous studies confirmed the relative volatility characteristics of SV-OOA and LV-OOA by thermodenuder measurements (Huffman et al., 2009;Cappa and Jimenez, 2010). However, the terminology LO-OOA and MO-OOA for less and more oxidized OOA, respectively, is also appropriate, especially for datasets for which volatility dates are not available (Zhang et al., 2011). In this study, the high $f44$ values were permanent in the mass spectrum of both LO-OOA and MO-OOA. The $f44$ values for MO-OOA and LO-OOA in the four seasons ranged from 16.3 to 23.5% and 8.1 to 13.8%, respectively. The $f43$ values for MO-OOA and LO-OOA ranged from 4.8 to 5.2% and 6.8 to 9.1%, respectively. These behaviors indicated that MO-OOA had a higher oxidation state than LO-OOA. The O/C ratios of the MO-OOA factors in the four seasons were 0.93, 0.93, 0.84 and 0.83, respectively—higher than those

in the corresponding LO-OOA (0.69, 0.58, 0.67 and 0.49). Please see detailed information of PMF analysis in the revised supplementary material.

3. The names of submicron aerosol species in the figures should be consistent, e.g., "Clâˇ AˇR" vs Chl, "NO3" vs. "NO3-".

[Response] Thanks for your suggestion and the names of submicron aerosol species have been changed to be consistent in the revised manuscript.

4. Line 77-80: please provide more discussion and reference(s) about the higher atmospheric oxidizing capacity in the background site.

[Response] Thanks for your suggestion and the related references have been added in the revised manuscript.

5. Line 153: Please add the standard deviation of the average PM1 concentrations of all seasons here.

[Response] Thanks for your suggestion and the standard deviation of the average $PM_1$ concentrations were added in the revised manuscript in both Section 3.1.1 and conclusion (Section 4).

6. Line 154: The full name is needed when abbreviation is shown at the first place, such as "SIA" here.

[Response] Thanks for your reminding. We have checked carefully and defined the abbreviation in the revised manuscript.

7. Line 337-339: "These characteristics were similar to the results found in previous researches conducted in urban Beijing showing that......" please add references.

[Response] Thanks for your reminding. The related references have been added.

8. Line 328: "in the each season" should be "in each season". Line 309: "high oxidized ability" should be "high atmospheric oxidizing capacity". Please go through the manuscript for similar typos.

[Response] We are so sorry for making the grammatical mistakes to make trouble to your review work. We have carefully checked and corrected the errors sentence by sentence.

**Anonymous Referee #2**

This paper reports the seasonal variations of submicron particles and its chemical composition at a background station using HR-ToF-AMS. Using ME-2 analysis, the authors identified different sources of organic aerosol and explored the oxidation degree and evolution process of different OA in four seasons. Backward trajectory analysis was conducted to see the influence of air mass transport. The seasonal dataset of HR-TOF-AMS measurements at a background station in north China is valuable and suitable for a measurement report. But to publish as a scientific article in ACP, this reviewer did not find good novelty or significance of this study compared to previous findings. Also, many conclusions drawn in this study are not well supported by the observations or interpretations. Overall, this reviewer do not think the scientific significance of the paper meet the scope of ACP scientific articles and cannot recommend the publication of the paper.

[Response] Thank you very much for the detailed comments and the very careful reading of our manuscript. The suggestions are quite helpful for us. We have done our best to incorporate them in the revised manuscript to improve the quality of our paper. Please see the detailed point-by-point response below. We list the comments in black, our replies in blue, and the changes in revised MS in red.

Firstly, we are grateful for your recognition of the significant of our experiment work in the background station in Northern China.

Secondly, we agree with you that there are many studies focusing on air quality and particle chemical composition in the background site in the NCP with a low temporal resolution of one or several days (Pan et al., 2013; Liu et al., 2018; Huang et al., 2017), which indicated that secondary aerosols dominate the aerosol particles at background sites and that regional transport affects the air pollution of the background atmosphere. However, the high time resolution studies at background sites in the NCP by HR-ToF-AMS are still limited. Zhang et al., (2017) investigated the NR-PM$_1$ species in 2015 in a background site (Shangdianzi) by a HR-ToF-AMS and Li et al., (2019b) conducted the observation in winter 2018 in a background site (Xinglong) by a Q-AMS in the NCP. Both studies showed the dominant role of organics and nitrate and organics were highly oxidized in the background atmosphere in the NCP. However, both of the two studies were concentrated in wintertime, and the compositions of PM$_1$, the sources and oxidation state of organic aerosol (OA) changed significantly in different background sites and seasons. For example, nitrate and OA dominated PM$_1$ in autumn and winter at Xinglong and Waliguan sites in the NCP and in spring at Mount Wuzhi site in Northern China (Zhu et al.,

2016), while sulfate and OA dominated $PM_1$ in summer at Waliguan site in Western China (Zhang et al., 2019) and in spring at Lake Hongze site in Southern China (Zhu et al., 2016). Organics were highly oxidized at these background sites, while the oxidation state of organics in southern China were higher than that in Northern China (Zhu et al., 2016). Hydrocarbon-like OA (HOA) accounted for 30% in OA in spring at Wuzhi Mount in northern China. Biomass burning OA and aged biomass burning OA accounted for 18% and 40% in OA in summer, respectively, while HOA only accounted for 6%, in western China (Zhang et al., 2019). No primary organic sources were found in Southern China in spring (Zhu et al., 2016). Theses substantial differences highlighted the importance of long-term observations for understanding the seasonal characteristics of aerosol species and sources in the background atmosphere. What's more, pollutants in the background areas are frequently influenced by the regional transport from the urban areas, highlighting the importance of the seasonal variations of the impacts of regional transport on aerosol mass loadings and chemistry at background site. However, no systematic measurements with high time resolution of the mass–size distributions of chemical components in fine aerosol particles, covering four seasons, have yet been reported in the background atmosphere in the NCP, which would hinder our understanding on the evolution and chemical processes of secondary aerosol and regional transport on a regional scale. Only a study investigated the seasonal variations in $PM_1$ species at the background site in central east China at Mount Tai (Zhang et al., 2014b) based on unit mass resolution (UMR). The study only presented the characterization of the total OA; the sources of OA were not identified. What's more, there was no elemental information, which can determine the oxidation state of OA and characterize the evolution and processes of secondary organic aerosol (SOA). Actually, the evolution and formation mechanisms of SOA in the background atmosphere are still unclear. Currently, research focusing on this field tends to be concentrated in urban areas, and the results vary in different places and seasons. For example, photochemical processing was found to dominate the oxidation state of organic aerosol (OA) in haze events, whereas aqueous-phase processing was the main reason during foggy events in Hong Kong (Li et al., 2013; Qin et al., 2016). In urban Beijing, Xu et al. (2017) found that aqueous-phase processing dominated MO-OOA (more oxidized OA) formation in all seasons. Meanwhile, more recently, Li et al (2020) found that the impact of photochemistry on MO-OOA formation enhanced as the photochemical age increased in early autumn in Beijing. Due to the stronger atmospheric oxidizing capacity and higher oxidation state of organics in the background atmosphere than in the urban atmosphere over the NCP, the evolution and formation mechanisms of SOA would be largely different from those of urban areas, mainly due to the complex interactions of local emissions, chemical reactions, and meteorological influences. Whereas, these kinds of studies were still limited and hinder our understanding of

background aerosol chemistry in polluted regions such as the NCP. Therefore, we present one year-round measurements of submicron aerosols at a regional background station in NCP, to explore the seasonal variations in aerosol sources and formation processes. Especially, the influence of photochemical and aqueous-phase processing on SOA productions [LO-OOA (less oxidized OA) and MO-OOA] were evaluated based on robust data analyses.

In our study, we found obviously seasonal variations in chemical species, aerosol acidity, regional transport, oxidation state of OA and evolution and formation mechanisms of SOA. We believe that our present study provides essential information to the scientific community to improve our understanding of aerosol chemistry in background atmosphere.

Thirdly, the discussions and interpretations in section 3.3 and 3.4 about the evolution of secondary aerosol especially for SOA have been improved to show the conclusions clearly. The discussions in the introduction have been also improved to clearly show the scientific goals for this work. Please see detailed replies below.

**Major concerns:**

1. With similar data analysis and similar results from this study compared to so many published AMS papers, it is difficult to see the novelty and significance of this paper. Regarding the background atmosphere in NCP, there have also been many studies focusing on air quality and particle chemical composition, including AMS studies. The authors should make it clear what is the specific values of this study. It should not be because you did measurements in a different location, or your measurement period is longer. Instead, the authors should state clearly the scientific questions or valuable findings from these measurements that can improve current understanding of aerosol chemistry.

[Response] Thanks for your comments. There are many studies focusing on air quality and particle chemical composition in the background site in the NCP with a low temporal resolution of one or several days (Pan et al., 2013; Liu et al., 2018; Huang et al., 2017), which indicated that secondary aerosols dominate the aerosol particles at background sites and that regional transport affects the air pollution of the background atmosphere. However, the high time resolution studies at background sites in the NCP by HR-ToF-AMS are still limited. Zhang et al., (2017) investigated the NR-PM$_1$ species in 2015 in a background site (Shangdianzi) by a HR-ToF-AMS and Li et al., (2019b) conducted the observation in winter 2018 in a background site (Xinglong) by a Q-AMS in the NCP. Both studies showed the dominant role of organics and

[revised manuscript text omitted]

In our study, we found obviously seasonal variations in chemical species, aerosol acidity, regional transport, oxidation state of OA and evolution and formation mechanisms of SOA. Specifically, nitrate dominated $PM_1$ in spring, autumn and winter, while sulfate dominated $PM_1$ in summer. Aerosol was acidity in summer and moderate acidity in other three seasons. The oxidation state of OA was highest in summer because of the stronger photochemical processing. Backward trajectory analysis showed that higher concentrations of submicron particles were associated with air masses transported short distances from the southern regions in all four seasons, while long-range transport from Inner Mongolia (western and northern regions) also contributed to summertime particulate pollution in the background areas of the NCP. Our results suggested the dominant role of aqueous-phase processing on SOA formation in winter, while both of photochemical and aqueous-phase processing contribute to LO-OOA and MO-OOA production in spring and autumn. In summer, the photochemical processing dominant MO-OOA formation, but the role of aqueous-phase processing under moderate RH (40%<RH<60%) condition cannot be ruled out. In comparison, LO-OOA formation was mainly contributed by photochemical processing in summer. In addition, regional transport also played an important role in the variations of SOA, especially in summer that continuous increases in SOA concentration as a function of Ox was found to be associated with the increases of wind speed. These results showed a clearer picture on the evolution of $PM_1$ species and SOA at the background atmosphere, and could improve our understanding on the formation mechanism of SOA under the high atmospheric oxidizing capacity of NCP. The related discussion had been added in the introduction part and the conclusion part.

2. The reviewer cannot be convinced by the PMF analysis and result evaluation in this study, and details are lacking. (1) Regarding the first PMF run with PET, the authors showed in Fig. S2 that OOA was over split in a 5-factor solution

without only showing the similar mass profiles. But how about the time series, diurnal variations and O/C ratios of different OOA factors? The authors should not conclude this by only checking the mass profiles. Are they representative of other OOA rather than LO-OOA and MO-OOA? For example, as the authors have been emphasizing the significance of aqueous-phase processing, is there any single OOA factor related to aqueous phase chemistry? Please provide these details either in the manuscript or SI.

[Response] Thanks for your comments. We have added the details of the PMF analysis in the revised manuscript.

As shown in Fig. S4 in the supplementary material, in the 5-factor solution in spring, factor 1, factor 2 and factor 4 had similar mass spectra, time series and O/C ratios (0.87-0.96). It was unclear if the three OOA components represent distinct sources or chemical types. Therefore, it was over split by one OOA factor. Another OOA factor (factor 5) had different mass spectra and lower O/C ratio, suggesting different formation mechanism.

Previous studies showed aqueous-phase processing plays an important role in formation of nitrogen-containing compounds. $C_2O_2^+$, $C_2H_2O_2^+$ are typical fragment ions of methylglyoxal and glyoxal, that are precursors of SOA via cloud processing. $CH_2SO_2^+$, $CH_3SO_2^+$, and $CH_3SO^+$ are three typical fragment ions of methane sulfonic acid, which are products mainly from the oxidation of dimethyl sulfide and can be strongly enhanced by aqueous-phase processing (Xu et al., 2017a;Xu et al., 2017b;Xu et al., 2019). In this study, factor 2 and factor 4 were poorly correlated with aqueous-processing related fragment ions (e.g., $C_2O_2^+$, $C_2H_2O_2^+$, $CH_2SO_2^+$, $CH_3SO_2^+$, and $CH_3SO^+$; $R^2$=0.38-0.48). Factor 1 and Factor 5 were poorly correlated with N-containing ions (e.g., $CH_4N^+$, $C_2H_6N^+$, and $C_3H_8N^+$, $R^2$=0.42-0.53). Therefore, it was lack of sufficient evidence to identify a single OOA factor related to aqueous phase chemistry.

Because of the over split of the OOA factors in the 5-factor solution, we constrained the FFOA profiles separated by the 5-factor solution of PMF analysis in spring. As a result, three OA factors, including FFOA, LO-OOA and MO-OOA, were identified with ME-2 analysis in spring.

[Figure]

**Figure S4.** The mass spectra, time series, and diurnal variations of 5-factor solution of PMF analysis for the spring observation.

(2) To perform ME-2 analysis, the authors constrain the FFOA profiles with the POA factor resolved in the five-factor solution of PMF analysis in spring and apply it to all seasons. One concern here is that how do the authors believe the POA factor from the five-factor solution is good enough to represent the primary sources? How does the profile look like when performing PMF analysis to six or seven factors? Does the POA factor in this study comparable to those resolved from AMS studies previously, especially those in NCP area?

[Response] Thanks for your comments. The mass profile of the POA from the 5-factor solution was compared with those resolved from AMS studies previously in Xi'an and Beijing according to the method in Dou's et al (2009) research:

$\cos(\theta) = (\mathbf{MS_A MS_B}) / (|\mathbf{MS_A}||\mathbf{MS_B}|)$

where $\mathbf{MS_A}$ and $\mathbf{MS_B}$ are the two AMS mass spectra. The correlation coefficient R is equal to the cosine of the angle $\theta$ that we will use in this work for mass spectra comparison (Dou et al., 2009).

Table 1. Comparation of the angle $\theta$ between the mass profile of the POA factors resolved in the 5 and 6-factor solutions

in spring and HOA factor, CCOA factor, BBOA factor which resolved in Beijing (Hu et al., 2017) and Xi'an (Elser et al., 2016). The correlation coefficients between POA factors and NOx and chloride were also in the table.

| | 5-factor solution | 6-factor solution | 6-factor solution |
|---|---|---|---|
| | Factor 3 | Factor 1 | Factor 4 |
| Angle θ (degree) | | | |
| HOA in Beijing | 23 | 29 | 30 |
| CCOA in Beijing | 17 | 16 | 40 |
| BBOA in Beijing | 32 | 41 | 28 |
| HOA in Xi'an | 24 | 15 | 47 |
| CCOA in Xi'an | 15 | 26 | 28 |
| BBOA in Xi'an | 28 | 37 | 28 |
| Correlation coefficient | | | |
| NOx | 0.58 | 0.43 | 0.42 |
| Cl- | 0.78 | 0.65 | 0.70 |

As shown in Table 1, the mass profile of POA factor resolved in the 5-factor solution had some similarity with mass profile of HOA and CCOA, but not exactly the same as that of HOA or CCOA. In addition, the correlation coefficient between POA and NOx was 0.58, and that between POA and chloride was 0.78, suggesting a significant contribution of coal combustion and traffic-related sources to the POA factor in Xinglong. Previous studies found that HOA and CCOA showed remarkably similar mass spectrum patterns when m/z is below 120 (Sun et al., 2016; Sun et al., 2018), which is sometimes difficult to be separated by PMF analysis, so the POA factor resolved in this study can be considered as a combined factor of HOA and CCOA. Therefore, the POA factor was identified as fossil fuel OA (FFOA).

The POA factor from the 5-factor solution in spring was compared with the POA factor from 6- and 7-factor solutions of PMF analysis. In the 6- or 7-factor solutions (Fig. S5-S6), two POA factors appeared, but none of them was good enough to represent the primary sources in Xinglong. Take 6-factor solution as an example, as shown in table 1, the mass profile of factor 4 resolved in the 6-factor solution was different with that of HOA or CCOA. Although the mass profile of factor 1 resolved in the 6-factor solution had some similarity with that of HOA and CCOA, the mass fraction of factor

1 in total OA was only 5%, which is in the uncertainty range of PMF analysis. Meanwhile, the correlation coefficients between the factor 4 and NOx and chloride were lower than those of the POA factor resolved in the 5-factor solution and NOx and chloride. Therefore, the POA factor resolved in the 5-factor solution was good enough to represent the primary sources in Xinglong.

In conclusion, the POA factor resolved from the 5-factor solution in spring was compared with the POA factors from 6- and 7-factor solutions of PMF analysis. Results showed that it was good enough for the POA factor to represent the primary source. The mass profile of the POA factor was also compared with primary sources resolved from previous studies in Beijing and Xi'an. Results showed that the mass profile of the POA factor had some similarity with that of HOA and CCOA. Therefore, it was identified as FFOA.

[Figure]

**Figure S5.** The mass spectra, time series, and diurnal variations of 6-factor solution of PMF analysis for the spring observation.

[Figure]

**Figure S6.** The mass spectra, time series, and diurnal variations of 7-factor solution of PMF analysis for the spring observation.

Another big concern is that how robust it is to apply this factor from spring to all the other seasons. To check this, the author should perform PET PMF analysis for the other season and see if a POA factor will appear when go to more factors, and then check if the mass profile of POA are comparable in all seasons. As shown in Fig. 5, the mass contribution of FFOA in total OA is 5%, which is in the uncertainty range of PMF analysis. The authors should prove if it is reasonable to manually constrain the ME-2 to separate a single FFOA factor in summer.

[Response] Thanks for your comments. Actually, we performed PMF analysis first in each season in the previous manuscript and found that the profile of the POA resolved in spring, autumn and winter were similar. The POA factor appeared until in the 5-factor solution in spring and in the 7-factor solution in autumn and the OOA factors were over spit. Therefore, ME-2 analysis was also used in the previous manuscript by constrained the POA profile resolved by the 5-factor solution of PMF analysis in spring during these three seasons. In comparison, no POA factor appeared in summer in the 2- to 9-factor solutions of the PMF analysis, suggesting the contribution of POA might be too small to be identified by PMF in summer. Therefore, to separate POA from OOA in summer for comparison with POA in other seasons, ME-2 was used in the previous MS by constrained the POA factor resolved in spring. However, results showed that the mass fraction of POA in total OA was only 5%, suggesting the POA might be compulsorily resolved by ME-2 instead of the real POA in summer. Instead, only two OOA components (LO-OOA and MO-OOA) were identified by PMF analysis in

summer. This situation with no POA factor identified was also reported at a remote site in a boreal forest in Finland, a rural site in the southeastern USA and a background site in South China, (Raatikainen et al., 2010; Xu et al., 2015a; Zhu et al., 2016). The two OOA factors resolved by PMF analysis were similar with those resolved by ME-2 analysis. Therefore, the PMF analysis results of two OA factors (LO-OOA and MO-OOA) were used in the revised manuscript in summer.

In autumn, the solution of the PMF analysis was similar with that in spring. A POA factor appeared until the 7-factor solution was selected. The angle θ between the mass spectra of the POA factor resolved in the 7-factor solution and HOA factor, CCOA factor, BBOA factor which resolved in Beijing were 26, 15, and 34 degrees, respectively (Hu et al., 2017). The angle θ between the mass spectra of the POA factor resolved in the 7-factor solution and HOA factor, CCOA factor, BBOA factor which resolved in Xi'an was 22, 19, and 30 degrees, respectively (Elser et al., 2016). These characteristics indicated the POA factor had some similarity with HOA and CCOA. Meanwhile, the POA factor corelated well with NOx ($R_2$=0.61). Therefore, the POA factor resolved in the 7-factor solution was good enough to represent the primary sources in autumn. In addition, the mass profile of the POA in autumn was comparable with that resolved in spring (the angle θ between the two mass spectra is 9 degree), suggesting our previous choice that applied the constrained POA factor resolved in spring for the autumn data are reasonable. However, considering that the proportion of vehicle sources and coal sources in POA varied obviously in different seasons, the POA factor resolved in the 7-factor solution in autumn, not that resolved in spring, was constrained by ME-2 analysis to further improve the accuracy of the ME-2 results.

In winter, in the 4- to 7- factor solutions, OOA was over split and the mass profile of the POA was not similar with that of HOA and CCOA. In the 3-factor solution, two OOA factors (LO-OOA and MO-OOA) were identified because of the different mass spectrum, time series and the O/C ratios. The correlation coefficient between the POA factor and NOx was 0.73, and that between POA and chloride was 0.61. Meanwhile, the mass profile of the POA factor showed similarity with that of HOA and CCOA. Therefore, the POA factor was identified to FFOA and the 3-factor solution from PMF analysis was good enough in winter. The mass profile of the POA resolved in the 3-factor solution by PMF analysis in winter was comparable with that resolved in spring (the angle θ between the two mass spectra is 10 degree) and the results of the two methods were similar, suggesting our previous choice that applied the constrained POA factor resolved in spring for the wither data are reasonable. However, considering that the proportion of vehicle sources and coal sources in FFOA varied obviously in different seasons, the 3-factor solution of PMF analysis, not the application of using

constrained POA factor resolved in spring, was used in the revised manuscript. Please see detailed information on how to select the optimum PMF solution in each season in FigureS18-S20 and Table S2 in the supplementary material:

**Determination of the PMF and ME-2 solution**

**Spring:**

Factor number from 1 to 8 were selected to run in the PMF model. For the spring observation, there was no POA factor appeared in the 2- to 4- factor solution. A POA factor appeared in the 5-factor solution and diagnostic plots of the PMF analysis were shown in Fig. S2. The mass profile of the POA factor had some similarities with that of HOA and CCOA. The correlation coefficient between POA and NOx was 0.58, and that between POA and chloride was 0.78, suggesting a significant contribution of coal combustion and traffic-related sources to the POA factor in Xinglong. Previous studies found that HOA and CCOA showed remarkably similar mass spectrum patterns when m/z was below 120 (Sun et al., 2016; Sun et al., 2018), which was sometimes difficult to be separated by PMF analysis, so FFOA could be considered as a combined factor of HOA and CCOA (Sun et al., 2018). Therefore, the POA factor was identified as FFOA.

As shown in Fig. S4, in the 5-factor solution, factor 1, factor 2 and factor 4 had similar mass spectra, time series and O/C ratios (0.87-0.96). It was unclear if the three OOA components represent distinct sources or chemical types. Therefore, it was over split by one OOA factor. Another OOA factor (factor 5) had different mass spectra and lower O/C ratio, suggesting different formation mechanism. Therefore, we constrained the FFOA profiles separated by the 5-factor solution of PMF analysis in spring. As a result, three OA factors, including FFOA, LO-OOA and MO-OOA, were identified with ME-2 analysis in spring. The mass spectra, time series, and diurnal variations of ME-2 result were shown in Fig. S7.

[Figure]

Figure S2. Diagnostic plots of the PMF analysis on OA mass spectral matrix for the spring observation.

[Figure]

**Figure S3.** The mass spectra, time series, and diurnal variations of 4-factor solution of PMF analysis for the spring observation.

[Figure]

**Figure S4.** The mass spectra, time series, and diurnal variations of 5-factor solution of PMF analysis for the spring observation.

[Figure]

**Figure S5.** The mass spectra, time series, and diurnal variations of 6-factor solution of PMF analysis for the spring observation.

[Figure]

**Figure S6.** The mass spectra, time series, and diurnal variations of 7-factor solution of PMF analysis for the spring observation.

[Figure]

**Figure S7.** The mass spectra, time series, and diurnal variations of ME-2 analysis for the spring observation.

**Summer:**

For the summer observation, the 2-factor, fpeak=0 solution was selected as the optimum solution. The diagnostic plots of the PMF analysis were shown in Fig. S8. The two OA factors are more oxidized (MO-OOA) and less oxidized OOA (LO-OOA). The mass spectrum, time series and diurnal variations of OA factors were different. The O/C of the two factors were 0.58 and 0.93, respectively. No POA factor appeared in the 2- to 9-factor solutions and OOA was over spilt

in the 3- to 9-factor solutions. The detailed information on how to select the optimum PMF solution can be found in Figure S9-S12 and Table S1.

[Figure]

Figure S8. Diagnostic plots of the PMF analysis on OA mass spectral matrix for the summer observation.

[Figure]

**Figure S9.** The mass spectra, time series, and diurnal variations of 2-factor solution of PMF analysis for the summer observation.

[Figure]

**Figure S10.** The mass spectra, time series, and diurnal variations of 3-factor solution of PMF analysis for the summer observation.

[Figure]

**Figure S11.** The mass spectra, time series, and diurnal variations of 4-factor solution of PMF analysis for the summer observation.

[Figure]

**Figure S12.** The mass spectra, time series, and diurnal variations of 5-factor solution of PMF analysis for the summer observation.

**Table S1** Descriptions of PMF solutions for the summer observation in Beijing.

| Factor number | Fpeak | Q/Qexp | Solution Description |
|---|---|---|---|
| 1 | 0 | 3.83 | Too few factors, large residuals at time periods and key m/z's |
| **2** | **0** | **3.32** | **Optimum solution for the PMF analysis (MO-OOA and LO-OOA). The mass spectrum, time series and diurnal variations of OA factors were different. The O/C of the two factors were 0.58 and 0.93, respectively.** |
| 3-9 | 0 | 3.11 | Factor split. Take 3 factor number solution as an example, factor 2 was similar to the factor 2 which resolved in the 2-factor solution with similar mass spectrum, time series, diurnal variation and O/C ratios. Factor 1 and factor 3 were likely over split with similar time series and different mass spectrum. However, it was difficult to explain if they represent distinct sources or chemical types. |

**Autumn:**

The solution of the PMF analysis for the autumn observation was similar with that in spring. A POA factor appeared until the 7-factor solution and OOA was over-split. The POA factor was also identified as FFOA. The diagnostic plots of the PMF analysis were shown in Fig. S13. We constrained the POA profile separated by the 7-factor solution of PMF analysis by ME-2 analysis. As a result, three OA factors, including FFOA, LO-OOA and MO-OOA, were identified with ME-2

analysis in autumn. The mass spectra, time series, and diurnal variations of ME-2 result were shown in Fig. S16.

[Figure]

Figure S13. Diagnostic plots of the PMF analysis on OA mass spectral matrix for the autumn observation.

[Figure]

**Figure S14.** The mass spectra, time series, and diurnal variations of 6-factor solution of PMF analysis for the autumn observation.

[Figure]

**Figure S15.** The mass spectra, time series, and diurnal variations of 7-factor solution of PMF analysis for the autumn observation.

[Figure]

**Figure S16.** The mass spectra, time series, and diurnal variations of ME-2 analysis for the autumn observation.

**Winter:**

For the winter observation, the 3-factor, fpeak=0 solution was selected as the optimum solution. When OA was separated into four factors, OOA was also split into three factors (Fig. S19). In the 4-factor solution, factor 2 was similar to factor 2 which was resolved in the 3-factor solution with similar mass spectra, time series, diurnal variation and O/C ratios. However, factor 3 and factor 4 in the four-factor solution had similar O/C ratios, time series and diurnal variation. It was

unclear if the two OOA components represent distinct sources or chemical types. When more than 5 factors, OOA decomposed into three or more factors. Thus, two OOA factors were combined into total OOA for further analysis. The correlation coefficient between the POA factor resolved in the 3-factor solution and NOx was 0.73, and that between POA and chloride was 0.61. Meanwhile, the mass profile of the POA factor showed similarities with that of HOA and CCOA. Therefore, the POA factor was identified as FFOA and the 3-factor solution (FFOA, LO-OOA and MO-OOA) from PMF analysis was good enough in winter. The detailed information on how to select the optimum PMF solution can be found in FigureS18-S20 and Table S2.

[Figure]

Figure S17. Diagnostic plots of the PMF analysis on OA mass spectral matrix for the winter observation.

[Figure]

**Figure S18.** The mass spectra, time series, and diurnal variations of 3-factor solution of PMF analysis for the winter observation.

[Figure]

**Figure S19.** The mass spectra, time series, and diurnal variations of 4-factor solution of PMF analysis for the winter observation.

[Figure]

**Figure S20.** The mass spectra, time series, and diurnal variations of 5-factor solution of PMF analysis for the winter observation.

**Table S2** Descriptions of PMF solutions for the winter observation in Beijing.

| Factor number | Fpeak | Q/Qexp | Solution Description |
|---|---|---|---|
| 1 | 0 | 2.65 | Too few factors, large residuals at time periods and key m/z's. |
| 2 | 0 | 2.34 | Too few factors, POA was mixed with OOA. |
| **3** | **0** | **2.12** | **Optimum solution for the PMF analysis (FFOA, MO-OOA and LO-OOA). The mass spectrum and time series of the two OOA factors were different. The O/C of the two factors were 0.49 and 0.83, respectively. Thus, two OOA factors were for further analysis. The correlation coefficient between POA and NOx was 0.73, and that between POA and chloride was 0.61. Meanwhile, the mass profile of the POA factor had similarity with that of HOA and CCOA. Therefore, the POA factor was identified to FFOA.** |
| 4-7 | 0 | 1.89-1.98 | Factor split. Take 4 factor number solution as an example, factor 2 was similar to the factor 2 which resolved in the 2-factor solution with similar mass spectrum, time series, diurnal variation and O/C ratios. Factor 1 and factor 3 were likely over split with similar time series and different mass spectrum. However, it was difficult to explain if they represent distinct sources or chemical types. |

(3) Details about the evaluation of PMF and ME-2 results are lacking. To interpret PMF/ME-2 results, the authors should

carefully follow the procedures proposed by Ulbrich et al. (2009) and Zhang et al. (2011). For example, evaluation of Q/Qexp as a function of factor number and Q/Qexp or scaled residual as a function of m/z should be provided in the manuscript or SI.

[Response] Thanks for your suggestion. We added more details about the evaluation of PMF and ME-2 results in the revised supplementary materials, as shown above.

3. It is questionable that the authors solely based on the ratio of measured NH4+ to predicted NH4+ to evaluate aerosol acidity. While this ratio has been used as an indicator of aerosol acidity by some previous AMS studies, it has been proved recently that this ion balance method is not reliable to evaluate particle acidity (Guo et al., 2015; Song et al., 2018). Thermodynamic models, i.e., E-AIM and ISORROPIA, should be used. With thermodynamic models, Guo et al. (2017) found that aerosol is always acidic in NPC region, which is contradictory to results from this study.

[Response] Thanks for your suggestion. We agree with you that the ion balance method is not reliable to evaluate particle acidity and we have reevaluated the particle acidity by the thermodynamic model. Liu et al. (2017) and Song et al. (2019) found that pH values in ISORROPIA were on average 0.3-0.4 units higher than that in E-AIM under winter haze conditions. In addition, the ISORROPIA model is more widely used and has more results to compare, so that it was used in the revised manuscript. The forward mode was used with just aerosol-phase data and $NH_3$ data input in this study, to avoid measurement error (Song et al., 2018;Guo et al., 2017). An ammonia ($NH_3$) analyzer ($NH_3$-$H_2O$, Model 911-0016, LGR) was used to simultaneously measure $NH_3$. Notably, data for RH < 30% and RH > 95% were excluded because of the large uncertainty in LWC and pH values (Ding et al., 2019; Guo et al., 2015). Results showed that the aerosol in Xinglong were acidity in summer (PH: 2.7 ± 0.6) and moderate acidity in spring (4.2 ± 0.7), autumn (3.5 ± 0.5) and winter (3.7 ± 0.6), consistent with previous studies that although $NH_3$ in the NCP was abundant, the aerosol was far from neutral (Ding et al., 2019). Please see detailed analysis as follows:

[revised manuscript text omitted]

4. Many interpretations or conclusions from this study cannot be well supported by the observations. For example, the authors conclude in the end of the abstract that the neutralized state of submicron particles highlight the significance of NOx and ammonia reduction (also in conclusion part, Line 486 and Line 510). But these two do not have causal relationship.

[Response] Thanks for pointing this out. We now admit that the mentioned conclusion in the abstracts is not accurate, and we revised this sentence as showed below:

"Our results illustrate that the background particles in the NCP are influenced significantly by aging processes and regional transport, and the increased contribution of aerosol nitrate highlights how regional reductions in emissions of nitrogen oxide are critical for remedying occurrence of nitrate-dominated haze events over the NCP."

Line 215, the authors should not make a conclusion that the high NOx concentration is solely due to strong influence of traffic.

[Response] Thanks for pointing this out. We agree that the high NOx concentration in winter would not be mainly contributed from the traffic emissions. NOx exhibited its highest concentration in winter and correlated well with chloride ($R^2$ = 0.6), suggesting a strong influence of fossil fuel combustion, such as coal combustion. Therefore, the high NOx concentration in winter was mainly due to the coal combustion, while regional transport from heavily polluted regions may also contribute partly.

Line 280, the authors conclude that POA factor was closely related to traffic emission without evaluating the characteristic of POA. While POA showed a good correlation with NOx, does its diurnal profile show morning and evening peaks, which is a typical feature of traffic-related pollutant? It is not correct to conclude that the high concentration of FFOA at night indicate the high primary emissions. The variations in boundary layer height play a significant role. Similar issues happened again in Sect. 3.3. The authors draw conclusions regarding species formation mechanisms during daytime and nighttime based on the variations in their concentrations without considering the major influence of boundary layer height.

[Response] Thanks for your suggestions. The mass profile of the POA factor had some similarities with that of HOA and CCOA (Hu et al., 2017;Elser et al., 2016), which was detailed compared in the response to the second major question. The correlation coefficient between POA and NOx was 0.58, and that between POA and chloride was 0.78, in spring, suggesting a significant contribution of coal combustion and traffic-related sources to the POA factor in Xinglong. Therefore, the POA factor in spring could be considered as FFOA. The concentrations of FFOA were obviously higher at night than during the daytime in each season, which was mainly due to the variations in PBL. The diurnal profile of

FFOA showed evening peak in autumn and winter, while showed no morning peaks. In spring, the diurnal profile of FFOA showed no morning and evening peaks. This diurnal profile of FFOA may be related to the observation position of Xinglong Station, which is located on a mountain with an altitude of 960 m, surrounded by forests and farmland, and more than 100 kilometers away from the urban site. The distance from urban area to the site and the altitude of the site might allow time for substantial vertical mixing and dilution. Therefore, the traffic emissions of diurnal profile of FFOA showed no obvious morning and evening peaks like Beijing and other urban areas. Similar diurnal profile of HOA was also observed in in suburbs site located in the downwind of the urban site (de Sá et al., 2018).

According to your suggestion, we considered the major influence of the variations in boundary layer height on the diurnal variation of $PM_1$ species. The concentrations of FFOA were higher at night than during the daytime during the four seasons, which was mainly due to the variations in the PBL. The lower PBL at night suppressed the diffusion of pollutants. Meanwhile, higher primary emissions, such as coal-burning emissions, at night than during the daytime, also partly contributed. Nitrate exhibited drastic diurnal variation in each season, with a high concentration at night and low concentration during the daytime. This behavior was closely related to the variation of the PBL, which reduced the concentration of nitrate during the daytime and suppressed the diffusion of nitrate at night. MO-OOA exhibited similar drastic diurnal profiles in spring, autumn and winter, peaking in the afternoon and at night. The high concentration of MO-OOA at night was likely due to the co-effect of the low PBL and aqueous chemistry under high RH conditions at night (Hu et al., 2017; Sun et al., 2018). Although the PBL expanded, the concentrations of MO-OOA increased significantly from 12:00 to 18:00, indicating an important role played by photochemical processes in MO-OOA production during the daytime in these three seasons. Please see detailed analysis in line 334 to 370 in the revised manuscript.

Line 323, why the authors think the increase in nitrate concentration is caused by regional transport? The observations cannot support this. In the end of this paragraph, the authors emphasize the strong effects of local chemical production and regional transport on nitrate diurnal pattern. But the reviewer did not find any related discussion about the relative contribution of local production and regional transport. Again, Line 336, why the increased concentration from noon to evening indicate the regional characteristics of MO-OOA? Overall, these conclusions without detailed interpretation and well supported observations would confuse the audience a lot.

[Response] Thanks for pointing this out. We now admit that the increase in nitrate concentration cannot be solely attributed to regional transport. As shown in Fig. 5, nitrate exhibited drastic diurnal variation in each season, with a high concentration at night and low concentration during the daytime. This behavior was closely related to the variation of the PBL, which reduced the concentration of nitrate during the daytime and suppressed the diffusion of nitrate at night. Heterogeneous/aqueous-phase reactions and gas-to-particle condensation processes are the main pathways to forming fine-mode nitrate (Sun et al., 2018; Hu et al., 2017). The high concentration of nitrate in each season suggested the pathway of the hydrolysis of dinitrogen pentoxide ($N_2O_5$) to nitrate formation at night in Xinglong might be strong due to low NO concentrations and high $O_3$ concentrations, even at night. The NO concentrations at Xinglong Station in the four seasons were as low as 0.2 to 0.7 ppb (Table 1). Because of the low concentration of NO, it would be difficult for NO to react with $O_3$ and thus deplete $O_3$ so that $O_3$ could accumulate, even at night. $O_3$ concentrations at night in the four seasons were about 45, 70, 35 and 25 ppb, respectively, which showed that the background atmosphere exhibited high atmospheric oxidation capacity, even at night, especially in summer. The diurnal variation of nitrate showed an obvious increase from noon through the afternoon in each season, suggesting the increased nitrate concentrations were influenced by photochemical production. Nitrate exhibited its lowest concentration in summer, which can be attributed to the evaporation of $NH_4NO_3$ due to the high temperatures (Fig. 5). Therefore, the nitrate diurnal pattern might be influenced by the variation of PBL and local chemical production, including the hydrolysis of $N_2O_5$ to nitrate formation at night and photochemical processes during the daytime.

Similar with nitrate, MO-OOA also exhibited drastic diurnal profiles in spring, autumn and winter, peaking in the afternoon and at night. The high concentration of MO-OOA at night was likely due to the co-effect of the low PBL and aqueous chemistry under high RH conditions at night (Hu et al., 2017; Sun et al., 2018). Although the PBL expanded in the afternoon, the concentrations of MO-OOA increased significantly from 12:00 to 18:00, indicating an important role played by photochemical processes in MO-OOA production during the daytime in these three seasons. In summer, the concentration of MO-OOA showed its greatest increase rate (0.18 $\mu g \cdot m^{-3} \cdot h^{-1}$) from 09:00 to 18:00, implying stronger photochemical production of MO-OOA than in other seasons. Note that the wind speed in summer also increased rapidly from 09:00 to 16:00, along with the increase of MO-OOA, which may suggest regional transport also partly contributed to the rapid increase in MO-OOA during the daytime in summer. The related discussion was added in the revised manuscript.

"MO-OOA exhibited similar drastic diurnal profiles in spring, autumn and winter, peaking in the afternoon and at night. The high concentration of MO-OOA at night was likely due to the co-effect of the low PBL and aqueous chemistry under high RH conditions at night (Hu et al., 2017; Sun et al., 2018). Although the PBL expanded, the concentrations of MO-OOA increased significantly from 12:00 to 18:00, indicating an important role played by photochemical processes in MO-OOA production during the daytime in these three seasons. In summer, the concentration of MO-OOA showed its greatest increase rate (0.18 μg·m$^{-3}$·h$^{-1}$) from 09:00 to 18:00, implying stronger photochemical production of MO-OOA than in other seasons. Note that the wind speed in summer also increased rapidly from 09:00 to 16:00, along with the increase of MO-OOA, which may suggest regional transport also partly contributed to the rapid increase in MO-OOA during the daytime in summer. The diurnal profiles of LO-OOA in each season were flatter than those of MO-OOA. The decreased PBL mainly resulted in a higher concentration of LO-OOA at night. The increased concentration of LO-OOA from noon through the afternoon was mainly due to photochemical processes. The highest concentration of LO-OOA in summer suggested a stronger photochemical production of LO-OOA than in other seasons."

Line 426, from the observations, the authors can only conclude that aqueous-phase and photochemical processing both play roles. How do they evaluate which is more important?

[Response] Thanks for pointing this out. Yes, we agree that it is hard to say which is more important but can only conclude that aqueous-phase and photochemical processing both play roles in spring, autumn and winter, as the two OOA factors increased with the elevation of both RH and Ox in these three seasons (Fig.7 and 8). Whereas, as showed in Fig. 7 and Fig.8, the mass fraction of MO-OOA in OA did not increase as Ox elevated in winter, while it increased from 30% to 40% as RH increased from 30 to 90%. This characteristic suggested a more important role of aqueous-phase processing on SOA formation than photochemical processing in winter. The related discussion was added in the revised manuscript.

"Note that the mass fraction of MO-OOA did not increase as Ox elevated in winter, while it increased ~from 30% to 40% as RH increased from 30 to 90%. This characteristic suggested a more dominant important role of aqueous-phase processing on SOA formation than photochemical processing in this season."

Line 432, how do the authors conclude that photochemical processing enhanced during regional transport without any observation or interpretation regarding this (Also in the conclusion part, Line 499)?

[Response] Thanks for your reminding, and we now admit that it cannot conclude that photochemical processing enhanced during regional transport based on the present results. In fact, both LO-OOA and MO-OOA showed overall increasing trends as Ox increased in summer. In addition, increases of LO-OOA and MO-OOA as functions of Ox were clearly associated with the increases of wind speed, which was more significant in summer than in other seasons. In comparison, in urban areas, such as Beijing, increase of LO-OOA were associated with the decreases of wind speed, that facilitated the accumulation of air pollutants (Xu et al., 2017). Such a difference between urban and background areas may be due to the influence of regional transport on the Ox and SOA concentrations in background areas. We revised this part and the related discussion was provided as below:

"In summer, both LO-OOA and MO-OOA showed overall increasing trends as Ox increased, while RH showed a corresponding overall decreasing trend. This behavior indicates a strong influence of photochemical processing on both LO-OOA and MO-OOA production. Meanwhile, LO-OOA showed a continuously decreasing trend as RH increased in summer, except for a slightly increasing trend when RH increased from 40 to 60%, indicating photochemical processing dominated LO-OOA formation. MO-OOA increased significantly with Ox, while it increased slightly with RH (40% < RH < 60%) firstly, and then decreased with RH when RH was above 60%. This characteristic suggested photochemical processing dominated MO-OOA formation, but the role of aqueous-phase processing under moderate RH (40% < RH < 60%) conditions cannot be ruled out in summer. In urban Beijing, the impact of photochemical processing on LO-OOA production was significant, while on MO-OOA production it was limited in summer (Xu et al., 2017; Duan et al., 2020), mainly due to the higher atmospheric oxidation capability in the background atmosphere than in the urban atmosphere in summer. Furthermore, increases of LO-OOA and MO-OOA as functions of Ox were clearly associated with the increases of wind speed, which was more significant in summer than in other seasons. In comparison, in urban areas, such as Beijing, increase of LO-OOA were associated with the decreases of wind speed, that facilitated the accumulation of air pollutants (Xu et al., 2017). Such a difference between urban and background areas may be due to the influence of regional transport on the Ox and SOA concentrations in background areas. The continuous increases in SOA concentrations associated with the increases of wind speed in the background atmosphere may imply important role of regional transport in SOA formation in summer."

Line 434, the authors said that the impact of photochemical processing on MO-OOA production was limited in summer. Then what is the major formation mechanisms of MO-OOA in summer? Do they provide any evidence regarding different formation mechanism of LO-OOA and MO-OOA from this study?

[Response] Sorry for the misunderstanding. The line 434 in previous manuscript referred to the impact of processing on MO-OOA production was limited in summer in the urban site in Beijing. According to Xu's et al (2017) research, MO-OOA had no obvious changes as Ox increased, while increased continuously as RH increased, indicating aqueous-phase processing dominant MO-OOA formation in summer. In comparison, in background Xinglong, MO-OOA showed overall increasing trends as Ox increased, while RH showed a corresponding overall decreasing trend. This behavior indicates the strong influence of photochemical processing on MO-OOA production. Meanwhile, MO-OOA increased slightly with RH (40%<RH<60%) firstly and then decreased with RH when RH was above 60%. This characteristic suggested the photochemical processing dominant MO-OOA formation, but the role of aqueous-phase processing under moderate RH (40%<RH<60%) condition cannot be ruled out in summer.

The differences of formation mechanisms of LO-OOA and MO-OOA in this study is not as obvious as those in urban areas, which might be due to the relative aged and regionally dispersed fine aerosols received at this mountain site compared with those from ground-based measurement in urban areas. There are still some differences existed. For example, in summer, MO-OOA increased significantly with Ox while increased slightly with RH (40%<RH<60%) firstly and then decreased with RH when RH was above 60%. This characteristic suggested the photochemical processing dominant MO-OOA formation, but the role of aqueous-phase processing under moderate RH (40%<RH<60%) condition cannot be ruled out in summer. In comparison, LO-OOA increased significantly with Ox while decreased continuously with RH, indicating LO-OOA formation was only affected by photochemical processing.

In autumn and winter, MO-OOA increased more rapidly than those of LO-OOA as RH increased. As a result, the mass fractions of MO-OOA increased by 25% in autumn and by 12% in winter when RH increased from 20 to 90%. Corresponding, the mass fraction of LO-OOA decreased by 15% in autumn. These characteristics indicated that aqueous-phase processing plays more important role in MO-OOA formation than that in LO-OOA in these two seasons. Detailed discussion can be seen in Line 405 to 417.

Line 505, the longer transport distance of air masses in summer does not mean that the influence of regional transport is strongest in summer. Not proper quantification.

[Response] Thanks for your reminding. We admit that the previous conclusion is not proper and we deleted it, and rewrite as follows:

"The air masses from the southern regions (clusters 2 and 3) accounted for 56% of all the air masses in summer, which was obviously higher than the percentage in other seasons (27–38%). Cluster 3 in summer started at Bohai Bay and passed through the Shandong Peninsula and over Bohai Bay. The $PM_1$ concentrations for clusters 2 (14.7 $\mu g\ m^{-3}$) and 3 (12.2 $\mu g\ m^{-3}$) were both high. These results suggest a dominant role played by southern transport in submicron aerosol concentrations over the NCP in summer."

**Specific comments:**

(1) In the introduction, Line 42, it's not proper to summarize that sulfate dominated in the south of NPC and nitrate dominate in the north of NCP. It is not determined by the location, but more by the emission characteristics.

[Response] Thanks for your reminding. This sentence was revised to "sulfate dominates the secondary inorganic aerosols (SIAs) in heavy-industry cities such as Shijiazhuang and Handan, while in recent years nitrate has dominated those in Beijing because of the strict emissions reduction measures for coal combustion."

(2) Line 55, the authors said that previous studies at background NCP site are limited by the low resolution. Any new findings do they draw from their high-resolution measurements?

[Response] Thanks for your comment. The HR-ToF-AMS can provide elemental information, such as hydrogen-to-carbon (H/C), organic-mass-to-organic-carbon (OM/OC), and oxygen-to-carbon (O/C), which can help to quantify the oxidation degree of OA (Jimenez, 2003). OOA can also be separated as MO-OOA and LO-OOA due to the different O/C ratios (Zhang et al., 2011). According to the high-resolution measurements, our results suggested the dominant role of aqueous-phase processing on SOA formation than photochemical processing in winter. Aqueous-phase processing plays

more important role in MO-OOA formation than that in LO-OOA in autumn and winter. Both of photochemical and aqueous-phase processing contribute to LO-OOA and MO-OOA production in spring. In summer, the photochemical processing dominant MO-OOA formation, but the role of aqueous-phase processing under moderate RH (40%<RH<60%) condition cannot be ruled out in summer. In comparation, LO-OOA formation was only affected by photochemical processing in summer. In addition, regional transport also played an important role in the variations of SOA, especially in summer that continuous increases in SOA concentration as a function of Ox was found to be associated with the increases of wind speed.

(3) Line 105, to quantify aerosol concentrations, what are the RIE values of sulfate and ammonium according to the standard calibration?

[Response] Thanks for pointing this out. In this study, the relative ionization efficiency (RIE) values used were 1.1, 1.2, 1.3 and 1.4 for nitrate, sulfate, chloride and organics, respectively. RIE value of 4.0 were used for ammonium based on the ionization efficiency calibration results in each season. This part has been added in the revised manuscript.

(4) Since the measurement station is 960 meters above sea level, why do the authors choose the height of 500 m to permafrost back trajectories? How does it compare to 1000m or 1500m?

[Response] Thanks for your reminding and sorry for the misunderstanding. The height of 500m was not the height above the sea level, but the height above ground level (AGL). Previous study showed that the planetary boundary layer height (PBL) in Xinglong could reduce to 100-200m during haze episode (Li, et al., 2020). Thus, we choose the height of 200m (AGL) and recalculate the air mass trajectories in the revised MS. According to the suggestion of comparing to 1000m and 1500m, we found these two heights was not suitable for our observation sites. If they are refer to the height above sea level, then it was too low for the height of 1000m which was only 40m above ground level that the air mass may be blocked by mountains on the moving ways, while for the height of 1500m (refer to 540m AGL), it would sometimes above the PBL and thus could not arrive to the receptor site. If they are refer to the height above ground level, then these two heights are further above the PBL and thus could not arrive to the receptor site. To make it clear, we revised this part.

"The 48-h back trajectories were calculated every hour at a height of 200 m (above ground level) using the HYSPLIT (Hybrid Single-Particle Lagrangian Integrated Trajectories) model"

(5) Line 114, the AMS does not measure chlorine but chloride. Please revise the whole manuscript accordingly.

[Response] Thanks for your reminding. We have carefully corrected similar mistakes in the revised manuscript.

(6) Line 162, are high RH and high PM1 concentration correspond to air masses from the south?

[Response] Thanks for your comment. As shown in Fig. 9, high RH and high $PM_1$ concentration correspond to air masses from the south in all seasons. In summer, the RH and $PM_1$ concentration were also at high levels of the air massed from the north and northwest regions of Xinglong. Detailed discussion can be seen in Section 3.4.

(7) Line 170, it should be clearly noted that the frequency distribution of PM1 is shown in Fig. 1 using the white curve.

[Response] Thanks for your reminding. We have clearly noted that the frequency distribution of $PM_1$ was shown in Fig. 1 using the white curve in the revised manuscript.

(8) Line 221, how do the authors define the wind dilution ratio? Please make it clear.

[Response] Thanks for your suggestion. The wind dilution ratio was defined as the percentage decrease in the concentration of the aerosol species for every 1 m $s^{-1}$ decrease in wind speed (Sun et al., 2013). We have made it clear in the revised manuscript.

(9) In Fig. 3, how do the authors average WD? Do they follow the vector average wind direction?

[Response] Thanks for your comment. In the previous manuscript, we divided the data into 8 directions (0-45°, 45-90°, 90-135°, 135-180°, 180-225°, 225-270°, 270-315°, 315-360°) according to the wind direction and calculated the average wind direction by arithmetic average. However, it's more reasonable to divided the data according to the vector average wind direction. Therefore, we divided the data into 8 directions (N, NE, E, SE, S, SW, W, NW) in the revised manuscript and now the revised Fig.2 (Fig. 3 in previous manuscript) is as follows:

[Figure]

Figure 2. Variations in mass concentrations and mass fractions of PM$_1$ species as functions of wind speed (WS), wind direction (WD), and relative humidity (RH) during the spring (a, b, c), summer (d, e, f), autumn (g, h, i), and winter (j, k, l) observations. (OA: organic aerosol).

(10) Line 280, please provide the number of correlation coefficient of POA vs. NOx and POA vs. chloride.

[Response] Thanks for your suggestion. The correlation coefficient between POA and NOx was 0.58, and that between POA and chloride was 0.78, in spring. The correlation coefficient between POA and NOx was 0.78, and that between POA and chloride was 0.61, in winter. The correlation coefficient between POA and NOx was 0.61 in autumn. Detailed information can be seen in the supplementary materials.

(11) Line 300, how do their correlations look like? According to previous studies, LO-OOA correlates better with nitrate, while MO-OOA better with sulfate.

[Response] Thanks for your suggestion. The correlation coefficient between LO-OOA and nitrate (sulfate) was 0.79 (0.58) in autumn and 0.71 (0.44) in winter. Therefore, LO-OOA correlated well with nitrate in autumn and winter.

MO-OOA, meanwhile, correlated well with both nitrate and sulfate in autumn (MO-OOA vs. nitrate: 0.86; MO-OOA vs. sulfate: 0.87) and winter (MO-OOA vs. nitrate: 0.79; MO-OOA vs. sulfate: 0.72), similar to the findings of previous studies in wintertime in Beijing (Hu et al., 2017). In comparison, LO-OOA had a low correlation coefficient with nitrate or sulfate, and MO-OOA had a correlation coefficient of 0.65 with sulfate, in summer. The poor correlation between LO-OOA and secondary inorganic species has also been found in a previous study (Sun et al., 2018). Both of LO-OOA and MO-OOA correlated well with nitrate and sulfate in spring.

(12) In Fig. 5, the nitrate time series is missing in Fig. 5b.

[Response] Thanks for your reminding. The time series of nitrate was added in Fig. 5b

(13) Line 360, the higher O/C ratio in Xinglong should be due to the weak influence of primary emissions.

[Response] Thanks for your comment and we have corrected the sentence as follows:

"The O/C ratios in Xinglong in all seasons (0.54–0.75) were slightly higher than those in urban Beijing (0.47–0.53), mainly due to the weak influence of primary emissions."

(14) Line 390, please define the Ox.

[Response] Thanks for your reminding. Ox was defined in the revised manuscript.

(15) Figure 9, the plots as a function of Ox not RH.

[Response] Thanks for your reminding. The caption of Fig. 9 was corrected in the revised manuscript.

(16) Please define what is "dva" in the manuscript.

[Response] Thanks for your reminding. "dva" (vacuum dynamic diameter) was defined in the revised manuscript.

(17) Many grammatical mistakes. The reviewer recommends to do editing service. For example, in the title, it should be "highly time-resolved". Line 309, should be "be attributed to". Line 83, should be "a HR-ToF-AMS was deployed……with collocated measurements of meteorological parameters and gaseous species." Line 89, should change "of" to "on".

[Response] We are so sorry for making the grammatical mistakes to make trouble to your review work. Thanks for your useful comments and suggestions to improve the manuscript. We have done editing service and carefully checked and corrected the errors sentence by sentence.

[revised manuscript text omitted]

---

## Author Response (AR2)

**Response to referees' comments on "Seasonal variations of the highly time-resolved aerosol composition, sources, and chemical processes of background submicron particles in North China Plain"**

We highly appreciate the detailed valuable comments of the three referees on our manuscript. The suggestions are quite helpful for us to improve the quality of our paper. Please see the detailed point-by-point response below. We list the comments in black, our replies in blue, and the changes in revised manuscript in red.

**Anonymous Referee #1**

The authors have addressed my comments carefully. I therefore recommend publication in ACP after the technical corrections.

[Response] Thank you so much for your approval of our revised work. Please see the detailed response below.

1. Line 53-56: Rephrase this sentence to make it clearly.

[Response] Thanks for your suggestion. The sentence has been revised as follows:

"Early studies found that secondary aerosols dominate the aerosol particles at background sites and regional transport affects the air pollution of the background atmosphere."

2. Line 116: The time of MS and PToF modes are not clear in the text.

[Response] Thanks for your reminding. The time of MS and PToF modes has been added in the revised manuscript:

"Under V-mode operation, the AMS cycle through the mass spectrum (MS) mode and the particle time-of-flight (PToF) mode every 30s, spending 10s and 20s, respectively."

3. It is better to show the correlation coefficients in Figure 4.

[Response] Thanks for your suggestion. The correlation coefficients have been added in Figure 4.

4. Line 356-357: What does it mean "demonstrating the regional characteristics"? The authors should explain it.

[Response] Thanks for your reminding and we have rewritten the sentence as follows:

"In comparison to the diurnal patterns of nitrate, sulfate showed flatter diurnal cycles in each season, identifying the regional characteristics of sulfate formation (Sun et al., 2016)."

5. In Section 3.5: It is interesting that relatively high PM1 concentrations were also observed with air masses form north and west regions of Xinglong in summer. The authors should explain more about the result.

[Response] Thanks for your suggestion and more explanation was added in the revised manuscript:

"The air masses from the southern regions (clusters 2 and 3) were dominant in summer and accounted for 56% of all the air masses in summer, which was obviously higher than the percentage in other seasons (27–38%). Cluster 3 in summer started at Bohai Bay and passed through the Shandong Peninsula and over Bohai Bay. The PM1 concentrations for clusters 2 (14.7  $\mu$ g m-3) and 3 (12.2  $\mu$ g m-3) were both high. These results suggest a dominant role played by southern transport in submicron aerosol concentrations over the NCP in summer. Furthermore, the transport distances of clusters from the north and west regions in summer were shorter than those in other seasons. In general, with a decrease in the transport distance of clusters from the north and west regions, particle concentrations gradually increase (Hu et al., 2017). Although the clusters from these regions in summer only accounted for 15% and 8% of all the air masses, respectively, the PM1 concentrations for the two clusters (cluster 1: 12.8  $\mu$ g m-3; cluster 5: 10.2  $\mu$ g m-3) were both at high levels and similar to those associated with the southern air masses (cluster 2: 14.7  $\mu$ g m-3 and cluster 3: 12.2  $\mu$ g m-3). All these characteristics suggest that regional transport from Inner Mongolia (west and north regions of Xinglong) also partially contributes to the particle pollution in the background area of the NCP in summer."

**Reference:**

Sun, Y., Du, W., Fu, P., Wang, Q., Li, J., Ge, X., Zhang, Q., Zhu, C., Ren, L., and Xu, W.: Primary and secondary aerosols in Beijing in winter: sources, variations and processes, Atmospheric Chemistry and Physics, 16, 8309-8329, 2016.

**Anonymous Referee #2**

While the dataset is valuable, there are still many mistakes or improper interpretations in this study.

[Response] Thank you very much for your helpful comments and we have taken all of them into account in the revised version of the manuscript. Please see the detailed response marked blue below and the changes marked red in the revised manuscript.

For example, Line 215: This reviewer does not agree that higher acidity of sulfate is due to the low volatility.

[Response] Thanks for your comments. We now admit that the higher acidity of sulfate was not directly due to the low volatility, but for leading to a much higher concentration of  $H_{air}^+$  than nitrate. To make it clear, we revised it in the manuscript as follows:

"Previous studies show that sulfate can lead to a much higher concentration of  $H_{air}^+$  than nitrate due to its low volatility (Ding et al., 2019; Tan et al., 2018; Xu et al., 2019)."

Line 245: The indication is quite confusing as the strong influence of fossil fuel combustion is due to regional transport at this background site.

[Response] Thanks for your reminding. To make it clear, we revised it as follow:

"As shown in Table 2, NOx exhibited its highest concentration in winter and correlated well with chloride ( $R^2 = 0.6$ ), suggesting a strong influence of fossil fuel combustion, such as coal combustion. The enhancement of emission in winter in the NCP leaded to the increase of NOx in background atmosphere, which was mainly due to the regional transport."

Line 268: The indication here is not correct. How previous studies in urban Beijing indicate the formation mechanisms in background atmosphere?

[Response] Thanks for your reminding. We have revised the sentence as follows:

"Previous studies in urban Beijing show a successive increase in SIA with increased RH (Li et al., 2019;Liu et al., 2016), suggesting that aqueous-phase processing affects nitrate formation in urban atmospheres."

Line 275: The authors should not draw a conclusion that the impact of regional transport weakened as the regional transport can be related to air masses both from clean and polluted areas. With the decrease in wind speed, do the authors see changes in back trajectories?

[Response] Thanks for your reminding. We agree with you that the decrease of surface wind speed does not indicate the weakening of regional transportation. When RH was above 60%, the  $PM_1$  species decreased in spring as RH increased, which was different from other seasons. This behavior may be due to the low overall humidity in spring, the number of samples with humidity above 60% was limited, and the statistical results may be not representative enough. Therefore, the sentence was rewritten as follows:

"When RH was > 60%, the SIA concentrations decreased rapidly, which would be mainly due to insufficient samples (n<30) with humidity above 60%."

Line 277: The authors should conclude the important role of photochemistry in the formation of sulfate and OOA based on the relationship between OA/sulfate and RH.

[Response] Thanks for your suggestion. The sentence was revised as follows:

"Notably, the OA and sulfate concentrations were high even when RH was low (RH < 40) in summer, which was significantly different from what occurred in other seasons, suggesting the important role of photochemistry in the formation of sulfate and OOA."

Line 293: Did the greater level of NPF in winter occur in Xinglong? It's confusing by adding this description.

[Response] Thanks for your reminding. The sentence was deleted in the revised manuscript.

Line 296-321: This part regarding the evaluation of PMF results is partly repetitive of those in the Methods.

[Response] Thanks for your reminding. We have checked carefully and deleted the repetitive part in the revised manuscript.

Line 322-325: The characteristics of f44 and f43 are indicators to name the factors as MO-OOA or LV-OOA. It's not because of these behaviors that MO-OOA has a higher oxidation state than LO-OOA.

[Response] Thanks for your reminding. We agree with you and revised this part as follow:

"The high f44 values were permanent in the mass spectrum of both LO-OOA and MO-OOA. The f44 values for MO-OOA and LO-OOA in the four seasons ranged from 16.3 to 23.5% and 8.1 to 13.8%, respectively. The f43 values for MO-OOA and LO-OOA ranged from 4.8 to 5.2% and 6.8 to 9.1%, respectively. The O/C ratios of the MO-OOA factors in the four seasons were 0.93, 0.93, 0.84 and 0.83, respectively—higher than those in the corresponding LO-OOA (0.69, 0.58, 0.67 and 0.49)."

Line 344: The diurnal of nitrate is not only related to the PBL influence but also contributions from lower temperature and higher RH at night.

[Response] Thanks for your reminding. We agree with you and revised this part as follow:

"This behavior was closely related to the variation of the PBL, temperature and RH. The lower concentration of nitrate during the daytime can be attributed to the higher PBL and the evaporation of NH4NO3 due to the higher temperatures. In comparison, the higher concentration at night was closely related to lower temperature and higher RH, which favor to nitrate formation through heterogeneous reactions."

Line 347: It's not correct to conclude the importance of N2O5 hydrolysis based on low NO concentration and high O3 concentration. Do they check NO2 concentration?

[Response] Thanks for your reminding and suggestions. We agree with you and revised this part as follow:

"High concentrations of  $O_3$  were observed during nighttime in the four seasons, which were about 45, 70, 35 and 25 ppb, for spring, summer, autumn and winter, respectively. In addition, the NO2 concentrations ranged from 4 to 14 ppb in all seasons. These results suggested the background atmosphere exhibited high atmospheric oxidation capacity, even at night, especially in summer, which indicated the hydrolysis of dinitrogen pentoxide (N2O5) would contribute significantly to nitrate formation at night under high RH conditions."

Line 362: The high sulfate concentration at night can also be contributed by lower PBL.

[Response] Thanks for your reminding. We revised it following your suggestion:

"At night, however, the high sulfate concentration might be attributable to the lower PBL and enhancement of aqueousphase processing under high temperatures and humidity"

Line 387: The high O/C ratio indicates the high aging of the aerosols.

[Response] Thanks for your reminding. We have changed the sentence in the revised manuscript:

"Mainly due to the weak influence of primary emissions, the O/C ratios in Xinglong in all seasons (0.54–0.75) were slightly higher than those in urban Beijing (0.47-0.53), suggesting the high aging of the aerosols in Xinglong."

Line 439-441: The reviewer cannot understand the meaning or the main point here.

[Response] Thanks for your reminding. We have rewritten this sentence to make it clearly:

"MO-OOA increased significantly with Ox, while it increased slightly with RH (40% < RH < 60%) firstly, and then decreased with RH when RH was above 60%. This characteristic suggested photochemical processing dominated MO-OOA formation, but the role of aqueous-phase processing under moderate RH (40% < RH < 60%) conditions cannot be ruled out in summer in the background atmosphere. In comparison, previous studies found that aqueous-phase processing dominated MO-OOA formation in urban Beijing in summer (Xu et al., 2017;Duan et al., 2020). This kind of difference in MO-OOA formation between urban and background site in the NCP would be mainly due to the higher atmospheric oxidation capability in the background atmosphere. Previous study showed that Ox concentration in the background sites was 30% higher than that in the urban site during summertime (Wang et al., 2013)."

Line 468-473: The authors said that the medium-distance cluster (1, 3, 4, and 5) were dominant in summer but then they focused on cluster 2 and 3 from the southern regions and said southern transport played a dominant role. This is contradictory.

[Response] Sorry for the misleading. Based on the distances over which the air masses were transported, the mediumdistance cluster (1, 3, 4, and 5) accounted for 57% in all the clusters and the short-distance cluster (cluster 2) accounted for 43%. Therefore, we said the medium-distance cluster (1, 3, 4, and 5) were dominant in summer in the previous manuscript. However, based on the directions of the air masses, clusters from the southern regions (cluster 2 and cluster 3) accounted for 66%, so that southern transport played a dominant role. To avoid the misleading, we have rewritten this part as follow: "The air masses from the southern regions (clusters 2 and 3) dominated in summer and accounted for 56% of all the air masses, which was obviously higher than the percentages in other seasons (27–38%). Cluster 3 in summer started at Bohai Bay and passed through the Shandong Peninsula and over Bohai Bay. The PM1 concentrations for clusters 2 (14.7  $\mu$ g m-3) and 3 (12.2  $\mu$ g m-3) were both high. These results suggest a dominant role played by southern transport in submicron aerosol concentrations over the NCP in summer. Furthermore, the transport distances of clusters from the north and west regions in summer were shorter than those in other seasons. In general, with a decrease in the transport distance of clusters from the north and west regions in summer only accounted for 15% and 8% of all the air masses, respectively, the PM1 concentrations for the two clusters (cluster 1: 12.8  $\mu$ g m-3; cluster 5: 10.2  $\mu$ g m-3) were both at high levels and similar to those associated with the southern air masses (cluster 2: 14.7  $\mu$ g m-3 and cluster 3: 12.2  $\mu$ g m-3). All these characteristics suggest that regional transport from Inner Mongolia (west and north regions of Xinglong) also partially contributes to the particle pollution in the background area of the NCP in summer."

Line 507: The reviewer cannot agree that sulfate dominates submicron particles in southern and western China generally.

[Response] Thanks for your reminding. We admit that this conclusion was obtained from a few previous studies and it is improper to represent for the southern and western China. Therefore, we deleted this sentence revised this part as follows:

"In addition, the high contribution of aerosol nitrate in background NCP highlights how regional reductions in nitrogen oxide emissions are critical for remedying the occurrence of haze events over the NCP."

**Reference:**

Ding, J., Zhao, P., Su, J., Dong, Q., Du, X., and Zhang, Y.: Aerosol pH and its driving factors in Beijing, Atmospheric Chemistry and Physics, 19, 7939-7954, 10.5194/acp-19-7939-2019, 2019.

Tan, T., Hu, M., Li, M., Guo, Q., Wu, Y., Fang, X., Gu, F., Wang, Y., and Wu, Z.: New insight into PM2.5 pollution patterns in Beijing based on one-year measurement of chemical compositions, Sci Total Environ, 621, 734-743, 10.1016/j.scitotenv.2017.11.208, 2018.

Xu, W., Sun, Y., Wang, Q., Zhao, J., Wang, J., Ge, X., Xie, C., Zhou, W., Du, W., Li, J., Fu, P., Wang, Z., Worsnop, D. R., and Coe, H.: Changes in Aerosol Chemistry From 2014 to 2016 in Winter in Beijing: Insights From High-Resolution Aerosol Mass Spectrometry, Journal of Geophysical Research: Atmospheres, 124, 1132-1147, 10.1029/2018jd029245, 2019.

**Anonymous Referee #3**

This paper presented a four-season measurement of aerosol composition using an AMS at Xinglong site and reported the seasonal variation, diurnal variation, as well as the source appointment. The authors revised the manuscript largely according to the referee's comments, especially add the calculation of aerosol acidity by the thermodynamic model of ISORROPIA II. This is a good improvement. However, similar to the comments of Referee II, this study is to date more like a data report without too much insight to improve the current understanding. I, therefore, recommend a major revision before it can be published in ACP.

**[Response] Thank you so much for your approval of our revised work.**

Although there are many highly time-resolved aerosols studies in background atmosphere, they mainly conducted in the remote areas, which represent for global background atmosphere rather than regional background atmosphere, while regional background atmosphere is more critical to reflect the general picture of anthropogenic emissions in a hot polluted region, such as the NCP region. However, chemical composition and evolution and formation mechanisms of secondary organic aerosol (SOA) in the background atmosphere in the NCP are still not fully understood. As far as we known, only two studies investigated the aerosol chemical composition and OA sources in the background site in the NCP, using the high-resolution AMS so far (Li et al., 2019;Yan et al., 2020). What's more, both of the two studies were concentrated in wintertime, and didn't investigated the oxidation state of SOA and the evolution and processes of SOA.

In our study, we present the OA variations and SOA formation mechanism in a background site in each season in the NCP. Our results showed that the secondary organic aerosols in the background area in the NCP was highly oxidized in all seasons. This suggests the high atmospheric oxidation capacity in the NCP on a regional scale. What's more, we analyzed the evolution and formation mechanisms of SOA. These results showed a clearer picture on the evolution of PM1 species and SOA at the background atmosphere, and could improve our understanding on the formation mechanism of SOA under the high atmospheric oxidizing capacity of NCP, which could provide the direct observation results to verify the models. For example, we found the dominant role of aqueous-phase processing on SOA formation in winter, while both of photochemical and aqueous-phase processing contribute to LO-OOA and MO-OOA production in spring and autumn. In summer, the photochemical processing dominant MO-OOA formation, but the role of aqueous-phase processing under moderate RH (40%<RH<60%) condition cannot be ruled out. In comparison, LO-OOA formation was mainly contributed by photochemical processing in summer.

Therefore, we believe that our study provides valuable information to the scientific community on understanding OA sources and SOA formation in the background atmosphere in the NCP.

1. I did suggest to make the analysis more focus on 1-2 scientific question, other than report the data as seasonal variation, diurnal variation et al

[Response] Thanks for your suggestion.

Actually, the compositions of PM1, the sources and oxidation state of OA and impacts of regional transport on aerosol mass loadings and chemistry changed significantly in different background sites and seasons. However, no systematic measurements with high time resolution of the mass–size distributions of chemical components in fine aerosol particles, covering four seasons, have yet been reported in the background atmosphere in the NCP, which would hinder our understanding on the seasonal variations of evolution and chemical processes of secondary aerosol and regional transport on a regional scale. Therefore, we reported the data as seasonal variation and diurnal variation to fill in the gap of characteristics of background aerosols in the NCP.

We found obvious seasonal variations in chemical species, aerosol acidity and regional transport. Specifically, nitrate dominated  $PM_1$  in spring, autumn and winter, while sulfate dominated  $PM_1$  in summer. Aerosol was acidity in summer and moderate acidity in other three seasons. Backward trajectory analysis showed that higher concentrations of submicron particles were associated with air masses transported short distances from the southern regions in all four seasons, while long-range transport from Inner Mongolia (western and northern regions) also contributed to summertime particulate pollution in the background areas of the NCP.

As we found SOA in the background atmosphere was highly oxidized in the NCP, we focused on the specific question of the formation mechanisms of SOA in the background atmosphere and found that aqueous-phase processing dominated SOA formation in winter, while both of photochemical and aqueous-phase processing contribute to LO-OOA and MO-OOA production in spring and autumn. In summer, the photochemical processing dominant MO-OOA formation, but the role of aqueous-phase processing under moderate RH (40%<RH<60%) condition cannot be ruled out. In comparison, LO-OOA formation was mainly contributed by photochemical processing in summer. These results showed a clearer picture on the evolution of PM1 species and SOA at the background atmosphere, and could improve our understanding on the formation mechanism of SOA under the high atmospheric oxidizing capacity of NCP. The related discussion had been

**added in the revised manuscript.**

2. In section 3.4: the authors drew their conclusions without enough supports. E.g. it's hard to conclude the role of aqueousphase reactions solely by the observational that MO-OOA increased with RH.

[Response] Thanks for your reminding. We agree with you and have added the discussion about the change of SOA with LWC in the revised manuscript to further prove the conclusion.

"Both LO-OOA and MO-OOA increased significantly as RH increased when RH was < -90% in autumn and winter. Meanwhile, both of LO-OOA and MO-OOA increased significantly as LWC increased when LWC was below 160 µg m-3 and 140 µg m-3 in autumn and winter, respectively (Figure S 21). These behaviors suggested aqueous-phase processing had a significant influence on OOA formation in these two seasons. MO-OOA increased more rapidly than those of LO-OOA as RH increased. As a result, the mass fractions of MO-OOA increased by 25% in autumn and by 12% in winter when RH increased from 20 to 90%. Corresponding, the mass fraction of LO-OOA decreased by 15% in autumn. These characteristics indicated that aqueous-phase processing plays more important role in MO-OOA formation than that in LO-OOA in these two seasons. Note that the mass fraction of MO-OOA did not increase as Ox elevated in winter, while it increased ~from 30% to 40% as RH increased from 30 to 90%. This characteristic suggested a more dominant important role of aqueous-phase processing on SOA formation than photochemical processing in this season."

"In spring, LO-OOA and MO-OOA only increased under moderate RH (RH < 70%) as RH increased. Notably, Ox also increased when RH was < 70% as RH increased. The LWC value in spring was far lower than those in other three seasons. LO-OOA and MO-OOA only increased when LWC was below 30  $\mu$ g m-3, and then decreased when LWC was above 30  $\mu$ g m-3. This characteristic suggested the weaker effect of aqueous-phase processing on SOA formation in spring than in autumn and winter. Ox concentration in spring was as high as other three seasons. Meanwhile, LO-OOA and MO-OOA increased rapidly at moderate Ox levels when Ox changed from 50 to 70 ppb, and then remained unchanged at high Ox levels. RH maintained at low levels (RH < 40%) as Ox increased, suggesting a more important role of photochemical processing on SOA formation than aqueous-phase processing in spring."

"In summer, both LO-OOA and MO-OOA showed overall increasing trends as Ox increased, while RH showed a corresponding overall decreasing trend. This behavior indicates a strong influence of photochemical processing on both LO-OOA and MO-OOA production. LO-OOA concentration decreased significantly as LWC increased and maintained

low concentration (< 1  $\mu$ g m-3) when LWC was above 40  $\mu$ g m-3. Meanwhile, LO-OOA showed a continuously decreasing trend as RH increased in summer, except for a slightly increasing trend when RH increased from 40 to 60%, indicating photochemical processing dominated LO-OOA formation. MO-OOA increased significantly with Ox, while it increased slightly with RH (40%

Figure S21. Variations in the mass concentrations of LO-OOA and MO-OOA as a function of LWC in (a) spring, (b) summer, (c) autumn, and (d) winter.

3. Line 194-195: Similarly, it's hard to evaluate the role of aqueous-phase reactions in nitrate formation solely by the observational fact that nitrate proportion increased with PM1.

[Response] Thanks for your comments and we agree with you. We have revised the sentence as follows:

"The proportions of nitrate in  $PM_1$  increased slightly in spring, summer and autumn. In comparison, the proportion of nitrate increased first and then decreased with a high  $PM_1$  concentration ( $PM_1 > 50 \ \mu g \ m^{-3}$ ) in winter. The difference may be due to the seasonal differences of heterogeneous reactions on nitrate formation."

The role of aqueous-phase reactions in nitrate formation was detailed evaluated in Section 3.1.3.

4. The language writing needs to be polished.

[Response] Thanks for your suggestion and we have polished the language of the revised manuscript.

5. Line 146: there is a lack of evidence to support the identification/definition of FFOA.

[Response] Thanks for your reminding. We have added the evidence to support the identification/definition of FFOA in the revised manuscript.

"The mass spectrum pattern of the POA factor (Fig. S4, factor 3) mainly consisted of hydrocarbon ions ( $C_nH_{2n+1}^+$  and  $C_nH_{2n-1}^+$ ), which are commonly related to combustion emissions (Zhang et al., 2015;Sun et al., 2013). The mass profile of the POA factor had some similarity with that of HOA and coal combustion OA (CCOA) (Hu et al., 2017;Elser et al., 2016). The correlation coefficient between POA and NOx was 0.58, and that between POA and chloride was 0.78, in spring, suggesting a significant contribution of coal combustion and traffic-related sources to the POA factor in Xinglong. Moreover, HOA and CCOA show remarkably similar mass spectrum patterns when m/z is below 120 (Sun et al., 2016;Sun et al., 2018), which is sometimes difficult to be separated by PMF analysis, so FFOA can be considered as a combined factor of HOA and CCOA (Sun et al., 2018). In this study, it was difficult to separate CCOA form HOA because of the low percentage of POA in OA. Therefore, the POA factor in spring could also be considered as FFOA, which is a typical profile in Xinglong."

6. Figure 2: Figure numbers did not match with the figure capture

[Response] Thanks for your reminding. The figure numbers have matched the figure capture in the revised manuscript.

7. Correlation analysis for the PMF factors: since a regional background site, the temporal variation of most aerosol compositions can be predicted to be correlated to each other, but cannot say too much on the source appointment. For instance, in Figure 4, the LO-OOA, MO-OOA, nitrate, and sulfate were correlated well to each other.

[Response] Thanks for your comments and we agree with you that the temporal variation of most aerosol compositions can be predicted to be correlated to each other in a regional background site.

Actually, LO-OOA, MO-OOA, nitrate, and sulfate were correlated well to each other in spring, autumn and winter. However, the mass spectrum of LO-OOA and MO-OOA showed different characteristics in these three seasons. For example, in spring, the LO-OOA was characterized by a high 43/44 ratio, and the MO-OOA was defined by having a dominant peak at m/z 44 (Fig. 1). What's more, the O/C ratio of MO-OOA was obviously higher than that of LO-OOA. In addition, both of the mass spectrum and time series of LO-OOA and MO-OOA showed different characteristics in summer (Fig. 2). Therefore, the formation of LO-OOA and MO-OOA may be influenced by different chemical processes.

The similar phenomenon has been found in a background site (1570m a.s.l.) in the western Mediterranean Basin (WMB). The diurnal cycles of OA components were studied as a function of air mass origin (Fig. 3). The SV-OOA was characterized by a high 43/44 ratio, and the LV-OOA was defined by having a dominant peak at m/z 44 (Fig. 4). Both of SV-OOA and LV-OOA showed similar diurnal cycles (Fig. 3).